# Implicit Bias of Spectral Descent and Muon on Multiclass Separable Data

**Chen Fan**
Department of Computer Science
University of British Columbia

**Mark Schmidt**
Department of Computer Science
University of British Columbia
Canada CIFAR AI Chair (Amii)

**Christos Thrampoulidis**
Department of Electrical and Computer Engineering
University of British Columbia

## Abstract

Different gradient-based methods for optimizing overparameterized models can all achieve zero training error yet converge to distinctly different solutions inducing different generalization properties. We provide the first complete characterization of implicit optimization bias for p-norm normalized steepest descent (NSD) and momentum steepest descent (NMD) algorithms in multi-class linear classification with cross-entropy loss. Our key theoretical contribution is proving that these algorithms converge to solutions maximizing the margin with respect to the classifier matrix's p-norm, with established convergence rates. These results encompass important special cases including Spectral Descent and Muon, which we show converge to max-margin solutions with respect to the spectral norm. A key insight of our contribution is that the analysis of general entry-wise and Schatten p-norms can be reduced to the analysis of NSD/NMD with max-norm by exploiting a natural ordering property between all p-norms relative to the max-norm and its dual sum-norm. For the specific case of descent with respect to the max-norm, we further extend our analysis to include preconditioning, showing that Adam converges to the matrix's max-norm solution. Our results demonstrate that the multi-class linear setting, which is inherently richer than the binary counterpart, provides the most transparent framework for studying implicit biases of matrix-parameter optimization algorithms.

## 1 Introduction

The ever-increasing training cost of large language models (LLMs) has demanded better optimizer designs with improved performance and efficiency [11, 1, 23]. The de facto standard optimizers for deep learning training are Adam and AdamW [33, 39]. However, these algorithms that employ diagonal preconditioners to independently adjust the learning rate of each coordinate, may fail to capture their inter-dependencies and fully leverage the geometry of the loss landscape [79]. This has spurred a series of research efforts on improving Adam or AdamW's computational efficiency [20, 24, 51, 80], with LLM-training as the target application domain [31, 67, 46, 37].

A noticeable work by Jordan et al. [31] proposed the Muon optimizer, which was shown to have remarkable performances on NanoGPT benchmarks. More recently, it has been shown that Muon can be used for large-scale LLM training with the potential to replace AdamW as the standard choice [37]. The key step in Muon is to orthogonalize the updates via the Newton-Schulz iteration [31, 7]. More precisely, the update (denoted as $\boldsymbol{\Delta}$) is (approximately) replaced by the product of its singular-vector

Table 1: Summary of margin convergence rates for NSD and NMD algorithms of different norm constraints for linear multiclass separable data with the CE loss. The (truncated) SVDs of the gradient and momentum are denoted as $\boldsymbol{\nabla} = \boldsymbol{U}\boldsymbol{\Sigma}\boldsymbol{V}^T$ and $\boldsymbol{M} = \tilde{\boldsymbol{U}}\tilde{\boldsymbol{\Sigma}}\tilde{\boldsymbol{V}}^T$ respectively.

| Method | Norm Constraint | Update $\boldsymbol{\Delta}$ | Reference | Rate [2] |
|---|---|---|---|---|
| NGD | Unit $\|\cdot\|_2$-ball | $\frac{\boldsymbol{\nabla}}{\|\boldsymbol{\nabla}\|_2}$ | Hazan et al. [25] | - |
| NMD-GD | | $\frac{\boldsymbol{M}}{\|\boldsymbol{M}\|_2}$ | Cutkosky and Mehta [14] | - |
| SignGD | Unit $\|\cdot\|_{\max}$-ball | $\mathrm{sign}(\boldsymbol{\nabla})$ | Bernstein et al. [8] | - |
| Signum | | $\mathrm{sign}(\boldsymbol{M})$ | Bernstein et al. [8] | - |
| *Spectral-GD* | Unit $\|\|\|\cdot\|\|\|_\infty$-ball | $\boldsymbol{U}\boldsymbol{V}^T$ | Bernstein and Newhouse [7] | $O(\frac{\log t + n}{t^{1/2}})$ |
| *Muon* [1] | | $\tilde{\boldsymbol{U}}\tilde{\boldsymbol{V}}^T$ | Jordan et al. [31] | $O(\frac{d\log t + dn}{t^{1/2}})$ |

[1] We consider EMA-style momentum of the form (5).

[2] NGD and SignGD rates are the same as Spectral-GD; Signum and NMD-GD rates are the same as Muon.

matrices $\boldsymbol{U}\boldsymbol{V}^T$ (where the (truncated) singular value decomposition (SVD) of $\boldsymbol{\Delta}$ is $\boldsymbol{\Delta} = \boldsymbol{U}\boldsymbol{\Sigma}\boldsymbol{V}^T$). Even though the benefits of orthogonalization are not fully understood, Jordan et al. [31] pointed out that it could promote updates in directions of small magnitudes given the weight matrices are typically low-rank. Moreover, if the above SVD approximation is exact and gradient accumulations are turned off, then Muon becomes spectral descent [13, 7], which is the (normalized) steepest descent w.r.t the spectral norm [7]. As noted by Bernstein and Newhouse [7], spectral descent is also Shampoo (which won the AlgoPerf competition [52, 15]) without accumulations in preconditioners. Thus, Muon can be viewed as (approximate) Shampoo when both optimizers are without accumulations. In essence, we observe that one important ingredient of Muon and Shampoo (without accumulations) is the spectral-descent step,

$$\boldsymbol{W}^\dagger = \boldsymbol{W} - \eta\boldsymbol{U}\boldsymbol{V}^T \quad \text{where} \quad \nabla\mathcal{L}(\boldsymbol{W}) = \boldsymbol{U}\boldsymbol{\Sigma}\boldsymbol{V}^T.$$

Theoretical investigations of spectral descent or Muon mainly focus on characterizing the convergence rates of the algorithm (e.g., the rate of decrease of the gradient norm in the non-convex setting [3, 36, 46]). However, modern machine learning models are overparameterized, leading to multiple weight configurations that achieve identical training loss but exhibit markedly different generalization properties [78, 6]. The key insight is that gradient-based methods inherently prefer "simple" solutions according to optimizer-specific notions of simplicity. Understanding this implicit bias/regularization requires analyzing not just loss convergence, but the geometric trajectory of parameter updates throughout training. To this end, our work aims to address the fundamental question:

> *What is the **implicit bias** of **spectral descent** (and its momentum variants) in linear multiclass classification with separable data and cross-entropy loss?*

The multiclass setting where the parameter is a **matrix**, is a natural place to study the class of spectral-descent algorithms, and provides an inherently richer setting. Our work captures this richness by establishing convergence with respect to not only entry-wise matrix norms, but also matrix Schatten norms. Hence, while the focus is on spectral descent and Muon, the analysis establishes implicit bias rates for a wide family of algorithms (Table 1), and we state the results in the most general form from the perspective of steepest descent with (unit) norm-ball constraints. Our contributions are as follows:

1. For multiclass separable data trained with the cross-entropy (CE) loss, we show that the iterates of normalized steepest descent (NSD) defined with respect to (w.r.t.) any matrix entry-wise or Schatten norms converge to a solution that maximizes the margin defined w.r.t. the same norm, with a rate $\mathcal{O}(\frac{1}{t^{1/2}})$. This includes sign descent (entry-wise max-norm) [8], normalized gradient descent (entry-wise 2-norm) [25], and spectral descent (Schatten $\infty$-norm) [7] as special cases. To achieve this, we introduce a unified analysis framework that relates entry-wise and Schatten $p$-norms to the entry-wise max-norm, and construct a proxy function for the loss that closely traces both its value and gradient. We also show the same machinery applies to other multiclass losses such as the exponential loss [41] and the PairLogLoss [72].

2. Under the same setting, we utilize the same framework and proxy function to show that a $\mathcal{O}(\frac{1}{t^{1/2}})$ margin convergence rate also holds for normalized momentum steepest descent (NMD). This includes the following algorithms in analogy to the ones above: sign momentum descent [8],

normalized momentum gradient descent [14], and Muon [31]. The key step of the analysis is to use the proxy function to bound the sum-norm difference between the gradient and the momentum (i.e., the exponential moving averages (EMA) of the gradient), which translates to a bound on the dual norm through the fundamental norm-relationships used in the study of NSD. The margin convergence rates of various algorithms are summarized in Table 1. Furthermore, we extend the analysis to Adam (without the stability constant) and show its iterates maximize the margin w.r.t. the matrix max-norm (proof details and numerical validations in App. G).

3. We experimentally verify our theoretical predictions across all considered algorithms. First, for sign descent (SignGD) and Signum, we demonstrate that solutions favor the max-norm margin over the 2-norm margin—the opposite behavior to normalized gradient descent (NGD) and normalized momentum gradient descent (NMD-GD). Moreover, we show that both spectral descent (Spectral-GD) and Muon favor the spectral-norm margin over the other norms. We further extend the experiments to the non-linear setting with a two-layer neural network. We observe the (unnormalized) spectral-norm margin of Spectral-GD and Muon grow faster than other algorithms. Hence, the norm-preference trend in the linear setting can also exhibit in the non-linear setting.

## 2  Preliminaries

**Notations**  Matrices, vectors, and scalars are denoted by $\boldsymbol{A}$, $\boldsymbol{a}$, and $a$ respectively. For matrix $\boldsymbol{A}$, we denote its $(i,j)$-th entry as $\boldsymbol{A}[i,j]$, and for vector $\boldsymbol{a}$, its $i$-th entry as $\boldsymbol{a}[i]$ or $a_i$. We consider entry-wise matrix p-norms defined as $\|\boldsymbol{A}\|_p = (\sum_{i,j} |\boldsymbol{A}[i,j]|^p)^{1/p}$. Central to our results are: the infinity norm, denoted as $\|\boldsymbol{A}\|_{\max} := \|\boldsymbol{A}\|_\infty = \max_{i,j} |\boldsymbol{A}[i,j]|$ and called the **max-norm**, and the **entry-wise 1-norm**, denoted as $\|\boldsymbol{A}\|_{\mathrm{sum}} := \|\boldsymbol{A}\|_1 = \sum_{i,j} |\boldsymbol{A}[i,j]|$. The entry-wise 1-norm is dual to the max-norm. For vectors, the max-norm is equivalent to the infinity norm, denoted as $\|\boldsymbol{a}\|_\infty$, while we denote the $\ell_1$ norm as $\|\boldsymbol{a}\|_1$. We further denote the Schatten p-norm of $\boldsymbol{A}$ as $\||\boldsymbol{A}\||_p := (\sum_{i=1}^r \sigma_i^p)^{1/p}$, where $\sigma_1, \sigma_2, \ldots, \sigma_r$ are the non-zero singular values of $\boldsymbol{A}$. Let $r = \mathrm{rank}(\boldsymbol{A})$, then special cases of Schatten p-norm include: nuclear norm $\||\boldsymbol{A}\||_1 = \sum_{i=1}^r \sigma_i$, Frobenius norm $\||\boldsymbol{A}\||_2 = \sqrt{\sum_{i=1}^r \sigma_i^2}$, and **spectral norm** $\||\boldsymbol{A}\||_\infty = \sigma_1$. To simplify the discussions, we sometimes write $\|\boldsymbol{A}\|$ (dropping subscripts) to refer to any entry-wise or Schatten p-norm with $p \geq 1$. We denote by $\|\boldsymbol{A}\|_*$ the dual-norm with respect to the standard matrix inner product $\langle \boldsymbol{A}, \boldsymbol{B} \rangle = \mathrm{tr}(\boldsymbol{A}^\top \boldsymbol{B})$. We denote the gradient and its value at iteration $t$ as $\boldsymbol{\nabla} := \nabla\mathcal{L}(\boldsymbol{W})$ and $\boldsymbol{\nabla}_t := \nabla\mathcal{L}(\boldsymbol{W}_t)$ respectively.

Let $\mathbb{S} : \mathbb{R}^k \to \triangle^{k-1}$ the softmax map of $k$-dimensional vectors to the probability simplex $\triangle^{k-1}$ such that for any $\boldsymbol{a} \in \mathbb{R}^k$, it holds that $\mathbb{S}(\boldsymbol{a}) = \left[ \frac{\exp(\boldsymbol{a}[c])}{\sum_{c \in [k]} \exp(\boldsymbol{a}[c])} \right]_{c=1}^k \in \triangle^{k-1}$. Let $\mathbb{S}_c(\boldsymbol{v})$ denote the $c$-th entry of $\mathbb{S}(\boldsymbol{v})$. Let $\mathbb{S}'(\boldsymbol{a}) = \mathrm{diag}(\mathbb{S}(\boldsymbol{a})) - \mathbb{S}(\boldsymbol{a})\mathbb{S}(\boldsymbol{a})^\top$ denote the softmax gradient, with $\mathrm{diag}(\cdot)$ a diagonal matrix. Finally, let $\{\boldsymbol{e}_c\}_{c=1}^k$ be the standard basis vectors of $\mathbb{R}^k$, and indicator $\delta_{ij}$ be such that $\delta_{ij} = 1$ if and only if $i = j$. For any integer $k$, $[k]$ denotes $\{1, \ldots, k\}$.

**Setup**  Consider a multiclass classification problem with training data $\boldsymbol{h}_1, \ldots, \boldsymbol{h}_n$ and labels $y_1, \ldots, y_n$. Each datapoint $\boldsymbol{h}_i \in \mathbb{R}^d$ is a vector in a $d$-dimensional embedding space (denote data matrix $\boldsymbol{H} = [\boldsymbol{h}_1, \ldots, \boldsymbol{h}_n]^\top \in \mathbb{R}^{n \times d}$), and each label $y_i \in [k]$ represents one of $k$ classes. We assume each class contains at least one datapoint. The classifier $f_{\boldsymbol{W}} : \mathbb{R}^d \to \mathbb{R}$ is a linear model with weight matrix $\boldsymbol{W} \in \mathbb{R}^{k \times d}$. The model outputs logits $\boldsymbol{\ell}_i = f_{\boldsymbol{W}}(\boldsymbol{h}_i) = \boldsymbol{W}\boldsymbol{h}_i$ for $i \in [n]$, which are passed through the softmax map to produce class probabilities $\hat{p}(c|\boldsymbol{h}_i) = \mathbb{S}_c(\boldsymbol{\ell}_i)$.

We train using empirical risk minimization (ERM): $\mathcal{L}_{\mathrm{ERM}}(\boldsymbol{W}) := -\frac{1}{n}\sum_{i \in [n]} \ell(\boldsymbol{W}\boldsymbol{h}_i; y_i)$ , where the loss function $\ell$ takes as input the logits of a datapoint and its label. The predominant choice in classification is the CE loss

$$\mathcal{L}(\boldsymbol{W}) := -\frac{1}{n} \sum_{i \in [n]} \log\left(\mathbb{S}_{y_i}(\boldsymbol{W}\boldsymbol{h}_i)\right). \tag{1}$$

We focus our discussions on the CE loss due to its ubiquity in practice. However, our results hold for other multiclass losses such as the exponential [41] and the PairLogLoss [72] (see App. F). Define the maximum margin of the dataset w.r.t. any entry-wise or Schatten p-norm $\|\cdot\|$ as

$$\gamma := \max_{\|\boldsymbol{W}\| \leq 1} \min_{i \in [n], c \neq y_i} (\boldsymbol{e}_{y_i} - \boldsymbol{e}_c)^\top \boldsymbol{W}\boldsymbol{h}_i. \tag{2}$$

**Optimization Methods** We study iterative algorithms that update the weight matrix by

$$\boldsymbol{W}_{t+1} = \boldsymbol{W}_t - \eta_t \boldsymbol{\Delta}_t.$$

For the NSD family [10], the update direction[1] w.r.t. the norm $\|\cdot\|$ is

$$\boldsymbol{\Delta}_t := \arg\max_{\|\boldsymbol{\Delta}\| \leq 1} \langle \boldsymbol{\nabla}_t, \boldsymbol{\Delta} \rangle. \tag{3}$$

Note that this reduces to SignGD, Coordinate Descent (e.g., Nutini et al. [44]), or NGD when the max-norm (i.e. $\|\cdot\|_\infty$), the entry-wise 1-norm (i.e. $\|\cdot\|_{\text{sum}}$), or the Frobenius Euclidean-norm (i.e. $\|\cdot\|_2$) is used, respectively. Concretely, the update directions for SignGD and NGD are:

$$\text{SignGD:} \quad \boldsymbol{\Delta}_t = \texttt{sign}(\boldsymbol{\nabla}_t), \quad \text{and} \quad \text{NGD:} \quad \boldsymbol{\Delta}_t = \boldsymbol{\nabla}_t / \|\boldsymbol{\nabla}_t\|_2,$$

where the $\texttt{sign}(\cdot)$ and division $\div$ operations are applied entry-wise. In the special case of spectral norm (i.e. $\|\|\cdot\|\|_\infty$), this becomes the Spectral-GD, for which $\boldsymbol{\Delta}_t = \boldsymbol{U}_t \boldsymbol{V}_t^T$, where $\boldsymbol{U}_t$ and $\boldsymbol{V}_t$ are the left/right singular matrices of $\boldsymbol{\nabla}_t$ respectively (i.e., $\boldsymbol{\nabla}_t = \boldsymbol{U}_t \boldsymbol{\Sigma}_t \boldsymbol{V}_t^T$ with singular values in $\boldsymbol{\Sigma}_t > 0$ arranged in non-increasing order). Finally, note that the Schatten 2-norm case reduces to NGD (as $\|\|\cdot\|\|_2 = \|\cdot\|_2$).

We also consider the NMD family with the following update direction w.r.t. the norm $\|\cdot\|$

$$\boldsymbol{\Delta}_t := \arg\max_{\|\boldsymbol{\Delta}\| \leq 1} \langle \boldsymbol{M}_t, \boldsymbol{\Delta} \rangle, \tag{4}$$

where the momentum $\boldsymbol{M}_t$ is computed as the EMA of the gradient given by

$$\boldsymbol{M}_t = \beta_1 \boldsymbol{M}_{t-1} + (1 - \beta_1) \boldsymbol{\nabla}_t. \tag{5}$$

This form of momentum is also known as the heavy-ball or the SGDM-style momentum [47, 19, 38]. Thus, an NMD algorithm chooses the update direction (among all feasible directions in the unit $\|\cdot\|$-ball) that best aligns with the momentum instead of the gradient direction (as chosen by an NSD algorithm). Similar to above, when the max-norm and the Frobenius-norm are used, the resulting Signum and NMD-GD update directions are:

$$\text{Signum:} \quad \boldsymbol{\Delta}_t = \texttt{sign}(\boldsymbol{M}_t), \quad \text{and} \quad \text{NMD-MD:} \quad \boldsymbol{\Delta}_t = \boldsymbol{M}_t / \|\boldsymbol{M}_t\|_2.$$

When spectral norm is used in (4), this becomes Muon[2] for which the SVD is on $\boldsymbol{M}_t$ (i.e. $\boldsymbol{M}_t = \tilde{\boldsymbol{U}}_t \tilde{\boldsymbol{\Sigma}}_t \tilde{\boldsymbol{V}}_t^T$) and the update direction is $\boldsymbol{\Delta}_t = \tilde{\boldsymbol{U}}_t \tilde{\boldsymbol{V}}_t^T$. Note that Muon reduces to Spectral-GD when the momentum parameter $\beta_1$ is set to 0. Similar reductions also hold for Signum (to SignGD) and NMD-GD (to NGD) as well.

**Assumptions** Establishing the implicit bias of the above mentioned gradient-based optimization algorithms requires the following assumptions. First, we assume data are linearly separable, ensuring the margin $\gamma$ is strictly positive, an assumption routinely used in previous works [53, 48, 22, 43, 75].

**Assumption 1.** *There exists $\boldsymbol{W} \in \mathbb{R}^{k \times d}$ such that $\min_{c \neq y_i} (\boldsymbol{e}_{y_i} - \boldsymbol{e}_c)^T \boldsymbol{W} \boldsymbol{h}_i > 0$ for all $i \in [n]$.*

In this work, we consider learning rate schedule $\eta_t = \Theta(\frac{1}{t^a})$, where $a \in (0, 1]$. Such schedules have been studied in the convergence and implicit bias of various optimization algorithms (e.g., Bottou et al. [9], Nacson et al. [43], and Sun et al. [55]) including Adam [77].

**Assumption 2.** *The learning rate schedule $\{\eta_t\}$ is decreasing with respect to $t$ and satisfies the following conditions: $\lim_{t \to \infty} \eta_t = 0$ and $\sum_{t=0}^\infty \eta_t = \infty$.*

Assumption 3 can be satisfied by the above learning rate for a sufficiently large $t$ as shown in Zhang et al. [77, Lemma C.1]. It is used in our analysis of NMD and Adam.

**Assumption 3.** *The learning rate schedule satisfies the following: let $\beta \in (0, 1)$ and $c_1 > 0$ be two constants, there exist time $t_0 \in \mathbb{N}_+$ and constant $c_2 = c_2(c_1, \beta) > 0$ such that $\sum_{s=0}^t \beta^s (e^{c_1 \sum_{\tau=1}^s \eta_{s-\tau}} - 1) \leq c_2 \eta_t$ for all $t \geq t_0$.*

Finally, we assume that the 1-norm of the data is bounded. Similar assumptions were used in Ji and Telgarsky [26], Nacson et al. [43], Wu et al. [75], and Zhang et al. [77].

**Assumption 4.** *There exists constant $B > 0$ such that $\|\boldsymbol{h}_i\|_1 \leq B$ for all $i \in [n]$.*

---

[1]For $p \in (1, \infty)$, the norms $\|\cdot\|_p$ and $\|\|\cdot\|\|_p$ are strictly convex, thus there is a unique maximizer defining the update in Eqn. (3). For $p = 1, \infty$ the maximizer is not necessarily unique and our results hold for any choice of $\boldsymbol{\Delta}_t$ in the set of maximizers; see e.g. Ziętak [81].

[2]The implementation in Jordan et al. [31] uses Nesterov-type momentum: Newton-Schulz iteration applied to $\beta_1 \boldsymbol{M}_t + \boldsymbol{\nabla}_t$ instead of $\beta_1 \boldsymbol{M}_{t-1} + \boldsymbol{\nabla}_t$ [37].

## 3 A Unified Framework with a Proxy Function

Analyzing margin convergence begins with studying loss convergence through second-order Taylor expansion of the CE loss (recall that $\mathbb{S}'(\boldsymbol{v}) = \text{diag}(\boldsymbol{v}) - \boldsymbol{v}\boldsymbol{v}^\top$):

$$\mathcal{L}(\boldsymbol{W} + \boldsymbol{\Delta}) = \mathcal{L}(\boldsymbol{W}) + \langle \nabla\mathcal{L}(\boldsymbol{W}), \boldsymbol{\Delta}\rangle + \frac{1}{2n}\sum_{i \in [n]} \boldsymbol{h}_i^\top \boldsymbol{\Delta}^\top \mathbb{S}'(\boldsymbol{W}\boldsymbol{h}_i)\boldsymbol{\Delta}\boldsymbol{h}_i + o(\|\boldsymbol{\Delta}\|_F^3), \quad (6)$$

To bound the loss at $\boldsymbol{W}_{t+1} = \boldsymbol{W}_t - \eta_t\boldsymbol{\Delta}_t$, we must bound both the first-order and second-order terms in (6). For NSD updates in Eq. (3), the first term evaluates to $-\eta_t\|\nabla\mathcal{L}(\boldsymbol{W})\|_*$ (recall that $\|\cdot\|_*$ is the dual norm). This leads to two key tasks: (1) Lower-bounding the dual gradient norm; (2) Upper-bounding the second-order term.

For the proof to proceed, these bounds should satisfy two desiderata: (1) They are expressible as the same function of $\boldsymbol{W}$, call it $\mathcal{G}(\boldsymbol{W})$, up to constants. (2) The function $\mathcal{G}(\boldsymbol{W})$ is a good proxy for the loss for small values of the latter. The former helps with combining the terms, while the latter helps with demonstrating descent. Next, we obtain these key bounds for the CE loss by determining the appropriate proxy $\mathcal{G}(\boldsymbol{W})$.

Besides the need for a proxy $\mathcal{G}(\boldsymbol{W})$, we use the following facts about the sum-norm dominating any entry-wise/Schatten p-norm. Concretely, for any matrix $\boldsymbol{A}$ and any $p \geq 1$:

$$\|\boldsymbol{A}\|_{\max} \leq |\|\boldsymbol{A}\|\|_p \leq \|\boldsymbol{A}\|_{\text{sum}}, \quad \text{and} \quad \|\boldsymbol{A}\|_{\max} \leq \|\boldsymbol{A}\|_p \leq \|\boldsymbol{A}\|_{\text{sum}}. \quad (7)$$

These relationships (proved in Lemma 11 in App. C) are crucial for unifying the analysis of NSD and NMD algorithms w.r.t. either the entry-wise or the Schatten norms (details below).

**Construction of $\mathcal{G}(\boldsymbol{W})$**   Before showing our construction for the CE loss, it is insightful to discuss how previous works do this in the binary case with labels $y_{b,i} \in \{\pm 1\}$, classifier vector $\boldsymbol{w} \in \mathbb{R}^d$ and binary margin $\gamma_b := \max_{\|\boldsymbol{w}\| \leq 1} \min_{i \in [n]} y_{b,i}\boldsymbol{w}^\top\boldsymbol{h}_i$. For exponential loss, Gunasekar et al. [22] showed that $\|\nabla\mathcal{L}(\boldsymbol{w})\| \geq \gamma_b\mathcal{L}(\boldsymbol{w})$. For logistic loss $\ell(t) = \log(1 + \exp(-t))$, Zhang et al. [77] proved $\|\nabla\mathcal{L}(\boldsymbol{w})\|_1 \geq \gamma_b\mathcal{G}(\boldsymbol{w})$, where $\mathcal{G}(\boldsymbol{w}) = \frac{1}{n}\sum_{i=1}^n |\ell'(y_{b,i}\boldsymbol{w}^\top\boldsymbol{h}_i)|$ and $\ell'$ is the first-order derivative. In both cases, one can take the common form $\mathcal{G}_b(\boldsymbol{w}) = \frac{1}{n}\sum_{i=1}^n |\ell'(y_{b,i}\boldsymbol{w}^\top\boldsymbol{h}_i)|$. The proof relies on showing $\gamma \leq \min_{\boldsymbol{r} \in \triangle^{n-1}} \|\boldsymbol{H}^T\boldsymbol{r}\|$ via Fenchel Duality [59, 22] and appropriately choosing $\boldsymbol{r}$.

In the multiclass setting, where the loss function is vector-valued, it is unclear how to extend the binary proof or definition of $\mathcal{G}(\boldsymbol{W})$. To this end, we realize that the key is in the proper manipulation of the gradient inner product $\langle \boldsymbol{A}, -\nabla\mathcal{L}(\boldsymbol{W})\rangle$ (for arbitrary matrix $\boldsymbol{A} \in \mathbb{R}^{k \times d}$). The CE gradient evaluates to $\nabla\mathcal{L}(\boldsymbol{W}) = \frac{1}{n}\sum_{i=1}^n (\boldsymbol{e}_{y_i} - \mathbb{S}(\boldsymbol{W}\boldsymbol{h}_i))\boldsymbol{h}_i^\top$ and using the fact that $\mathbb{S}(\boldsymbol{W}\boldsymbol{h}_i) \in \triangle^{k-1}$, it turns out that we can express (details in Lemma 9): $\langle \boldsymbol{A}, -\nabla\mathcal{L}(\boldsymbol{W})\rangle = \frac{1}{n}\sum_{i \in [n]}\sum_{c \neq y_i} \mathbb{S}_c(\boldsymbol{W}\boldsymbol{h}_i)(\boldsymbol{e}_{y_i} - \boldsymbol{e}_c)^\top\boldsymbol{A}\boldsymbol{h}_i$. This motivates defining $\mathcal{G}(\boldsymbol{W})$ as:

$$\mathcal{G}(\boldsymbol{W}) := \frac{1}{n}\sum_{i \in [n]}(1 - \mathbb{S}_{y_i}(\boldsymbol{W}\boldsymbol{h}_i)). \quad (8)$$

The lemma below, following from the inner-product calculation above and our definition of $\mathcal{G}(\boldsymbol{W})$, confirms this is the right choice. For convenience, denote $s_{ic} := \mathbb{S}_c(\boldsymbol{W}\boldsymbol{h}_i)$, for $i \in [n], c \in [k]$.

**Lemma 1** (Lower bounding the gradient dual-norm). *For any $\boldsymbol{W} \in \mathbb{R}^{k \times d}$ and any entry-wise or Schatten p-norm $\|\cdot\|$ with $p \geq 1$, it holds that $\|\nabla\mathcal{L}(\boldsymbol{W})\|_* \geq \gamma \cdot \mathcal{G}(\boldsymbol{W})$, where $\|\cdot\|_*$ is the dual-norm.*

The lemma completes the first task: lower bounding the gradient's dual norm. Importantly, the factor in front of $\mathcal{G}(\boldsymbol{W})$ is the margin $\gamma$ w.r.t. the norm $\|\cdot\|$, which is crucial in the forthcoming analysis.

**$\mathcal{G}(\boldsymbol{W})$ and second-order term**   We now show how to bound the second-order term in (6). For this, we establish the following essential lemma whose proof relies on the relationships in (7).

**Lemma 2.** *For any entry-wise or Schatten p-norm $\|\cdot\|$ with $p \geq 1$, any $\boldsymbol{s} \in \Delta^{k-1}$ in the k-dimensional simplex, any index $c \in [k]$, and $\boldsymbol{v} \in \mathbb{R}^k$, it holds that*

$$\boldsymbol{v}^\top\left(\text{diag}(\boldsymbol{s}) - \boldsymbol{s}\boldsymbol{s}^\top\right)\boldsymbol{v} \leq 4(1 - s_c)\|\boldsymbol{v}\boldsymbol{v}^T\|.$$

*Proof.* Let $\boldsymbol{S} := \mathrm{diag}(\boldsymbol{s}) - \boldsymbol{s}\boldsymbol{s}^\top$ and $q \geq 1$ such that $1/p + 1/q = 1$. By norm duality, it holds that

$$\boldsymbol{v}^\top \boldsymbol{S} \boldsymbol{v} = \mathrm{tr}\left(\boldsymbol{S}\boldsymbol{v}\boldsymbol{v}^\top\right) \leq \|\boldsymbol{S}\|_q \|\boldsymbol{v}\boldsymbol{v}^\top\| \leq \|\boldsymbol{S}\|_{\mathrm{sum}} \|\boldsymbol{v}\boldsymbol{v}^\top\|,$$

where $\|\cdot\|_q$ is the dual of $\|\cdot\|$ and the second inequality is by (7). Direct calculation yields $\|\boldsymbol{S}\|_{\mathrm{sum}} = 2\sum_{c\in[k]} s_c(1-s_c)$. The advertised bound then follows by noting the following $\sum_{c\in[k]} s_c(1-s_c) \leq 2(1-s_{c'})$ for any $c' \in [k]$ (verified in Lemma 13 in App. C). □

Next, we apply the above lemma with $\boldsymbol{v} \leftarrow \boldsymbol{\Delta}\boldsymbol{h}_i$ and $c \leftarrow y_i$, and further use the inequalities: $\|\boldsymbol{v}\boldsymbol{v}^T\|_p = \|\boldsymbol{v}\|_p^2 \leq \|\boldsymbol{\Delta}\|_p^2 \|\boldsymbol{h}\|_q^2$ for entry-wise norms and $\left\|\left|\boldsymbol{v}\boldsymbol{v}^\top\right|\right\|_p = \|\boldsymbol{v}\|_2^2 \leq \||\boldsymbol{\Delta}|\|_\infty^2 \|\boldsymbol{h}\|_2^2 \leq \||\boldsymbol{\Delta}|\|_p^2 \|\boldsymbol{h}\|_2^2$ for Schatten norms. Together with Ass. 4, this upper bounds the second-order term in the CE loss expansion in terms of the proxy function:

$$2B^2 \|\boldsymbol{\Delta}\|^2 \cdot \frac{1}{n} \sum_{i\in[n]} \left(1 - \mathbb{S}_{y_i}(\boldsymbol{W}\boldsymbol{h}_i)\right).$$

**Properties of $\mathcal{G}(\boldsymbol{W})$**  We now show that $\mathcal{G}(\boldsymbol{W})$ meets the second desiderata: being a good proxy for the loss $\mathcal{L}(\boldsymbol{W})$. This is rooted in the elementary relationships between $\mathcal{G}(\boldsymbol{W})$ and $\mathcal{L}(\boldsymbol{W})$, which are used in the various parts of the proof. Below, we summarize these key relationships.

**Lemma 3** (Properties of $\mathcal{G}(\boldsymbol{W})$ and $\mathcal{L}(\boldsymbol{W})$). *Let $\boldsymbol{W} \in \mathbb{R}^{k \times d}$. The followings hold: (i) Under Ass. 4, $2B \cdot \mathcal{G}(\boldsymbol{W}) \geq \|\nabla \mathcal{L}(\boldsymbol{W})\|_*$; (ii) $1 \geq \frac{\mathcal{G}(\boldsymbol{W})}{\mathcal{L}(\boldsymbol{W})} \geq 1 - \frac{n\mathcal{L}(\boldsymbol{W})}{2}$; (iii) If $\boldsymbol{W}$ satisfies $\mathcal{L}(\boldsymbol{W}) \leq \frac{\log 2}{n}$ or $\mathcal{G}(\boldsymbol{W}) \leq \frac{1}{2n}$, then $\mathcal{L}(\boldsymbol{W}) \leq 2\mathcal{G}(\boldsymbol{W})$.*

Lemma 3 (i) extends Lemma 1 by establishing a sandwich relationship between $\mathcal{G}(\boldsymbol{W})$ and the gradient's dual norm. The lemma's statements (ii) and (iii) show that $\mathcal{G}(\boldsymbol{W})$ can substitute for the loss - it lower bounds $\mathcal{L}(\boldsymbol{W})$ and serves as an upper bound when either $\mathcal{L}(\boldsymbol{W})$ or $\mathcal{G}(\boldsymbol{W})$ is sufficiently small. Specifically, the ratio $\mathcal{G}(\boldsymbol{W})/\mathcal{L}(\boldsymbol{W})$ converges to 1 as the loss decreases, with the convergence rate depending on the rate of loss decrease. The key property (ii) may seem algebraically complex, but it turns out (details in Lemma 18 in App. C) that both sides of the sandwich relationship follow from the elementary fact that $\forall x > 0 : 1 - x \leq e^{-x} \leq 1 - x + x^2/2$.

## 4 Implicit Bias of Normalized Steepest Descent

We now leverage our construction of $\mathcal{G}(\boldsymbol{W})$ to show that the margin of NSD's iterates converges to the data margin defined w.r.t. the same entry-wise or Schatten p-norm that is used to define the algorithm (refer to eqns. (2) and (3) for the definitions of margin and NSD). We only highlight the key steps in the proof and defer details to App. D.

**NSD Descent**  We start by showing a descent property. By applying Lemmas 1 and 2 to lower and upper bound the first and second order terms in Eq. (6) yields

$$\mathcal{L}(\boldsymbol{W}_{t+1}) \leq \mathcal{L}(\boldsymbol{W}_t) - \gamma\eta_t \mathcal{G}(\boldsymbol{W}_t) + 2\eta_t^2 B^2 \mathcal{G}(\boldsymbol{W}_t) \sup_{\zeta \in [0,1]} \frac{\mathcal{G}(\boldsymbol{W}_t - \zeta\eta_t \boldsymbol{\Delta}_t)}{\mathcal{G}(\boldsymbol{W}_t)}.$$

Algebraic manipulations of the definition of $\mathcal{G}(\boldsymbol{W})$ and the relationships in (7) allow us to bound the ratio in the right hand side.

**Lemma 4** (Ratio of $\mathcal{G}(\boldsymbol{W})$). *For any $\psi \in [0,1]$, we have the following: $\frac{\mathcal{G}(\boldsymbol{W} - \psi\eta\boldsymbol{\Delta})}{\mathcal{G}(\boldsymbol{W})} \leq e^{2B\eta\psi\|\boldsymbol{\Delta}\|_{\mathrm{max}}} \leq e^{2B\eta\psi\|\boldsymbol{\Delta}\|}$ (note that the second inequality is by (7)).*

From this and $\|\boldsymbol{\Delta}_t\| \leq 1$ for NSD, we obtain

$$\mathcal{L}(\boldsymbol{W}_{t+1}) \leq \mathcal{L}(\boldsymbol{W}_t) - \gamma\eta_t(1 - \alpha_{s_1}\eta_t)\mathcal{G}(\boldsymbol{W}_t), \tag{9}$$

where $\alpha_{s_1} = 2B^2 e^{2B\eta_0}/\gamma$. Given a decay learning rate of the form $\eta_t = \Theta(\frac{1}{t^a})$, we can conclude that the loss starts to monotonically decrease after some time.

**NSD Unnormalized Margin**   We now use the descent property in (9) to lower bound the un-normalized margin. An intermediate result towards this is recognizing that sufficiently small loss $\mathcal{L}(\boldsymbol{W}) \leq \frac{\log 2}{n}$ guarantees $\boldsymbol{W}$ separates the data (Lemma 19 in App. C). The descent property ensures that NSD iterates will eventually achieve this loss threshold, thereby guaranteeing separability. The main result of this section, shows that eventually the iterates achieve separability with a substantial (unnormalized) margin.

**Lemma 5** (NSD Unnormalized Margin). *Assume there exists $\tilde{t}$ such that $\mathcal{L}(\boldsymbol{W}_t) \leq \frac{\log 2}{n}, \forall t > \tilde{t}$. Then, it holds that for all $t \geq \tilde{t}$ ($\alpha_{s_2} = 2B^2 e^{2B\eta_0}$)*

$$\min_{i \in [n], c \neq y_i} (\boldsymbol{e}_{y_i} - \boldsymbol{e}_c)^T \boldsymbol{W}_t \boldsymbol{h}_i \geq \gamma \sum_{s=\tilde{t}}^{t-1} \eta_s \frac{\mathcal{G}(\boldsymbol{W}_s)}{\mathcal{L}(\boldsymbol{W}_s)} - \alpha_{s_2} \sum_{s=\tilde{t}}^{t-1} \eta_s^2. \tag{10}$$

**NSD Margin Convergence**   Proceeding from Eq. (10) requires showing the convergence of the ratio $\frac{\mathcal{G}(\boldsymbol{W})}{\mathcal{L}(\boldsymbol{W})}$. The two key ingredients are given in Lemma 3 (ii) and (iii). Lemma 3 (ii) suggests that it is sufficient to study the convergence of $\mathcal{L}(\boldsymbol{W})$, which is captured in (9). However, to obtain an explicit rate via (9), we need to rewrite $\mathcal{G}(\boldsymbol{W}_t)$ in terms of $\mathcal{L}(\boldsymbol{W}_t)$. This is where Lemma 3 (iii) helps. Putting them together, we arrive at the following theorem (see Thm. 3 and Cor. 1 for details).

**Theorem 1.** *Suppose that Ass. 1, 2, and 4 hold. Set learning rate $\eta_t = \Theta(\frac{1}{t^{1/2}})$. The following holds for the margin gap of NSD's iterates*

$$\gamma - \frac{\min_{i \in [n], c \neq y_i} (\boldsymbol{e}_{y_i} - \boldsymbol{e}_c)^T \boldsymbol{W}_t \boldsymbol{h}_i}{\|\boldsymbol{W}_t\|} \leq \mathcal{O}(\frac{\log t + n}{t^{1/2}}).$$

**Remark 1.** *For margin convergence rates of NSD, Nacson et al. [43] showed a rate of $\mathcal{O}(\frac{\log t}{t^{1/2}})$ in the binary setting, limited to the entry-wise p-norms and the exponential loss. Compared to this, our results hold for the more practical setting of multiclass data and CE loss. To the best of our knowledge, this is the first non-asymptotic result on the implicit bias of spectral-GD for linear multiclass separable data, and it holds for other p-norms as well. Upon completion of this work, we became aware of an update on the arXiv version of Tsilivis et al. [63], which includes an extension of their previous results to steepest descent w.r.t. the spectral norm. In comparison to ours, their gradient-flow analysis applies to homogeneous neural networks with the restriction of infinitesimal step-sizes. Moreover, it does not include normalization nor momentum (like Muon, which we analyze), and the convergence is (asymptotic) to a KKT point of a spectral-norm margin maximization problem.*

## 5   Implicit Bias of Normalized Momentum Steepest Descent

In this section, we study the implicit bias of NMD algorithms (proof details in App E). Similar to Sec. , its updates are defined w.r.t. either the entry-wise or the Schatten norm $\|\cdot\|$. The analysis relies on the relationships in (7) and we show the same proxy function $\mathcal{G}(\boldsymbol{W})$ naturally appears. Given that the NMD updates satisfy $\|\boldsymbol{\Delta}\| \leq 1$, the second-order term in (6) is bounded in the same way as NSD. The main difference is in bounding the first-order term as shown by the following lemma.

**Lemma 6.** *Let $\boldsymbol{\Omega}_t := \boldsymbol{M}_t - \boldsymbol{\nabla}_t$. It holds for all $t \geq 0$ that*

$$\langle \boldsymbol{\nabla}_t, \boldsymbol{W}_{t+1} - \boldsymbol{W}_t \rangle \leq 2\eta \|\boldsymbol{\Omega}_t\|_* - \eta\gamma\mathcal{G}(\boldsymbol{W}_t).$$

Given the relationships in (7) hold for any $p \geq 1$, we can bound the dual norm of $\|\boldsymbol{\Omega}_t\|_*$ via its sum norm (i.e. $\|\boldsymbol{\Omega}_t\|_* \leq \|\boldsymbol{\Omega}\|_{\text{sum}}$). Given the goal is to bound all the terms in the Taylor expansion (6) via the proxy function $\mathcal{G}(\boldsymbol{W}_t)$, an natural next step is to bound $\|\boldsymbol{\Omega}_t\|_{\text{sum}}$ using the same proxy function. To do this, we decompose the proxy function **per-class-wise**, and apply the per-class proxy functions to bound the entries of $\boldsymbol{\Omega}_t$ associated with their corresponding classes. Concretely, we write the function $\mathcal{G}(\boldsymbol{W})$ in two equivalent ways: $\mathcal{G}(\boldsymbol{W}) = \sum_{c \in [k]} \frac{1}{n} \sum_{i \in [n], y_i = c} (1 - s_{iy_i}) = \sum_{c \in [k]} \frac{1}{n} \sum_{i \in [n], y_i \neq c} s_{ic}$, which motivate the following definitions of the per-class proxy functions:

$$\mathcal{G}_c(\boldsymbol{W}) := \frac{1}{n} \sum_{i \in [n], y_i = c} (1 - s_{iy_i}), \quad \text{and} \quad \mathcal{Q}_c(\boldsymbol{W}) := \frac{1}{n} \sum_{i \in [n], y_i \neq c} s_{ic}.$$

Next, we bound the entries in each row of $\boldsymbol{\Omega}_t$ (thus belonging to the same class) via the corresponding proxy functions $\mathcal{G}_c(\boldsymbol{W}_t)$ and $\mathcal{Q}_c(\boldsymbol{W}_t)$ to arrive at the following lemma. Its proof utilizes the nice properties of softmax map given in Lemma 15 in App. B.

**Lemma 7.** *Suppose that Ass. 1, 2, 3, and 4 hold. Let $c \in [k]$ and $j \in [d]$. There exists time $t_0$ such that for all $t \geq t_0$ and for $\alpha_M := B(1 - \beta_1)c_2$:*

$$|\mathbf{M}_t[c,j] - (1 - \beta_1^{t+1})\nabla\mathcal{L}(\boldsymbol{W}_t)[c,j]| \leq \alpha_M \eta_t \big(\mathcal{G}_c(\boldsymbol{W}_t) + \mathcal{Q}_c(\boldsymbol{W}_t)\big).$$

Given the result in Lemma 7, we can show that $|\boldsymbol{\Omega}_t[c,j]| \leq \beta_1^{t+1}|\nabla\mathcal{L}(\boldsymbol{W}_t)[c,j]| + \alpha_M \eta_t \mathcal{T}_c(\boldsymbol{W}_t)$, where $\mathcal{T}_c(\boldsymbol{W}_t)$ is defined to be $\mathcal{T}_c(\boldsymbol{W}_t) := \mathcal{G}(\boldsymbol{W}_t) + \mathcal{Q}(\boldsymbol{W}_t)$. Then, we sum over indices $c \in [k]$ and $j \in [d]$ and apply $\|\boldsymbol{\nabla}\|_{\text{sum}} \leq 2B \cdot \mathcal{G}(\boldsymbol{W})$ (from Lemma 3 (i)) to obtain:

**Lemma 8.** *It holds for all $t \geq 0$ that $\|\boldsymbol{\Omega}_t\|_{\text{sum}} \leq 2B\beta_1^{t/2}\mathcal{G}(\boldsymbol{W}_t) + 2\alpha_M d\eta_t \mathcal{G}(\boldsymbol{W}_t)$.*

This completes the bound on the first-order term for NMD algorithms via the proxy $\mathcal{G}(\boldsymbol{W})$. The rest proof follows similar steps as NSD. We note that without the above per-class decomposition, an extra $k$-factor would appear in the second term of the bound on $\|\boldsymbol{\Omega}_t\|_{\text{sum}}$ (and thus also show up in the final rate). We state the main theorem for NMD algorithms.

**Theorem 2.** *Under the setting of Lem. 7, the margin gap of NMD with $\eta_t = \Theta(\frac{1}{t^{1/2}})$ is $O(\frac{d\log t + dn}{t^{1/2}})$.*

**Remark 2.** *Wang et al. [69] studied implicit bias of un-normalized GD with momentum, and showed its iterates converge asymptotically to the max 2-norm margin solution. In contrast, our rates are non-asymptotic and cover a much wider family of algorithms converging to non-Euclidean geometric margins (w.r.t. entry-wise/Schatten norms). Note the convergence rate of NMD matches that of NSD (Thm. 1) up to a factor of $d$. It could be interesting to remove this dependence in a future work.*

**Implicit Bias of Adam** Finally, observing that Adam [33] (without the stability constant, i.e., eqns (33a), (33b), and (33c) in App. G) shares the same form of momentum as NMD and the (entry-wise) updates are bounded by some constant as shown in Zhang et al. [80] and Xie and Li [76]. Thus, our analysis extends to Adam. Concretely, a similar proof strategy can be adapted once a bound on the second gradient moment via the proxy function is established (Lemma 35). In App. G, we prove a $O(\frac{d\log(t) + nd}{t^{1/3}})$ max-norm margin convergence rate for Adam (details in Thm. 5 and Cor. 2).

# 6 Experiments

**Synthetic Experiments** We generate snythetic multiclass separable data as follows: $k = 10$ class centers are sampled from a standard normal distribution; within each class, data is sampled from normal distribution $\mathcal{N}(0, \sigma^2 I), \sigma = 0.1$. We set $d = 25$, sample 50 data points for each class, and ensure that margin is positive (thus data is separable). We run different algorithms to minimize CE loss using $\eta_t = \frac{\eta_0}{t^a}$ ($\eta_0 = 0.1$ for SignGD and NGD; $\eta_0 = 0.05$ for Spectral-GD and Muon), where (based on our theorems) $a$ is set to $1/2$. We apply truncated SVD on the gradient and momentum for Spectral-GD and Muon respectively. Data margins w.r.t. different norms are found via CVXPY [17]. We denote max-margin classifiers defined w.r.t. the 2-norm, the max-norm, and the spectral-norm as $\boldsymbol{V}_2$, $\boldsymbol{V}_\infty$, and $\boldsymbol{V}_{\text{spec}}$, respectively. Based on the margin-gap results in Figure 1, we observe that SignGD, NGD, and Spectral-GD favor max-norm, 2-norm, and spectral-norm margin respectively. Besides this, the behavior of Muon is very similar to that of Spectral-GD (in agreement with our theories). Figure 2 further confirms that the iterates of these algorithms correlate well with the corresponding max margin separators. Experiments on Signum, NMD-GD, and Adam are provided in App. A and App. G.

**Two-layer Neural Network** We extend the experiments to the non-linear classification setting using the cross-entropy loss. We sample 100 data points from each of the 10 classes of the MNIST dataset [35]. The model is a two-layer neural network with the hidden dimension being 100 (the first and second layer weights are denoted as $\boldsymbol{V}$ and $\boldsymbol{W}$ respectively). The training is done in two ways: (a) Train the first-layer weight with the second-layer weight fixed and (b) Train both the first and second-layer weights. For options (a) and (b), the respective spectral-norm margins of the overall network are defined as

$$\text{(a) } \gamma_a^{\boldsymbol{V}} := \min_{i \in [n], c \neq y_i} \frac{(\boldsymbol{e}_{y_i} - \boldsymbol{e}_c)^T \boldsymbol{W}\sigma(\boldsymbol{V}\boldsymbol{h}_i)}{\||\boldsymbol{V}|\|_\infty}, \quad \text{(b) } \gamma_b^{\boldsymbol{V},\boldsymbol{W}} := \min_{i \in [n], c \neq y_i} \frac{(\boldsymbol{e}_{y_i} - \boldsymbol{e}_c)^T \boldsymbol{W}\sigma(\boldsymbol{V}\boldsymbol{h}_i)}{\max\{\||\boldsymbol{V}|\|_\infty, \||\boldsymbol{W}|\|_\infty\}},$$

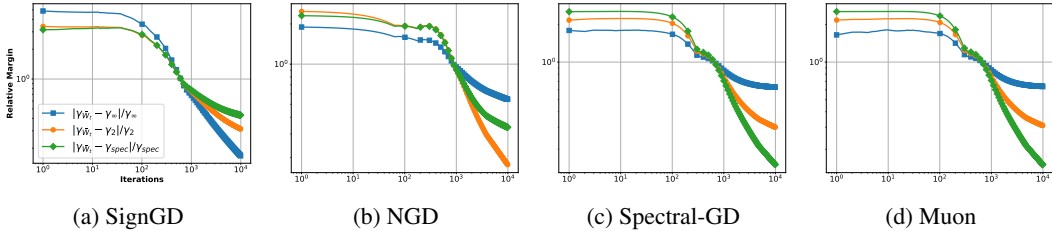

(a) SignGD    (b) NGD    (c) Spectral-GD    (d) Muon

Figure 1: **(a)** We normalize the iterates of SignGD w.r.t. the max-norm (denoted as $\bar{\boldsymbol{W}}_t$), compute the margin (denoted as $\gamma_{\bar{\boldsymbol{W}}_t}$), then plot its difference to data margins $\gamma_{\|\cdot\|_\infty}, \gamma_{\|\cdot\|_2}$, and $\gamma_{\||\cdot\||_\infty}$ (note that the margin difference is further divided by the corresponding data margin for comparisons). SignGD favors the margin defined w.r.t. the max-norm. **(b, c, and d)** Same as (a) with SignGD (max-norm) replaced by NGD (2-norm), Spectral-GD (spectral-norm), and Muon (spectral-norm), respectively. NGD favors the 2-norm margin, while Spectral-GD and Muon favor the spectral-norm margin.

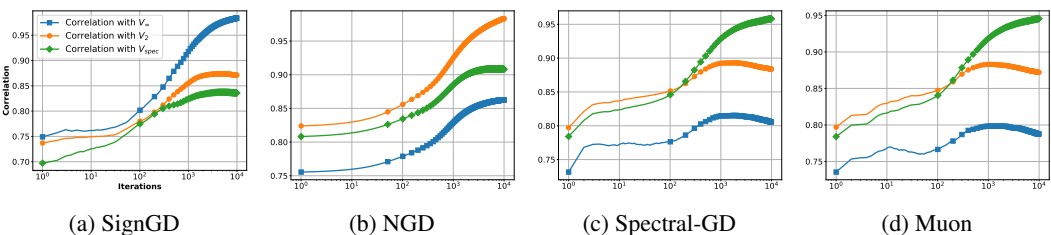

(a) SignGD    (b) NGD    (c) Spectral-GD    (d) Muon

Figure 2: **(a)** Correlations between the iterates of SignGD ($\boldsymbol{W}_t$) and max margin separators $\boldsymbol{V}_\infty$, $\boldsymbol{V}_2$, and $\boldsymbol{V}_{\text{spec}}$ against iterations (correlation defined as $\frac{\langle \boldsymbol{W}, \boldsymbol{V} \rangle}{\|\boldsymbol{W}\|_2 \|\boldsymbol{V}\|_2}$). **(b, c, and d)** Same as (a) with SignGD replaced by NGD, Spectral-GD, and Muon, respectively. SignGD and NGD correlate well with $\boldsymbol{V}_\infty$ and $\boldsymbol{V}_2$, respectively, while Spectral-GD and Muon correlate well with $\boldsymbol{V}_{\text{spec}}$.

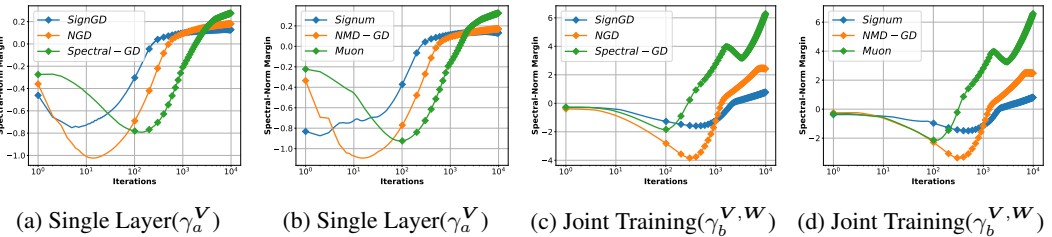

(a) Single Layer($\gamma_a^{\boldsymbol{V}}$)  (b) Single Layer($\gamma_a^{\boldsymbol{V}}$)  (c) Joint Training($\gamma_b^{\boldsymbol{V},\boldsymbol{W}}$)  (d) Joint Training($\gamma_b^{\boldsymbol{V},\boldsymbol{W}}$)

Figure 3: **(a)** Spectral-norm margin $\gamma_a^{\boldsymbol{V}_t}$ as a function of $t$ for SignGD, NGD, and Spectral-GD. **(b)** Same as (a) with algorithms replaced by Signum, NMD-GD, and Muon, respectively. **(c)** Spectral-norm margin $\gamma_b^{\boldsymbol{V}_t,\boldsymbol{W}_t}$ as a function of $t$ for SignGD, NGD, and Spectral-GD. **(d)** Same as (c) with algorithms replaced by Signum, NMD-GD, and Muon, respectively.

where $\sigma(\cdot)$ is the sigmoid function. In Figure 3, we track the quantities $\gamma_a^{\boldsymbol{V}_t}$ (Figure 3a and 3b) and $\gamma_a^{\boldsymbol{V}_t,\boldsymbol{W}_t}$ (Figure 3c and 3d) as a function of the iteration counter $t$. For both definitions, we observe that the spectral-norm margin of the iterates of Spectral-GD and Muon grow faster than other algorithms. Hence, the observations in the linear settings can also hold in the non-linear settings.

## 7 Related Works

Starting with GD, the foundational result by Soudry et al. [53] showed that gradient descent optimization of logistic loss on linearly separable data converges in direction to the $L_2$ max-margin classifier at a rate $O(1/\log(t))$. Contemporaneous work by Ji and Telgarsky [26] generalized this by relaxing the data separability requirement. Ji et al. [29] later connected these findings to earlier work on regularization paths of logistic loss minimization [49], which enabled extensions to other loss functions (e.g., those with polynomial tail decay). More recently, Wu et al. [75] extends these results to the large step size regime with the same $O(1/\log(t))$ rate. The relatively slow convergence rate to

the max-margin classifier motivated investigation into adaptive step-sizes. Nacson et al. [43] showed that NGD with decaying step-size $\eta_t = 1/\sqrt{t}$ achieves $L_2$-margin convergence at rate $O(1/\sqrt{t})$. This rate was improved to $O(1/t)$ by Ji and Telgarsky [28] using constant step-sizes, and further to $O(1/t^2)$ through a specific momentum formulation [30]. Besides linear classifications, implicit bias of GD has been studied for least squares [21, 22, 5], homogeneous [40, 27, 74] and non-homogeneous neural networks [12], as well as matrix factorization [21]; see Vardi [64] for a survey.

All the above mentioned works focus almost exclusively on binary classification. The noticeable gap in analysis of multiclass classification in most existing literature is highlighted by Thrampoulidis et al. [61], and more recently emphasized by Ravi et al. [48], who extended the implicit bias result of Soudry et al. [53] to multiclass classification for losses with exponential tails, including CE, multiclass exponential, and PairLogLoss. Their approach leverages a framework of Wang and Scott [73] that allows multiclass losses and separability conditions to be written in margin-based forms similar to binary cases. However, these works only focus on GD with the $L_2$-geometry. In this work, we consider a wide range of algorithms with different geometries for multiclass classification.

Beyond GD, Gunasekar et al. [22] and Nacson et al. [43] showed that steepest descent w.r.t. entry-wise p-norms yields updates that in the limit maximize the margin w.r.t the same norm. Sun et al. [54, 55] showed that the iterates of mirror descent with the potential function chosen as the p-th power of the p-norm converge to the classifier that maximizes the margin w.r.t. the p-norm. In both cases, the convergence rate is slow at $O(1/\log(t))$. Wang et al. [70] further improved the rates for both steepest descent and mirror descent when $p \in (1, 2]$. Note that all these results apply only to the exponential loss. More recently, Tsilivis et al. [63] showed that the iterates of steepest descent algorithms converge to a KKT point of a generalized margin maximization problem in homogeneous neural networks. Moreover, the implicit bias of Adam (with or without the stability constant) has been studied in both linear and non-linear settings. Wang et al. [68] demonstrated the normalized iterates of Adam (with non-negligible stability constant) converge to a KKT point of a $L_2$-margin maximization problem for homogeneous neural networks. Zhang et al. [77] studied the implicit bias of Adam without the stability constant on (linearly) binary separable data. They showed that unlike GD, the Adam's iterates converge to a solution that maximizes the margin w.r.t the $L_\infty$-norm. The study of excluding the stability constant is also the focus of another recent work on the implicit bias of AdamW [76], where the authors again establish that convergence aligns with the $L_\infty$ geometry.

## 8 Conclusion

We have characterized the margin convergence rates of Spectral-GD and Muon for multiclass linear separable data. Given they are special cases of NSD and NMD w.r.t the spectral norm, the analysis is done on a wider scale by studying NSD/NMD w.r.t any entry-wise or Schatten p-norms. Thus, the rates also hold for optimizers of other geometries, such as the sign-descent (max-norm) or gradient-descent (2-norm) family. We further extend the analysis to Adam using the same framework. Future directions include removing the factor-$d$ from the bound of NMD, obtaining a tighter convergence rate for Adam, and studying other related algorithms such as Shampoo that involves non-diagonal preconditioners. It is also important to extend our results to (multiclass) non-separable settings [60] and nonlinear models such as diagonal neural nets [45], self-attention mechanisms [58, 4, 57, 66, 32] and homogeneous neural nets [40, 63, 12], helping further bridge the gap to deep learning practices. Finally, from a complementary statistical perspective, future work could seek identifying specific scenarios where margin maximization with respect to norms other than Frobenius leads to better generalization (extending a long line of prior works, e.g., [50, 16, 34, 42, 71, 18, 56, 65, 2, 62]).

## Acknowledgement

This work is funded partially by NSERC Discovery Grants RGPIN-2021-03677 and RGPIN-2022-03669, Alliance GrantALLRP 581098-22, a CIFAR AI Catalyst grant, and the Canada CIFAR AI Chair Program.

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

# Appendix

## A  Additional Experiments

We present additional experiments on Signum and NMD-GD in this section. Based on Figure 4, their margin convergence properties are very similar to those of SignGD and NGD respectively (see Sec. in the main text for the experimental setup).

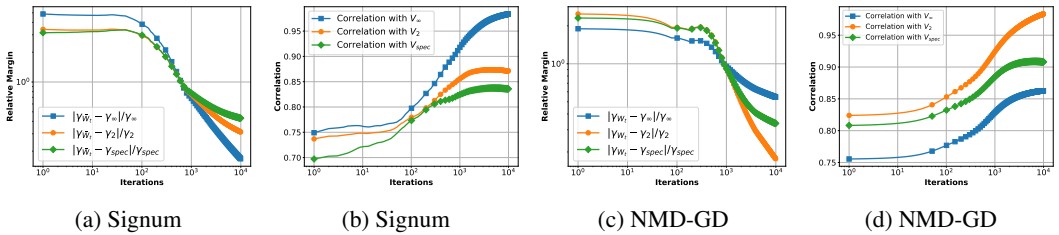

| (a) Signum | (b) Signum | (c) NMD-GD | (d) NMD-GD |

Figure 4: Implicit bias of Signum and NMD-GD on multiclass separable data. **(a)** Relative margin gap of Signum's iterates against iterations. **(b)** Correlation of Signum's iterates to $V_\infty$, $V_2$, and $V_{\mathrm{spec}}$ against iterations. See Figure 1 and 2 for the definitions of relative margin and correlation. **(c)** and **(d)** Same as (a) and (b) with Signum replaced by NMD-GD.

## B  Facts about CE loss and Softmax

Lemma 9 is on the gradient of the cross-entropy loss. It will be used for showing the form of $\mathcal{G}(\boldsymbol{W})$ in (8) lower bounds $\|\nabla\mathcal{L}(\boldsymbol{W})\|$ in Lemma 16.

**Lemma 9** (Gradient). *Let CE loss*

$$\mathcal{L}(\boldsymbol{W}) := -\frac{1}{n}\sum_{i\in[n]}\log\big(\mathbb{S}_{y_i}(\boldsymbol{W}\boldsymbol{h}_i)\big).$$

*For any $\boldsymbol{W}$, it holds*

- $\nabla\mathcal{L}(\boldsymbol{W}) = -\frac{1}{n}\sum_{i\in[n]}\left(\boldsymbol{e}_{y_i} - \boldsymbol{s}_i\right)\boldsymbol{h}_i^\top = -\frac{1}{n}(\boldsymbol{Y}-\boldsymbol{S})\boldsymbol{H}^\top$
- $\mathbb{1}_k^\top\nabla\mathcal{L}(\boldsymbol{W}) = 0$

- *For any matrix $\boldsymbol{A} \in \mathbb{R}^{k \times d}$,*

$$\langle \boldsymbol{A}, -\nabla \mathcal{L}(\boldsymbol{W}) \rangle = \frac{1}{n} \sum_i (1 - s_{iy_i}) \left( \boldsymbol{e}_{y_i}^\top \boldsymbol{A} \boldsymbol{h}_i - \frac{\sum_{c \neq y_i} s_{ic} \boldsymbol{e}_c^\top \boldsymbol{A} \boldsymbol{h}_i}{(1 - s_{iy_i})} \right)$$

$$= \frac{1}{n} \sum_{i \in [n]} \sum_{c \neq y_i} s_{ic} (\boldsymbol{e}_{y_i} - \boldsymbol{e}_c)^\top \boldsymbol{A} \boldsymbol{h}_i \tag{11}$$

*where we simplify $\boldsymbol{S} := \mathbb{S}(\boldsymbol{W}\boldsymbol{H}) = [\boldsymbol{s}_1, \ldots, \boldsymbol{s}_n] \in \mathbb{R}^{k \times n}$. The last statement yields*

$$\langle \boldsymbol{A}, -\nabla \mathcal{L}(\boldsymbol{W}) \rangle \geq \frac{1}{n} \sum_{i \in [n]} (1 - s_{iy_i}) \cdot \min_{c \neq y_i} (\boldsymbol{e}_{y_i} - \boldsymbol{e}_c)^\top \boldsymbol{A} \boldsymbol{h}_i. \tag{12}$$

*Proof.* First bullet is by direct calculation. Second bullet uses the fact that $\mathbb{1}^\top (\boldsymbol{y}_i - \boldsymbol{s}_i) = 1 - 1 = 0$ since $\mathbb{1}^\top \boldsymbol{s}_i = 1$. The third bullet follows by direct calculation and writing $\boldsymbol{s}_i^\top \boldsymbol{A} \boldsymbol{h}_i = (\sum_c s_{ic} \boldsymbol{e}_c)^\top \boldsymbol{A} \boldsymbol{h}_i = \sum_c s_{ic} \boldsymbol{e}_c^\top \boldsymbol{A} \boldsymbol{h}_i$. $\qquad\square$

Lemma 10 is on the Taylor expansion of the loss. It will be used in showing the descent properties of NSD and NMD.

**Lemma 10** (Hessian). *Let perturbation $\boldsymbol{\Delta} \in \mathbb{R}^{k \times d}$ and denote $\boldsymbol{W}' = \boldsymbol{W} + \boldsymbol{\Delta}$. Then,*

$$\mathcal{L}(\boldsymbol{W}') = \mathcal{L}(\boldsymbol{W}) - \frac{1}{n} \sum_{i \in [n]} \langle (\boldsymbol{e}_{y_i} - \mathbb{S}(\boldsymbol{W}\boldsymbol{h}_i)) \boldsymbol{h}_i^\top, \boldsymbol{\Delta} \rangle$$

$$+ \frac{1}{2n} \sum_{i \in [n]} \boldsymbol{h}_i^\top \boldsymbol{\Delta}^\top \left( \mathrm{diag}(\mathbb{S}(\boldsymbol{W}\boldsymbol{h}_i)) - \mathbb{S}(\boldsymbol{W}\boldsymbol{h}_i)\mathbb{S}(\boldsymbol{W}\boldsymbol{h}_i)^\top \right) \boldsymbol{\Delta} \boldsymbol{h}_i + o(\|\boldsymbol{\Delta}\|^3). \tag{13}$$

*Proof.* Define function $\ell_y : \mathbb{R}^k \to \mathbb{R}$ parameterized by $y \in [k]$ as follows:

$$\ell_y(\boldsymbol{l}) := -\log(\mathbb{S}_y(\boldsymbol{l})).$$

From Lemma 9,

$$\nabla \ell_y(\boldsymbol{l}) = -(\boldsymbol{e}_y - \mathbb{S}(\boldsymbol{l})).$$

Thus,

$$\nabla^2 \ell_y(\boldsymbol{l}) = \nabla \mathbb{S}(\boldsymbol{l}) = \mathrm{diag}(\mathbb{S}(\boldsymbol{l})) - \mathbb{S}(\boldsymbol{l})\mathbb{S}(\boldsymbol{l})^\top$$

Combining these the second-order taylor expansion of $\ell_y$ writes as follows for any $\boldsymbol{l}, \boldsymbol{\delta} \in \mathbb{R}^k$:

$$\ell_y(\boldsymbol{l} + \boldsymbol{\delta}) = \ell_y(\boldsymbol{l}) - (\boldsymbol{e}_y - \mathbb{S}(\boldsymbol{l}))^\top \boldsymbol{\delta} + \frac{1}{2} \boldsymbol{\delta}^\top \left( \mathrm{diag}(\mathbb{S}(\boldsymbol{l})) - \mathbb{S}(\boldsymbol{l})\mathbb{S}(\boldsymbol{l})^\top \right) \boldsymbol{\delta} + o(\|\boldsymbol{\delta}\|^3).$$

To evaluate this with respect to a change on the classifier parameters, set $\boldsymbol{l} = \boldsymbol{W}\boldsymbol{h}$ and $\boldsymbol{\delta} = \boldsymbol{\Delta}\boldsymbol{h}$ for $\boldsymbol{\Delta} \in \mathbb{R}^{k \times d}$. Denoting $\boldsymbol{W}' = \boldsymbol{W} + \boldsymbol{\Delta}$, we then have

$$\ell_y(\boldsymbol{W}') = \ell_y(\boldsymbol{W}) - \langle (\boldsymbol{e}_y - \mathbb{S}(\boldsymbol{l}))\boldsymbol{h}^\top, \boldsymbol{\Delta} \rangle + \frac{1}{2} \boldsymbol{h}^\top \boldsymbol{\Delta}^\top \left( \mathrm{diag}(\mathbb{S}(\boldsymbol{l})) - \mathbb{S}(\boldsymbol{l})\mathbb{S}(\boldsymbol{l})^\top \right) \boldsymbol{\Delta}\boldsymbol{h} + o(\|\boldsymbol{\Delta}\|^3).$$

This shows the desired since $n\mathcal{L}(\boldsymbol{W}) := \sum_{i \in [n]} \ell_{y_i}(\boldsymbol{W}\boldsymbol{h}_i)$ and we can further obtain

$$\ell_y(\boldsymbol{W}') = \ell_y(\boldsymbol{W}) - \langle (\boldsymbol{e}_y - \mathbb{S}(\boldsymbol{l}))\boldsymbol{h}^\top, \boldsymbol{\Delta} \rangle + \frac{1}{2} \boldsymbol{h}^\top \boldsymbol{\Delta}^\top \left( \mathrm{diag}(\mathbb{S}(\boldsymbol{l}')) - \mathbb{S}(\boldsymbol{l}')\mathbb{S}(\boldsymbol{l}')^\top \right) \boldsymbol{\Delta}\boldsymbol{h}, \tag{14}$$

where $\boldsymbol{l}' = \boldsymbol{l} + \zeta\boldsymbol{\delta}$ for some $\zeta \in [0, 1]$. $\qquad\square$

We prove the relationships in (7), which are useful for unifying the analysis of entry-wise and Schatten norms.

**Lemma 11.** *For any matrix $\boldsymbol{A} \in \mathbb{R}^{m \times n}$ and any entry-wise or Schatten $p$-norm $\|\cdot\|$ with $p \geq 1$, it holds that*

$$\|\boldsymbol{A}\|_{\max} \leq \|\boldsymbol{A}\| \leq \|\boldsymbol{A}\|_{\mathrm{sum}}.$$

*Proof.* The entry-wise p-norm case is trivial. Here, we focus the Schatten p-norm case. Note that $|||\boldsymbol{A}|||_2$ coincides with the entrywise 2-norm $\|\boldsymbol{A}\|_2$, but in general Schatten norms are different from entry-wise norms. On the other hand, Schatten norms preserve the ordering of norms. Specifically, por any $p \geq 1$, it holds:

$$|||\boldsymbol{A}|||_\infty = \sigma_1 \leq |||\boldsymbol{A}|||_p = \left(\sum_{i=1}^r \sigma_i^p\right)^{1/p} \leq \sum_{i=1}^r \sigma_i = |||\boldsymbol{A}|||_1 \,. \tag{15}$$

It is also well-known that

$$|||\boldsymbol{A}|||_\infty = \max_{\|\boldsymbol{u}\|_2=\|\boldsymbol{v}\|_2=1} \boldsymbol{u}^\top \boldsymbol{A}\boldsymbol{v} \geq \max_{i,j} |\boldsymbol{A}[i,j]| = \|\boldsymbol{A}\|_{\max} \tag{16}$$

where the inequality follows by selecting $\boldsymbol{u} = \texttt{sign}(\boldsymbol{A}[i',j']) \cdot \boldsymbol{e}_{i'}$ and $\boldsymbol{v} = \boldsymbol{e}_{j'}$ for $(i',j')$ such that $|\boldsymbol{A}[i',j']| = \|\boldsymbol{A}\|_{\max}$ and $\boldsymbol{e}_{i'}, \boldsymbol{e}_{j'}$ corresponding basis vectors.

Using this together with duality, it also holds that

$$|||\boldsymbol{A}|||_1 \leq \|\boldsymbol{A}\|_{\text{sum}} \,. \tag{17}$$

This follows from the following sequnece of inequalities

$$|||\boldsymbol{A}|||_1 = \max_{|||\boldsymbol{B}|||_\infty \leq 1} \langle \boldsymbol{A}, \boldsymbol{B} \rangle \leq \|\boldsymbol{A}\|_{\text{sum}} \cdot \max_{|||\boldsymbol{B}|||_\infty \leq 1} \|\boldsymbol{B}\|_{\max} \leq \|\boldsymbol{A}\|_{\text{sum}} \cdot \max_{|||\boldsymbol{B}|||_\infty \leq 1} |||\boldsymbol{B}|||_\infty \leq \|\boldsymbol{A}\|_{\text{sum}},$$
$$\tag{18}$$

where the first inequality follows from generalized Cauchy-Scwhartz and the second inequality by (16). $\square$

Lemma 12 is used in bounding the second order term in the Taylor expansion of $\mathcal{L}(\boldsymbol{W})$.

**Lemma 12.** *For any entry-wise or Schatten p-norm $\|\cdot\|$ with $p \geq 1$, any $\boldsymbol{s} \in \Delta^{k-1}$ in the k-dimensional simplex, any index $c \in [k]$, and $\boldsymbol{v} \in \mathbb{R}^k$, it holds that*

$$\boldsymbol{v}^\top \left(\text{diag}(\boldsymbol{s}) - \boldsymbol{s}\boldsymbol{s}^\top\right) \boldsymbol{v} \leq 4(1-s_c)\|\boldsymbol{v}\boldsymbol{v}^T\|.$$

*Proof.* See main text. $\square$

Lemma 13 is used in the proof of Lemma 12.

**Lemma 13.** *For any $\boldsymbol{s} \in \Delta^{k-1}$ in the k-dimensional simplex and any index $c \in [k]$ it holds that*

$$\sum_{c'} s_{c'}(1-s_{c'}) \leq 2(1-s_c)\,.$$

*Proof.* With a bit of algebra and using $\sum_{c' \neq c} s_{c'} = 1 - s_c$ the claim becomes equivalent to

$$\sum_{c' \neq c} s_{c'}^2 + s_c^2 - 2s_c + 1 \geq 0.$$

Since this holds true, the lemma holds. $\square$

**Lemma 14.** *For any $\boldsymbol{s} \in \Delta^{k-1}$ in the k-dimensional simplex, any index $c \in [k]$, any $\boldsymbol{\Delta} \in \mathbb{R}^{k \times d}$, and any $\boldsymbol{h} \in \mathbb{R}^d$, it holds:*

$$\boldsymbol{h}^\top \boldsymbol{\Delta}^\top \left(\text{diag}(\boldsymbol{s}) - \boldsymbol{s}\boldsymbol{s}^\top\right) \boldsymbol{\Delta}\boldsymbol{h} \leq 4B^2 \|\boldsymbol{\Delta}\|^2 (1-s_c)\,.$$

*Proof.* We let $\boldsymbol{v} := \boldsymbol{\Delta}\boldsymbol{h}$. For any Schatten p-norm, we have

$$|||\boldsymbol{v}\boldsymbol{v}^\top||| = \|\boldsymbol{v}\|_2^2 \leq |||\boldsymbol{\Delta}|||_\infty^2 \|\boldsymbol{h}\|_2^2 \leq |||\boldsymbol{\Delta}|||^2 \|\boldsymbol{h}\|_2^2 \leq B^2 |||\boldsymbol{\Delta}|||^2 \,.$$

For any entry-wise p-norm, we have

$$\|\boldsymbol{\Delta}\boldsymbol{h}\|^p = \|\boldsymbol{\Delta}\boldsymbol{h}\|^p = \sum_j |\boldsymbol{e}_j^\top \boldsymbol{\Delta}\boldsymbol{h}|^p \leq \sum_j \|\boldsymbol{e}_j^\top \boldsymbol{\Delta}\|_p^p \|\boldsymbol{h}\|^p = \|\boldsymbol{h}\|_*^p \sum_{ij} |\boldsymbol{\Delta}[i,j]|^p = \|\boldsymbol{h}\|_*^p \|\boldsymbol{\Delta}\|^p \,.$$

This implies

$$\|\boldsymbol{v}\boldsymbol{v}^T\| = \|\boldsymbol{v}\|^2 = \|\boldsymbol{\Delta}\boldsymbol{h}\|^2 \leq \|\boldsymbol{\Delta}\|^2 \|\boldsymbol{h}\|_*^2 \leq B^2 \|\boldsymbol{\Delta}\|^2.$$

Combine these results and apply Lemma 12, we obtain the desired. $\square$

The following lemma summarizes the properties of the softmax map that will be used in the proof of Lemma 25 and 35.

**Lemma 15.** *For any $v, v', q, q' \in \mathbb{R}^k$ and $c \in [k]$, the following inequalities hold:*

(i) $|\frac{\mathbb{S}_c(v')}{\mathbb{S}_c(v)} - 1| \leq e^{2\|v - v'\|_\infty} - 1$

(ii) $|\frac{1 - \mathbb{S}_c(v')}{1 - \mathbb{S}_c(v)} - 1| \leq e^{2\|v - v'\|_\infty} - 1$

(iii) $|\frac{\mathbb{S}_c(v')\mathbb{S}_c(q')}{\mathbb{S}_c(v)\mathbb{S}_c(q)} - 1| \leq e^{2(\|v' - v\|_\infty + \|q' - q\|_\infty)} - 1$

(iv) $|\frac{\mathbb{S}_c(v')(1 - \mathbb{S}_c(q'))}{\mathbb{S}_c(v)(1 - \mathbb{S}_c(q))} - 1| \leq e^{2(\|v' - v\|_\infty + \|q' - q\|_\infty)} - 1$

(v) $|\frac{(1 - \mathbb{S}_c(v'))(1 - \mathbb{S}_c(q'))}{(1 - \mathbb{S}_c(v))(1 - \mathbb{S}_c(q))} - 1| \leq e^{2(\|v' - v\|_\infty + \|q' - q\|_\infty)} - 1$

*Proof.* We prove each inequality:

(i) First, observe that

$$|\frac{\mathbb{S}_c(v')}{\mathbb{S}_c(v)} - 1| = |\frac{e^{v'_c} \sum_{i \in [k]} e^{v_i}}{e^{v_c} \sum_{i \in [k]} e^{v'_i}} - 1|$$

$$= |\frac{\sum_{i \in [k]} e^{v'_c + v_i} - \sum_{i \in [k]} e^{v_c + v'_i}}{\sum_{i \in [k]} e^{v_c + v'_i}}|$$

$$\leq \frac{\sum_{i \in [k]} |e^{v'_c + v_i} - e^{v_c + v'_i}|}{\sum_{i \in [k]} e^{v_c + v'_i}}$$

For any $i \in [k]$, we have $\frac{|e^{v'_c + v_i} - e^{v_c + v'_i}|}{e^{v_c + v'_i}} = |e^{v'_c - v_c + v_i - v'_i} - 1| \leq e^{|v'_c - v_c + v_i - v'_i|} - 1 \leq e^{2\|v - v'\|_\infty} - 1$.
This implies $\sum_{i \in [k]} |e^{v'_c + v_i} - e^{v_c + v'_i}| \leq (e^{2\|v - v'\|_\infty} - 1) \sum_{i \in [k]} e^{v_c + v'_i}$, from which we obtain the desired inequality.

(ii) For the second inequality:

$$|\frac{1 - \mathbb{S}_c(v')}{1 - \mathbb{S}_c(v)} - 1| = |\frac{1 - \frac{e^{v'_c}}{\sum_{i \in [k]} e^{v'_i}}}{1 - \frac{e^{v_c}}{\sum_{i \in [k]} e^{v_i}}} - 1|$$

$$= |\frac{(\sum_{j \in [k], j \neq c} e^{v'_j})(\sum_{i \in [k]} e^{v_i})}{(\sum_{j \in [k], j \neq c} e^{v_j})(\sum_{i \in [k]} e^{v'_i})} - 1|$$

$$= |\frac{\sum_{j \in [k], j \neq c} \sum_{i \in [k]} [e^{v'_j + v_i} - e^{v_j + v'_i}]}{\sum_{j \in [k], j \neq c} \sum_{i \in [k]} e^{v_j + v'_i}}|$$

$$\leq \frac{\sum_{j \in [k], j \neq c} \sum_{i \in [k]} |e^{v'_j + v_i} - e^{v_j + v'_i}|}{\sum_{j \in [k], j \neq c} \sum_{i \in [k]} e^{v_j + v'_i}}$$

For any $j \in [k]$, $j \neq c$, and $i \in [k]$, we have $\frac{|e^{v'_j + v_i} - e^{v_j + v'_i}|}{e^{v_j + v'_i}} \leq e^{2\|v - v'\|_\infty} - 1$. This implies that $\sum_{j \in [k], j \neq c} \sum_{i \in [k]} |e^{v'_j + v_i} - e^{v_j + v'_i}| \leq (e^{2\|v - v'\|_\infty} - 1) \sum_{j \in [k], j \neq c} \sum_{i \in [k]} e^{v_j + v'_i}$, from which the result follows.

(iii) For the third inequality:

$$|\frac{\mathbb{S}_c(\boldsymbol{v}')\mathbb{S}_c(\boldsymbol{q}')}{\mathbb{S}_c(\boldsymbol{v})\mathbb{S}_c(\boldsymbol{q})} - 1| = |\frac{\frac{e^{v'_c}}{\sum_{i\in[k]} e^{v'_i}}\frac{e^{q'_c}}{\sum_{i\in[k]} e^{q'_i}}}{\frac{e^{v_c}}{\sum_{i\in[k]} e^{v_i}}\frac{e^{q_c}}{\sum_{i\in[k]} e^{q_i}}} - 1|$$

$$= |\frac{\frac{e^{v'_c}}{\sum_{i\in[k]} e^{v'_i}}\frac{e^{q'_c}}{\sum_{i\in[k]} e^{q'_i}}}{\frac{e^{v_c}}{\sum_{i\in[k]} e^{v_i}}\frac{e^{q_c}}{\sum_{i\in[k]} e^{q'_i}}} - \frac{\frac{e^{v_c}}{\sum_{i\in[k]} e^{v'_i}}\frac{e^{q_c}}{\sum_{i\in[k]} e^{q'_i}}}{\frac{e^{v_c}}{\sum_{i\in[k]} e^{v'_i}}\frac{e^{q_c}}{\sum_{i\in[k]} e^{q'_i}}}|$$

$$= |\frac{e^{v'_c}e^{q'_c}\sum_{i\in[k]} e^{v_i}\sum_{j\in[k]} e^{q_j}}{e^{v_c}e^{q_c}\sum_{i\in[k]} e^{v'_i}\sum_{j\in[k]} e^{q'_j}} - \frac{e^{v_c}e^{q_c}\sum_{i\in[k]} e^{v'_i}\sum_{j\in[k]} e^{q'_j}}{e^{v_c}e^{q_c}\sum_{i\in[k]} e^{v'_i}\sum_{j\in[k]} e^{q'_j}}|$$

$$= |\frac{\sum_{i\in[k]}\sum_{j\in[k]}\left[e^{v'_c+v_i+q'_c+q_j} - e^{v_c+v'_i+q_c+q'_j}\right]}{\sum_{i\in[k]}\sum_{j\in[k]} e^{v_c+v'_i+q_c+q'_j}}|$$

$$\leq \frac{\sum_{i\in[k]}\sum_{j\in[k]} |e^{v'_c+v_i+q'_c+q_j} - e^{v_c+v'_i+q_c+q'_j}|}{\sum_{i\in[k]}\sum_{j\in[k]} e^{v_c+v'_i+q_c+q'_j}}$$

For any $i \in [k]$ and $j \in [k]$, $\frac{|e^{v'_c+v_i+q'_c+q_j}-e^{v_c+v'_i+q_c+q'_j}|}{e^{v_c+v'_i+q_c+q'_j}} = |e^{v'_c-v_c+v_i-v'_i+q'_c-q_c+q_j-q'_j} - 1| \leq e^{|v'_c-v_c|+|v_i-v'_i|+|q'_c-q_c|+|q_j-q'_j|} - 1 \leq e^{2(\|\boldsymbol{v}'-\boldsymbol{v}\|_\infty+\|\boldsymbol{q}'-\boldsymbol{q}\|_\infty)} - 1$. Then, rearranging and summing over $i$ and $j$ leads to the result.

(iv) For the fourth inequality:

$$|\frac{\mathbb{S}_c(\boldsymbol{v}')(1 - \mathbb{S}_c(\boldsymbol{q}'))}{\mathbb{S}_c(\boldsymbol{v})(1 - \mathbb{S}_c(\boldsymbol{q}))} - 1| = |\frac{\frac{e^{v'_c}}{\sum_{s\in[k]} e^{v'_s}}(1 - \frac{e^{q'_c}}{\sum_{t\in[k]} e^{q'_t}})}{\frac{e^{v_c}}{\sum_{s\in[k]} e^{v_s}}(1 - \frac{e^{q_c}}{\sum_{t\in[k]} e^{q_t}})} - 1|$$

$$= |\frac{\frac{e^{v'_c}}{\sum_{s\in[k]} e^{v'_s}}\frac{\sum_{i\in[k],i\neq c} e^{q'_i}}{\sum_{t\in[k]} e^{q'_t}}}{\frac{e^{v_c}}{\sum_{s\in[k]} e^{v_s}}\frac{\sum_{i\in[k],i\neq c} e^{q_t}}{\sum_{t\in[k]} e^{q_t}}} - 1|$$

$$= |\frac{\sum_{i\in[k],i\neq c}\sum_{t\in[k]}\sum_{s\in[k]} e^{v'_c+q'_i+v_s+q_t}}{\sum_{i\in[k],i\neq c}\sum_{t\in[k]}\sum_{s\in[k]} e^{v_c+q_i+v'_s+q'_t}} - 1|$$

$$\leq \frac{\sum_{i\in[k],i\neq c}\sum_{t\in[k]}\sum_{s\in[k]} |e^{v'_c+q'_i+v_s+q_t} - e^{v_c+q_i+v'_s+q'_t}|}{\sum_{i\in[k],i\neq c}\sum_{t\in[k]}\sum_{s\in[k]} e^{v_c+q_i+v'_s+q'_t}}$$

For each $i \in [k], i \neq c$, $s \in [k]$, and $t \in [k]$, we obtain $\frac{|e^{v'_c+q'_i+v_s+q_t}-e^{v_c+q_i+v'_s+q'_t}|}{e^{v_c+q_i+v'_s+q'_t}} \leq e^{2(\|\boldsymbol{v}'-\boldsymbol{v}\|_\infty+\|\boldsymbol{q}'-\boldsymbol{q}\|_\infty)} - 1$. Then, rearranging and summing over $i$, $s$, and $t$ leads to the result.

(v) Finally, for the fifth inequality:

$$|\frac{(1 - \mathbb{S}_c(\boldsymbol{v}'))(1 - \mathbb{S}_c(\boldsymbol{q}'))}{(1 - \mathbb{S}_c(\boldsymbol{v}))(1 - \mathbb{S}_c(\boldsymbol{q}))} - 1| = |\frac{(1 - \frac{e^{v'_c}}{\sum_{s\in[k]} e^{v'_s}})(1 - \frac{e^{q'_c}}{\sum_{t\in[k]} e^{q'_t}})}{(1 - \frac{e^{v_c}}{\sum_{s\in[k]} e^{v_s}})(1 - \frac{e^{q_c}}{\sum_{t\in[k]} e^{q_t}})} - 1|$$

$$= |\frac{\frac{\sum_{j\in[k],j\neq c} e^{v'_j}}{\sum_{s\in[k]} e^{v'_s}}\frac{\sum_{i\in[k],i\neq c} e^{q'_i}}{\sum_{t\in[k]} e^{q'_t}}}{\frac{\sum_{j\in[k],j\neq c} e^{v_j}}{\sum_{s\in[k]} e^{v_s}}\frac{\sum_{i\in[k],i\neq c} e^{q_i}}{\sum_{t\in[k]} e^{q_t}}} - 1|$$

$$= |\frac{\sum_{j\in[k],j\neq c}\sum_{i\in[k],i\neq c}\sum_{t\in[k]}\sum_{s\in[k]} e^{v'_j+q'_i+v_s+q_t}}{\sum_{j\in[k],j\neq c}\sum_{i\in[k],i\neq c}\sum_{t\in[k]}\sum_{s\in[k]} e^{v_j+q_i+v'_s+q'_t}} - 1|$$

$$\leq \frac{\sum_{j\in[k],j\neq c}\sum_{i\in[k],i\neq c}\sum_{t\in[k]}\sum_{s\in[k]} |e^{v'_j+q'_i+v_s+q_t} - e^{v_j+q_i+v'_s+q'_t}|}{\sum_{j\in[k],j\neq c}\sum_{i\in[k],i\neq c}\sum_{t\in[k]}\sum_{s\in[k]} e^{v_j+q_i+v'_s+q'_t}}.$$

For each $j \in [k]$ $(j \neq c)$, $i \in [k]$ $(i \neq c)$, $s \in [k]$, and $t \in [k]$, we have

$$
\frac{|e^{v'_j + q'_i + v_s + q_t} - e^{v_j + q_i + v'_s + q'_t}|}{e^{v_j + q_i + v'_s + q'_t}} = |e^{v'_j - v_j + q'_i - q_i + v_s - v'_s + q_t - q'_t} - 1|
$$
$$
\leq e^{|v'_j - v_j| + |q'_i - q_i| + |v_s - v'_s| + |q_t - q'_t|} - 1
$$
$$
\leq e^{2(\|\boldsymbol{v}' - \boldsymbol{v}\|_\infty + \|\boldsymbol{q}' - \boldsymbol{q}\|_\infty)} - 1
$$

Then, rearranging and summing over $j$, $i$, $s$, and $t$ leads to the result. $\qquad \square$

## C  Lemmas on Loss and Proxy Function

Lemma 16 shows that $\mathcal{G}(\boldsymbol{W})$ upper and lower bound the dual norm of the loss gradient.

**Lemma 16** ($\mathcal{G}(\boldsymbol{W})$ as proxy to the loss-gradient norm). *Under Assumption 4. For any $\boldsymbol{W} \in \mathbb{R}^{k \times d}$, it holds that*
$$
2B \cdot \mathcal{G}(\boldsymbol{W}) \geq \|\nabla \mathcal{L}(\boldsymbol{W})\|_* \geq \gamma \cdot \mathcal{G}(\boldsymbol{W}).
$$

*Proof.* First, we prove the lower bound. By duality and direct application of (12)

$$
\|\nabla \mathcal{L}(\boldsymbol{W})\|_* = \max_{\|\boldsymbol{A}\| \leq 1} \langle \boldsymbol{A}, -\nabla \mathcal{L}(\boldsymbol{W}) \rangle
$$
$$
\geq \max_{\|\boldsymbol{A}\| \leq 1} \frac{1}{n} \sum_{i \in [n]} (1 - s_{iy_i}) \min_{c \neq y_i} (\boldsymbol{e}_{y_i} - \boldsymbol{e}_c)^T \boldsymbol{A} \boldsymbol{h}_i
$$
$$
\geq \frac{1}{n} \sum_{i \in [n]} (1 - s_{iy_i}) \cdot \max_{\|\boldsymbol{A}\| \leq 1} \min_{i \in [n], c \neq y_i} (\boldsymbol{e}_{y_i} - \boldsymbol{e}_c)^T \boldsymbol{A} \boldsymbol{h}_i.
$$

Second, for the upper bound, it holds by triangle inequality and relationships (7) that

$$
\|\nabla \mathcal{L}(\boldsymbol{W})\|_* \leq \|\nabla \mathcal{L}(\boldsymbol{W})\|_{\text{sum}} \leq \frac{1}{n} \sum_{i \in [n]} \|\nabla \ell_i(\boldsymbol{W})\|_{\text{sum}},
$$

where $\ell_i(\boldsymbol{W}) = -\log(\mathbb{S}_{y_i}(\boldsymbol{W} \boldsymbol{h}_i))$. Recall that

$$
\nabla \ell_i(\boldsymbol{W}) = -(\boldsymbol{e}_y - \mathbb{S}_{y_i}(\boldsymbol{W} \boldsymbol{h}_i)) \boldsymbol{h}_i^\top,
$$

and, for two vectors $\boldsymbol{v}, \boldsymbol{u}$: $\|\boldsymbol{u} \boldsymbol{v}^\top\|_{\text{sum}} = \|\boldsymbol{u}\|_1 \|\boldsymbol{v}\|_1$. Combining these and noting that

$$
\|\boldsymbol{e}_{y_i} - \mathbb{S}_{y_i}(\boldsymbol{W} \boldsymbol{h}_i)\|_1 = 2(1 - s_{y_i})
$$

together with using the assumption $\|\boldsymbol{h}_i\| \leq B$ yields the advertised upper bound. $\qquad \square$

Built upon Lemma 16, we obtain a simple bound on the loss difference at two points.

**Lemma 17.** *For any $\boldsymbol{W}, \boldsymbol{W}_0 \in \mathbb{R}^{k \times d}$, suppose that $\mathcal{L}(\boldsymbol{W})$ is convex, we have*
$$
|\mathcal{L}(\boldsymbol{W}) - \mathcal{L}(\boldsymbol{W}_0)| \leq 2B \|\boldsymbol{W} - \boldsymbol{W}_0\|.
$$

*Proof.* By convexity of $\mathcal{L}$, we have

$$
\mathcal{L}(\boldsymbol{W}_0) - \mathcal{L}(\boldsymbol{W}) \leq \langle \nabla \mathcal{L}(\boldsymbol{W}_0), \boldsymbol{W}_0 - \boldsymbol{W} \rangle \leq \|\nabla \mathcal{L}(\boldsymbol{W}_0)\|_* \|\boldsymbol{W}_0 - \boldsymbol{W}\| \leq 2B \|\boldsymbol{W}_0 - \boldsymbol{W}\|,
$$

where the last inequality is by Lemma 16. Similarly, we can also show that $\mathcal{L}(\boldsymbol{W}) - \mathcal{L}(\boldsymbol{W}_0) \leq 2B \|\boldsymbol{W}_0 - \boldsymbol{W}\|$. $\qquad \square$

Lemma 18 shows the close relationships between $\mathcal{G}(\boldsymbol{W})$ and $\mathcal{L}(\boldsymbol{W})$. The proxy $\mathcal{G}(\boldsymbol{W})$ not only lower bounds $\mathcal{L}(\boldsymbol{W})$, but also upper bounds $\mathcal{L}(\boldsymbol{W})$ up to a factor depending on $\mathcal{L}(\boldsymbol{W})$. Moreover, the rate of convergence $\frac{\mathcal{G}(\boldsymbol{W})}{\mathcal{L}(\boldsymbol{W})}$ depends on the rate of decrease in the loss.

**Lemma 18** ($\mathcal{G}(\boldsymbol{W})$ as proxy to the loss). *Let $\boldsymbol{W} \in \mathbb{R}^{k \times d}$, we have*

*(i)* $1 \geq \frac{\mathcal{G}(\boldsymbol{W})}{\mathcal{L}(\boldsymbol{W})} \geq 1 - \frac{n\mathcal{L}(\boldsymbol{W})}{2}$

*(ii)* Suppose that $\boldsymbol{W}$ satisfies $\mathcal{L}(\boldsymbol{W}) \leq \frac{\log 2}{n}$ or $\mathcal{G}(\boldsymbol{W}) \leq \frac{1}{2n}$, then $\mathcal{L}(\boldsymbol{W}) \leq 2\mathcal{G}(\boldsymbol{W})$.

*Proof.* (i) Denote for simplicity $s_i := s_{iy_i} = \mathbb{S}_{y_i}(\boldsymbol{W}\boldsymbol{h}_i)$, thus $\mathcal{L}(\boldsymbol{W}) = \frac{1}{n}\sum_{i\in[n]}\log(1/s_i)$ and $\mathcal{G}(\boldsymbol{W}) = \frac{1}{n}\sum_{i\in[n]}(1-s_i)$. For the upper bound, simply use the fact that $e^{x-1} \geq x$, for all $x \in [0,1]$, thus $\log(1/s_i) \geq 1 - s_i$ for all $i \in [n]$.

The lower bound can be proved using the exact same arguments in the proof of Zhang et al. [77, Lemma C.7] for the binary case. For completeness, we provide an alternative elementary proof. It suffices to prove for $n = 1$ that for $s \in (0,1)$:

$$1 - s \geq \log(1/s) - \frac{1}{2}\log^2(1/s). \tag{19}$$

The general case follows by summing over $s = s_i$ and using $\sum_{i\in[n]}\log^2(1/s_i) \leq \left(\sum_{i\in[n]}\log(1/s_i)\right)^2$ since $\log(1/s_i) > 0$. For (19), let $x = \log(1/s) > 0$. The inequality becomes $e^{-x} \leq 1 - x + x^2/2$, which holds for $x > 0$ by the second-order Taylor expansion of $e^{-x}$ around 0.

(ii) The sufficiency of $\mathcal{L}(\boldsymbol{W}) \leq \frac{\log 2}{n}$ (to guarantee that $\mathcal{L}(\boldsymbol{W}) \leq 2\mathcal{G}(\boldsymbol{W})$) follows from (i) and $\mathcal{L}(\boldsymbol{W}) \leq \frac{\log 2}{n} \leq \frac{1}{n}$. The inequality $\log(\frac{1}{x}) \leq 2(1-x)$ holds when $x \in [0.2032, 1]$. This translates to the following sufficient condition on $s_{iy_i}$

$$s_i = \frac{e^{\boldsymbol{\ell}_i[y_i]}}{\sum_{c\in[k]}e^{\boldsymbol{\ell}_i[c]}} = \frac{1}{1 + \sum_{c\in[k],c\neq y_i}e^{\boldsymbol{\ell}_i[c]-\boldsymbol{\ell}_i[y_i]}} \geq 0.2032.$$

Under the assumption $\mathcal{G}(\boldsymbol{W}) \leq \frac{1}{2n}$, we have $1 - s_i \leq \sum_{i\in[n]}(1-s_i) = n\mathcal{G}(\boldsymbol{W}) \leq \frac{1}{2}$, from which we obtain $s_i \geq \frac{1}{2} \geq 0.2032$ for all $i \in [n]$. $\qquad\square$

Lemma 19 shows that the data becomes separable when the loss is small. It is used in deriving the lower bound on the un-normalized margin.

**Lemma 19** (Low $\mathcal{L}(\boldsymbol{W})$ implies separability). *Suppose that there exists $\boldsymbol{W} \in \mathbb{R}^{k\times d}$ such that $\mathcal{L}(\boldsymbol{W}) \leq \frac{\log 2}{n}$, then we have*

$$(\boldsymbol{e}_{y_i} - \boldsymbol{e}_c)^T\boldsymbol{W}\boldsymbol{h}_i \geq 0, \quad \text{for all } i \in [n] \text{ and for all } c \in [k] \text{ such that } c \neq y_i. \tag{20}$$

*Proof.* We rewrite the loss into the form:

$$\mathcal{L}(\boldsymbol{W}) = -\frac{1}{n}\sum_{i\in[n]}\log\left(\frac{e^{\boldsymbol{\ell}_i[y_i]}}{\sum_{c\in[k]}e^{\boldsymbol{\ell}_i[c]}}\right) = \frac{1}{n}\sum_{i\in[n]}\log\left(1 + \sum_{c\neq y_i}e^{-(\boldsymbol{\ell}_i[y_i]-\boldsymbol{\ell}_i[c])}\right).$$

Fix any $i \in [n]$, by the assumption that $\mathcal{L}(\boldsymbol{W}) \leq \frac{\log 2}{n}$, we have the following:

$$\log\left(1 + \sum_{c\neq y_i}e^{-(\boldsymbol{\ell}_i[y_i]-\boldsymbol{\ell}_i[c])}\right) \leq n\mathcal{L}(\boldsymbol{W}) \leq \log(2).$$

This implies:

$$e^{-\min_{c\neq y_i}(\boldsymbol{\ell}_i[y_i]-\boldsymbol{\ell}_i[c])} = \max_{c\neq y_i}e^{-(\boldsymbol{\ell}_i[y_i]-\boldsymbol{\ell}_i[c])} \leq \sum_{c\neq y_i}e^{-(\boldsymbol{\ell}_i[y_i]-\boldsymbol{\ell}_i[c])} \leq 1.$$

After taking log on both sides, we obtain the following: $\boldsymbol{\ell}_i[y_i] - \boldsymbol{\ell}_i[c] = (\boldsymbol{e}_{y_i} - \boldsymbol{e}_c)^T\boldsymbol{W}\boldsymbol{h}_i \geq 0$ for any $c \in [k]$ such that $c \neq y_i$. $\qquad\square$

Lemma 20 shows that the ratio of $\mathcal{G}(\boldsymbol{W})$ at two points can be bounded by exponentiating the max-norm of their differences. It is used in handling the second order term in the Taylor expansion of the loss.

**Lemma 20** (Ratio of $\mathcal{G}(\boldsymbol{W})$). *For any $\psi \in [0,1]$, we have the following:*

$$\frac{\mathcal{G}(\boldsymbol{W} - \psi\eta\boldsymbol{\Delta})}{\mathcal{G}(\boldsymbol{W})} \leq e^{2B\psi\eta\|\boldsymbol{\Delta}\|_{\max}} \leq e^{2B\psi\eta\|\boldsymbol{\Delta}\|}.$$

*Proof.* Note that the second inequality is by relationships (7). Here, we only prove the first inequality. By the definition of $\mathcal{G}(\boldsymbol{W})$, we have:

$$\frac{\mathcal{G}(\boldsymbol{W} - \psi\eta\boldsymbol{\Delta})}{\mathcal{G}(\boldsymbol{W})} = \frac{\sum_{i\in[n]}\big(1 - \mathbb{S}_{y_i}((\boldsymbol{W} - \psi\eta\boldsymbol{\Delta})\boldsymbol{h}_i)\big)}{\sum_{i\in[n]}\big(1 - \mathbb{S}_{y_i}(\boldsymbol{W}\boldsymbol{h}_i)\big)}.$$

For any $c \in [k]$ and $\boldsymbol{v}, \boldsymbol{v}' \in \mathbb{R}^k$, we have:

$$\begin{aligned}
\frac{1 - \mathbb{S}_c(\boldsymbol{v}')}{1 - \mathbb{S}_c(\boldsymbol{v})} &= \frac{1 - \frac{e^{v'_c}}{\sum_{i\in[k]} e^{v'_i}}}{1 - \frac{e^{v_c}}{\sum_{i\in[k]} e^{v_i}}}\\
&= \frac{\frac{\sum_{j\in[k],j\neq c} e^{v'_j}}{\sum_{i\in[k]} e^{v'_i}}}{\frac{\sum_{j\in[k],j\neq c} e^{v_j}}{\sum_{i\in[k]} e^{v_i}}}\\
&= \frac{\sum_{j\in[k],j\neq c} \sum_{i\in[k]} e^{v'_j+v_i}}{\sum_{j\in[k],j\neq c} \sum_{i\in[k]} e^{v_j+v'_i}}\\
&\leq e^{2\|\boldsymbol{v}-\boldsymbol{v}'\|_\infty}.
\end{aligned}$$

The last inequality is because $\frac{e^{v'_j+v_i}}{e^{v_j+v'_i}} \leq e^{|v'_j-v_j|+|v_i-v'_i|} \leq e^{2\|\boldsymbol{v}-\boldsymbol{v}'\|_\infty}$, which implies that $\sum_{j\in[k],j\neq c} \sum_{i\in[k]} e^{v'_j+v_i} \leq e^{2\|\boldsymbol{v}-\boldsymbol{v}'\|_\infty} \sum_{j\in[k],j\neq c} \sum_{i\in[k]} e^{v_j+v'_i}$. Next, we specialize this result to $\boldsymbol{v}' = (\boldsymbol{W} - \psi\eta\boldsymbol{\Delta})\boldsymbol{h}_i, \boldsymbol{v} = \boldsymbol{W}\boldsymbol{h}_i$, and $c = y_i$ for any $i \in [n]$ to obtain:

$$\frac{1 - \mathbb{S}_{y_i}((\boldsymbol{W} - \psi\eta\boldsymbol{\Delta})\boldsymbol{h}_i)}{1 - \mathbb{S}_{y_i}(\boldsymbol{W}\boldsymbol{h}_i)} \leq e^{2\eta\psi\|\boldsymbol{\Delta}\boldsymbol{h}_i\|_\infty} \leq e^{2B\eta\psi\|\boldsymbol{\Delta}\|_{\max}}.$$

Then, we rearrange and sum over $i \in [n]$ to obtain: $\sum_{i\in[n]}\big(1 - \mathbb{S}_{y_i}((\boldsymbol{W} - \psi\eta\boldsymbol{\Delta})\boldsymbol{h}_i)\big) \leq e^{2B\eta\psi\|\boldsymbol{\Delta}\|_{\max}} \sum_{i\in[n]}\big(1 - \mathbb{S}_{y_i}(\boldsymbol{W}\boldsymbol{h}_i)\big)$, from which the desired inequality follows. The second inequality in the lemma statement follows from the relationship (7). $\qquad\square$

## D   Implicit Bias of Normalized Steepest Descent

**Proof Overview**   We consider a decay learning rate schedule of the form $\eta_t = \Theta(\frac{1}{t^a})$ where $a \in (0,1]$. The first step is to show that the **loss monotonically decreases** after certain time and the rate depends on $\mathcal{G}(\boldsymbol{W})$. To obtain this, we apply Lemma 16 and Lemma 12 to upper bound the first-order and second-order terms in the Taylor expansion of the loss (21), respectively. Next, we use the decrease in loss to derive a lower bound on the unnormalized margin which involves the ratio $\frac{\mathcal{G}(\boldsymbol{W})}{\mathcal{L}(\boldsymbol{W})}$. A crucial step involved is to find a time $\bar{t}_2$ such that separability (32) holds for all $t \geq \bar{t}_2$, and the existence of $\bar{t}_2$ is guaranteed by loss monotonicity such that the condition $\mathcal{L}(\boldsymbol{W}_t) \leq \frac{\log 2}{n}$ will be satisfied for sufficiently large t's.

Then, we argue that the ratio $\frac{\mathcal{G}(\boldsymbol{W}_t)}{\mathcal{L}(\boldsymbol{W}_t)}$ converges to 1 exponentially fast (recalling that $1 \geq \frac{\mathcal{G}(\boldsymbol{W}_t)}{\mathcal{L}(\boldsymbol{W}_t)} \geq 1 - \frac{n\mathcal{L}(\boldsymbol{W}_t)}{2}$) by showing the loss $\mathcal{L}(\boldsymbol{W}_t)$ decreases exponentially fast. We first choose a time $t_1$ after $t_0$ (recall that $t_0$ is the time that satisfies Assumption 3) such that $\mathcal{L}(\boldsymbol{W}_{t+1}) \leq \mathcal{L}(\boldsymbol{W}_t) - \frac{\eta_t\gamma}{2}\mathcal{G}(\boldsymbol{W}_t)$ for all $t \geq t_1$. Next, we lower bound $G(\boldsymbol{W}_t)$ using $\mathcal{L}(\boldsymbol{W}_t)$. By Lemma 18, there are two sufficient conditions (namely, $\mathcal{L}(\boldsymbol{W}_t) \leq \frac{\log 2}{n} =: \tilde{\mathcal{L}}$ or $\boldsymbol{G}(\boldsymbol{W}_t) \leq \frac{1}{2n}$) that guarantee $\mathcal{L}(\boldsymbol{W}_t) \leq 2\mathcal{G}(\boldsymbol{W}_t)$. We choose a time $t_2$ (after $t_1$) that is sufficiently large such that there exists $t^* \in [t_1, t_2]$ for which we have $\mathcal{G}(\boldsymbol{W}_{t^*}) \leq \frac{\tilde{\mathcal{L}}}{2} \leq \frac{1}{2n}$. This not only guarantees that $\mathcal{L}(\boldsymbol{W}_{t^*}) \leq 2\mathcal{G}(\boldsymbol{W}_{t^*})$ at time $t^*$, but also (crucially due to monotonicity) implies that $\mathcal{L}(\boldsymbol{W}_t) \leq \mathcal{L}(\boldsymbol{W}_{t^*}) \leq 2\mathcal{G}(\boldsymbol{W}_{t^*}) \leq \frac{\log 2}{n}$ for all $t \geq t_2$. Thus, we observe that the other sufficient condition $\mathcal{L}(\boldsymbol{W}_t) \leq \frac{\log 2}{n}$ is satisfied, from which we conclude that $\mathcal{L}(\boldsymbol{W}_t) \leq 2\mathcal{G}(\boldsymbol{W}_t)$ for all $t \geq t_2$. We remark that the choice of $t_2$ depends on $\mathcal{L}(\boldsymbol{W}_{t_1})$ (whose magnitude is bounded using Lemma 17), and $t_2$ can be used as $\bar{t}_2$ above. To recap, $t_1$ is the time (after $t_0$) after which the successive loss decrease is lower bounded by the product $\eta_t\gamma\mathcal{G}(\boldsymbol{W}_t)$;

$t_2$ (after $t_1$) is the time after which $\mathcal{L}(\boldsymbol{W}_t) \leq \frac{\log 2}{n}$ (thus, both $\mathcal{L}(\boldsymbol{W}_t) \leq 2\mathcal{G}(\boldsymbol{W}_t)$ and separability condition (32) hold for all $t \geq t_2$).

In this following, we break the proof of implicit bias of NSD into several parts following previous arguments. Lemma 21 shows the descent properties of NSD. It is used in Lemma 22 to lower bound the un-normalized margin, and in the proof of Theorem 3 to show the convergence of $\frac{\mathcal{G}(\boldsymbol{W}_t)}{\mathcal{L}(\boldsymbol{W}_t)}$.

**Lemma 21** (NSD Descent). *Under the same setting as Theorem 3, it holds for all $t \geq 0$,*

$$\mathcal{L}(\boldsymbol{W}_{t+1}) \leq \mathcal{L}(\boldsymbol{W}_t) - \gamma \eta_t (1 - \alpha_{s_1} \eta_t) \mathcal{G}(\boldsymbol{W}_t),$$

*where $\alpha_{s_1}$ is some constant that depends on $B$ and $\gamma$.*

*Proof.* By Lemma 10, we let $\boldsymbol{W}' = \boldsymbol{W}_{t+1}$, $\boldsymbol{W} = \boldsymbol{W}_t$, $\tilde{\boldsymbol{\Delta}}_t = \boldsymbol{W}_{t+1} - \boldsymbol{W}_t$, and define $\boldsymbol{W}_{t,t+1,\zeta} := \boldsymbol{W}_t + \zeta(\boldsymbol{W}_{t+1} - \boldsymbol{W}_t)$. We choose $\zeta^*$ such that $\boldsymbol{W}_{t,t+1,\zeta^*}$ satisfies (14), we have:

$$\mathcal{L}(\boldsymbol{W}_{t+1}) = \mathcal{L}(\boldsymbol{W}_t) + \underbrace{\langle \nabla \mathcal{L}(\boldsymbol{W}_t), \tilde{\boldsymbol{\Delta}}_t \rangle}_{\spadesuit_t}$$

$$+ \frac{1}{2n} \sum_{i \in [n]} \underbrace{\boldsymbol{h}_i^\top \tilde{\boldsymbol{\Delta}}_t^\top \left( \operatorname{diag}(\mathbb{S}(\boldsymbol{W}_{t,t+1,\gamma} \boldsymbol{h}_i)) - \mathbb{S}(\boldsymbol{W}_{t,t+1,\zeta^*} \boldsymbol{h}_i) \mathbb{S}(\boldsymbol{W}_{t,t+1,\zeta^*} \boldsymbol{h}_i)^\top \right) \tilde{\boldsymbol{\Delta}}_t \boldsymbol{h}_i}_{\clubsuit_t}.$$

$$(21)$$

For the $\spadesuit_t$ term, we have by Lemma 16:

$$\spadesuit_t = -\eta_t \|\nabla \mathcal{L}(\boldsymbol{W}_t)\|_* \leq -\eta_t \gamma G(\boldsymbol{W}_t).$$

For the $\clubsuit_t$ term, we let $\boldsymbol{v} = \tilde{\boldsymbol{\Delta}}_t \boldsymbol{h}_i$ and $\boldsymbol{s} = \mathbb{S}(\boldsymbol{W}_{t,t+1,\zeta^*} \boldsymbol{h}_i)$, and apply Lemma 14 to obtain

$$\clubsuit_t \leq 4\|\tilde{\boldsymbol{\Delta}}_t\|^2 \|\boldsymbol{h}_i\|_*^2 (1 - \mathbb{S}_{y_i}(\boldsymbol{W}_{t,t+1,\zeta^*} \boldsymbol{h}_i)) \leq 4\eta_t^2 B^2 (1 - \mathbb{S}_{y_i}(\boldsymbol{W}_{t,t+1,\zeta^*} \boldsymbol{h}_i)),$$

where in the second inequality we have used $\|\tilde{\boldsymbol{\Delta}}_t\| \leq \eta_t$ and $\|\boldsymbol{h}_i\|_* \leq \|\boldsymbol{h}_i\|_1 \leq 1$. Putting these two pieces together, we obtain

$$\mathcal{L}(\boldsymbol{W}_{t+1}) \leq \mathcal{L}(\boldsymbol{W}_t) - \gamma \eta_t \mathcal{G}(\boldsymbol{W}_t) + 2\eta_t^2 B^2 \frac{1}{n} \sum_{i \in [n]} (1 - \mathbb{S}_{y_i}(\boldsymbol{W}_{t,t+1,\zeta^*} \boldsymbol{h}_i))$$

$$= \mathcal{L}(\boldsymbol{W}_t) - \gamma \eta_t \mathcal{G}(\boldsymbol{W}_t) + 2\eta_t^2 B^2 \mathcal{G}(\boldsymbol{W}_{t,t+1,\zeta^*})$$

$$\leq \mathcal{L}(\boldsymbol{W}_t) - \gamma \eta_t \mathcal{G}(\boldsymbol{W}_t) + 2\eta_t^2 B^2 \sup_{\zeta \in [0,1]} \mathcal{G}(\boldsymbol{W}_{t,t+1,\zeta})$$

$$= \mathcal{L}(\boldsymbol{W}_t) - \gamma \eta_t \mathcal{G}(\boldsymbol{W}_t) + 2\eta_t^2 B^2 \mathcal{G}(\boldsymbol{W}_t) \sup_{\zeta \in [0,1]} \frac{\mathcal{G}(\boldsymbol{W}_t + \zeta \tilde{\boldsymbol{\Delta}}_t)}{\mathcal{G}(\boldsymbol{W}_t)}$$

$$\overset{(a)}{\leq} \mathcal{L}(\boldsymbol{W}_t) - \gamma \eta_t \mathcal{G}(\boldsymbol{W}_t) + 2\eta_t^2 B^2 \mathcal{G}(\boldsymbol{W}_t) \sup_{\zeta \in [0,1]} e^{2B\zeta\|\tilde{\boldsymbol{\Delta}}_t\|}$$

$$\overset{(b)}{\leq} \mathcal{L}(\boldsymbol{W}_t) - \gamma \eta_t \mathcal{G}(\boldsymbol{W}_t) + 2\eta_t^2 B^2 e^{2B\eta_0} \mathcal{G}(\boldsymbol{W}_t), \quad (22)$$

where (a) is by Lemma 20 and (b) is by $\|\tilde{\boldsymbol{\Delta}}_t\| \leq \eta_t$. Letting $\alpha_{s_1} = \frac{2B^2 e^{2B\eta_0}}{\gamma}$, Eq. (22) simplifies to:

$$\mathcal{L}(\boldsymbol{W}_{t+1}) \leq \mathcal{L}(\boldsymbol{W}_t) - \gamma \eta_t (1 - \alpha_{s_1} \eta_t) \mathcal{G}(\boldsymbol{W}_t),$$

from which we observe that the loss starts to monotonically decrease after $\eta_t$ satisfies $\eta_t \leq \frac{1}{\alpha_{s_1}}$ for a decreasing learning rate schedule. $\square$

For a decaying learning rate schedule, Lemma 21 implies that the loss monotonically decreases after a certain time. Thus, we know that the assumption of Lemma 22 can be satisfied. In the proof of Theorem 3, we will specify a concrete form of $\tilde{t}$ in Lemma 22.

**Lemma 22** (NSD Unnormalized Margin). *Suppose that there exist $\tilde{t}$ such that $\mathcal{L}(\boldsymbol{W}_t) \leq \frac{\log 2}{n}$ for all $t > \tilde{t}$, then we have*

$$\min_{i \in [n], c \neq y_i} (\boldsymbol{e}_{y_i} - \boldsymbol{e}_c)^T \boldsymbol{W}_t \boldsymbol{h}_i \geq \gamma \sum_{s=\tilde{t}}^{t-1} \eta_s \frac{\mathcal{G}(\boldsymbol{W}_s)}{\mathcal{L}(\boldsymbol{W}_s)} - \alpha_{s_2} \sum_{s=\tilde{t}}^{t-1} \eta_s^2,$$

*where $\alpha_{s_2}$ is some constant that depends on $B$.*

*Proof.* We let $\alpha_{s_2} = 2Be^{2B\eta_0}$, then from (22), we have for $t > \tilde{t}$:

$$\mathcal{L}(\boldsymbol{W}_{t+1}) \leq \mathcal{L}(\boldsymbol{W}_t) - \gamma\eta_t \mathcal{G}(\boldsymbol{W}_t) + \alpha_{s_2}\eta_t^2 \mathcal{G}(\boldsymbol{W}_t)$$

$$= \mathcal{L}(\boldsymbol{W}_t)\Big(1 - \gamma\eta_t \frac{\mathcal{G}(\boldsymbol{W}_t)}{\mathcal{L}(\boldsymbol{W}_t)} + \alpha_{s_2}\eta_t^2 \frac{\mathcal{G}(\boldsymbol{W}_t)}{\mathcal{L}(\boldsymbol{W}_t)}\Big)$$

$$\leq \mathcal{L}(\boldsymbol{W}_t)\exp\Big(-\gamma\eta_t \frac{\mathcal{G}(\boldsymbol{W}_t)}{\mathcal{L}(\boldsymbol{W}_t)} + \alpha_{s_2}\eta_t^2 \frac{\mathcal{G}(\boldsymbol{W}_t)}{\mathcal{L}(\boldsymbol{W}_t)}\Big)$$

$$\leq \mathcal{L}(\boldsymbol{W}_{\tilde{t}})\exp\Big(-\gamma\sum_{s=\tilde{t}}^{t}\eta_s \frac{\mathcal{G}(\boldsymbol{W}_s)}{\mathcal{L}(\boldsymbol{W}_s)} + \alpha_{s_2}\sum_{s=\tilde{t}}^{t}\eta_s^2\Big).$$

$$\leq \frac{\log 2}{n}\exp\Big(-\gamma\sum_{s=\tilde{t}}^{t}\eta_s \frac{\mathcal{G}(\boldsymbol{W}_s)}{\mathcal{L}(\boldsymbol{W}_s)} + \alpha_{s_2}\sum_{s=\tilde{t}}^{t}\eta_s^2\Big), \tag{23}$$

where the penultimate inequality uses Lemma 18, and the last inequality uses the assumption that $\mathcal{L}(\boldsymbol{W}_t) \leq \frac{\log 2}{n}$ for all $t \geq \tilde{t}$. Then, we have for all $t > \tilde{t}$:

$$e^{-\min_{i \in [n], c \neq y_i} (\boldsymbol{e}_{y_i} - \boldsymbol{e}_c)^T \boldsymbol{W}_t \boldsymbol{h}_i} = \max_{i \in [n]} e^{-\min_{c \neq y_i}(\boldsymbol{e}_{y_i} - \boldsymbol{e}_c)^T \boldsymbol{W}_t \boldsymbol{h}_i}$$

$$\overset{(a)}{\leq} \max_{i \in [n]} \frac{1}{\log 2}\log\big(1 + e^{-\min_{c \neq y_i}(\boldsymbol{e}_{y_i} - \boldsymbol{e}_c)^T \boldsymbol{W}_t \boldsymbol{h}_i}\big)$$

$$\leq \max_{i \in [n]} \frac{1}{\log 2}\log\big(1 + \sum_{c \neq y_i} e^{-(\boldsymbol{e}_{y_i} - \boldsymbol{e}_c)^T \boldsymbol{W}_t \boldsymbol{h}_i}\big) \leq \frac{n\mathcal{L}(\boldsymbol{W}_t)}{\log 2}$$

$$\overset{(b)}{\leq} \exp\Big(-\gamma\sum_{s=\tilde{t}}^{t-1}\eta_s \frac{\mathcal{G}(\boldsymbol{W}_s)}{\mathcal{L}(\boldsymbol{W}_s)} + \alpha_{s_2}\sum_{s=\tilde{t}}^{t-1}\eta_s^2\Big).$$

(a) is by the following: the assumption $\mathcal{L}(\boldsymbol{W}_t) \leq \frac{\log 2}{n}$ implies that $\min_{c \neq y_i}(\boldsymbol{e}_{y_i} - \boldsymbol{e}_c)^T \boldsymbol{W}_t \boldsymbol{h}_i \geq 0$ for all $i \in [n]$ by Lemma 19. We also know the inequality $\frac{\log(1+e^{-z})}{e^{-z}} \geq \log 2$ holds for any $z \geq 0$. Then, for any $i \in [n]$, we can set $z = \min_{c \neq y_i}(\boldsymbol{e}_{y_i} - \boldsymbol{e}_c)^T \boldsymbol{W}_t \boldsymbol{h}_i$ to obtain the desired inequality; and (b) is by (23). Finally, taking log on both sides leads to the result. □

Next Lemma upper bounds the p-norm of NSD's iterates using learning rates. It is used in the proof of Theorem 3.

**Lemma 23** (NSD $\|\boldsymbol{W}_t\|$). *For NSD, we have for any $t > 0$ that*

$$\|\boldsymbol{W}_t\| \leq \|\boldsymbol{W}_0\| + \sum_{s=0}^{t-1}\eta_s.$$

*Proof.* By the NSD update rule (3), we have

$$\boldsymbol{W}_{t+1} = \boldsymbol{W}_0 - \sum_{s=0}^{t}\eta_s \boldsymbol{\Delta}_s.$$

This leads to $\|\boldsymbol{W}_t\| \leq \|\boldsymbol{W}_0\| + \sum_{s=0}^{t-1}\eta_s$ given $\boldsymbol{\Delta}_s \leq 1$ for all $s \geq 0$. □

The main step in the proof of Theorem 3 is to determine the time that satisfies the assumption in Lemma 22 and show the convergence of $\frac{\mathcal{G}(\boldsymbol{W}_t)}{\mathcal{L}(\boldsymbol{W}_t)}$. Then, Lemma 22 and Lemma 23 will be combined to obtain the final result.

**Theorem 3.** *Suppose that Assumption 1, 2, and 4 hold, then there exists $t_{s_2} = t_{s_2}(n, \gamma, B, \boldsymbol{W}_0)$ such that NSD achieves the following for all $t > t_{s_2}$*

$$\left| \frac{\min_{i \in [n], c \neq y_i}(\boldsymbol{e}_{y_i} - \boldsymbol{e}_c)^T \boldsymbol{W}_t \boldsymbol{h}_i}{\|\boldsymbol{W}_t\|} - \gamma \right| \leq \mathcal{O}\left( \frac{\sum_{s=t_{s_2}}^{t-1} \eta_s e^{-\frac{\gamma}{4}\sum_{\tau=t_{s_2}}^{s-1}\eta_\tau} + \sum_{s=0}^{t_{s_2}-1}\eta_s + \sum_{s=t_{s_2}}^{t-1}\eta_s^2}{\sum_{s=0}^{t-1}\eta_s} \right).$$

*Proof.* **Determination of $t_{s_1}$.** In Lemma 21 we choose $t_{s_1}$ such that $\eta_t \leq \frac{1}{2\alpha_{s_1}}$ for all $t \geq t_{s_1}$. Considering $\eta_t = \Theta(\frac{1}{t^a})$ (where $a \in (0, 1]$), we set $t_{s_1} = (2\alpha_{s_1})^{\frac{1}{a}} = (\frac{4B^2 e^{2B\eta_0}}{\gamma})^{\frac{1}{a}}$. Then, we have for all $t \geq t_{s_1}$

$$\mathcal{L}(\boldsymbol{W}_{t+1}) \leq \mathcal{L}(\boldsymbol{W}_t) - \frac{\eta_t \gamma}{2}\mathcal{G}(\boldsymbol{W}_t). \tag{24}$$

Rearranging this equation and using non-negativity of the loss we obtain $\gamma \sum_{s=t_{s_1}}^{t}\eta_s \mathcal{G}(\boldsymbol{W}_s) \leq 2\mathcal{L}(\boldsymbol{W}_{t_{s_1}})$.

**Determination of $t_{s_2}$.** By Lemma 17, we can bound $\mathcal{L}(\boldsymbol{W}_{t_{s_1}})$ as follows

$$|\mathcal{L}(\boldsymbol{W}_{t_{s_1}}) - \mathcal{L}(\boldsymbol{W}_0)| \leq 2B\|\boldsymbol{W}_{t_{s_1}} - \boldsymbol{W}_0\| \leq 2B\sum_{s=0}^{t_{s_1}-1}\eta_s\|\boldsymbol{\Delta}_s\| \leq 2B\sum_{s=0}^{t_{s_1}-1}\eta_s,$$

where the last inequality is by $\|\boldsymbol{\Delta}_s\| \leq 1$ for all $s \geq 0$. Combining this with the result above and letting $\tilde{\mathcal{L}} := \frac{\log 2}{n}$, we obtain

$$\mathcal{G}(\boldsymbol{W}_{t^*}) = \min_{s \in [t_{s_1}, t_{s_2}]}\mathcal{G}(\boldsymbol{W}_s) \leq \frac{2\mathcal{L}(\boldsymbol{W}_0) + 4B\sum_{s=0}^{t_{s_1}-1}\eta_s}{\gamma\sum_{s=t_{s_1}}^{t_{s_2}}\eta_s} \leq \frac{\tilde{\mathcal{L}}}{2} \leq \frac{1}{2n},$$

from which we derive the sufficient condition on $t_{s_2}$ to be $\sum_{s=t_{s_1}}^{t_{s_2}}\eta_s \geq \frac{4\mathcal{L}(\boldsymbol{W}_0)+8B\sum_{s=0}^{t_{s_1}-1}\eta_s}{\gamma\tilde{\mathcal{L}}}$.

**Convergence of $\frac{\mathcal{G}(\boldsymbol{W}_t)}{\mathcal{L}(\boldsymbol{W}_t)}$** Given $\mathcal{G}(\boldsymbol{W}_{t^*}) \leq \frac{\tilde{\mathcal{L}}}{2} \leq \frac{1}{2n}$, we obtain that $\mathcal{L}(\boldsymbol{W}_t) \leq \mathcal{L}(\boldsymbol{W}_{t^*}) \leq 2\mathcal{G}(\boldsymbol{W}_{t^*}) \leq \tilde{L}$ for all $t \geq t_{s_2}$, where the first and second inequalities are due to monotonicity in the risk and Lemma 18, respectively. Thus, the other sufficient condition $\mathcal{L}(\boldsymbol{W}_t) \leq \frac{\log 2}{n}$ in Lemma 18 is satisfied, from which we conclude that $\mathcal{L}(\boldsymbol{W}_t) \leq 2\mathcal{G}(\boldsymbol{W}_t)$ for all $t \geq t_{s_2}$. Substituting this into (24), we obtain for all $t > t_{s_2}$

$$\mathcal{L}(\boldsymbol{W}_t) \leq (1 - \frac{\gamma\eta_{t-1}}{4})\mathcal{L}(\boldsymbol{W}_{t-1}) \leq \mathcal{L}(\boldsymbol{W}_{t_{s_2}})e^{-\frac{\gamma}{4}\sum_{s=t_{s_2}}^{t-1}\eta_s} \leq \tilde{\mathcal{L}}e^{-\frac{\gamma}{4}\sum_{s=t_{s_2}}^{t-1}\eta_s}$$

Then, by Lemma 18, we obtain

$$\frac{\mathcal{G}(\boldsymbol{W}_t)}{\mathcal{L}(\boldsymbol{W}_t)} \geq 1 - \frac{n\mathcal{L}(\boldsymbol{W}_t)}{2} \geq 1 - \frac{n\tilde{\mathcal{L}}e^{-\frac{\gamma}{4}\sum_{s=t_{s_2}}^{t-1}\eta_s}}{2} \geq 1 - e^{-\frac{\gamma}{4}\sum_{s=t_{s_2}}^{t-1}\eta_s}. \tag{25}$$

**Margin Convergence** Finally, we combine Lemma 22, Lemma 23, and (25) to obtain

$$|\frac{\min_{i \in [n], c \neq y_i}(\boldsymbol{e}_{y_i} - \boldsymbol{e}_c)^T \boldsymbol{W}_t \boldsymbol{h}_i}{\|\boldsymbol{W}_t\|} - \gamma| \leq \frac{\gamma(\|\boldsymbol{W}_0\| + \sum_{s=t_{s_2}}^{t-1}\eta_s e^{-\frac{\gamma}{4}\sum_{\tau=t_{s_2}}^{s-1}\eta_\tau} + \sum_{s=0}^{t_{s_2}-1}\eta_s) + \alpha_{s_2}\sum_{s=t_{s_2}}^{t-1}\eta_s^2}{\|\boldsymbol{W}_0\| + \sum_{s=0}^{t-1}\eta_s}$$

$$\leq \mathcal{O}(\frac{\sum_{s=t_{s_2}}^{t-1}\eta_s e^{-\frac{\gamma}{4}\sum_{\tau=t_{s_2}}^{s-1}\eta_\tau} + \sum_{s=0}^{t_{s_2}-1}\eta_s + \sum_{s=t_{s_2}}^{t-1}\eta_s^2}{\sum_{s=0}^{t-1}\eta_s})$$

$\square$

Next, we explicitly upper bound $t_{s_2}$ in Theorem 3 to derive the margin convergence rates of NSD.

**Corollary 1.** *Consider learning rate schedule of the form $\eta_t = \Theta(\frac{1}{t^a})$ where $a \in (0,1]$, under the same setting as Theorem 3, then we have for SignGD*

$$|\frac{\min_{i \in [n], c \neq y_i}(\boldsymbol{e}_{y_i} - \boldsymbol{e}_c)^T \boldsymbol{W}_t \boldsymbol{h}_i}{\|\boldsymbol{W}_t\|} - \gamma| = \begin{cases} \mathcal{O}(\frac{t^{1-2a}+n}{t^{1-a}}) & \text{if} \quad a < \frac{1}{2} \\ \mathcal{O}(\frac{\log t + n}{t^{1/2}}) & \text{if} \quad a = \frac{1}{2} \\ \mathcal{O}(\frac{n}{t^{1-a}}) & \text{if} \quad \frac{1}{2} < a < 1 \\ \mathcal{O}(\frac{n}{\log t}) & \text{if} \quad a = 1 \end{cases}$$

*Proof.* Recall that $t_{s_1} = (\frac{4B^2 e^{2B\eta_0}}{\gamma})^{\frac{1}{a}} =: C_{s_1}$, and the condition on $t_{s_2}$ is $\sum_{s=t_{s_1}}^{t_{s_2}} \eta_s \geq \frac{4\mathcal{L}(\boldsymbol{W}_0) + 8B\sum_{s=0}^{t_{s_1}-1}\eta_s}{\gamma\tilde{\mathcal{L}}}$, where $\tilde{L} = \frac{\log 2}{n}$. We can apply integral approximations to the terms that involve sums of learning rates to obtain

$$t_{s_2} \leq C_{s_2} n^{\frac{1}{1-a}} t_{s_1} + C_{s_3} n^{\frac{1}{1-a}} \mathcal{L}(\boldsymbol{W}_0)^{\frac{1}{1-a}}.$$

Given $t_{s_1}$ is some constant, this further implies that

$$\sum_{s=0}^{t_{s_2}-1} \eta_s = \mathcal{O}(t_{s_2}^{1-a}) = \mathcal{O}(n + n\mathcal{L}(\boldsymbol{W}_0)).$$

Next, we focus on the term $\sum_{s=t_{s_2}}^{t-1} \eta_s^2$. For $a > \frac{1}{2}$, this term can be bounded by some constant. For $a < \frac{1}{2}$, we have $\sum_{s=t_{s_2}}^{t-1} \eta_s^2 = \mathcal{O}(t^{1-2a})$, and it evaluates to $\mathcal{O}(\log t)$ for $a = \frac{1}{2}$. Finally, we have that $\sum_{s=0}^{t-1} \eta_s = \mathcal{O}(t^{1-a})$ for $a < 1$ and $\sum_{s=0}^{t-1} \eta_s = \mathcal{O}(\log t)$ for $a = 1$. The term $\sum_{s=t_{s_2}}^{t-1} \eta_s e^{-\frac{\gamma}{4}\sum_{\tau=t_{s_2}}^{s-1}\eta_\tau}$ is bounded by some constant as shown in Zhang et al. [77, Corollary 4.7]. $\square$

# E Implicit Bias of Normalized Momentum Steepest Descent

Recall that $\|\cdot\|$ refer to either entry-wise or Schatten p-norm with its dual norm denoted as $\|\cdot\|_*$.

**Lemma 24.** *Consider the following $\boldsymbol{W}^\dagger := \boldsymbol{W} - \eta\boldsymbol{\Delta}$, where $\boldsymbol{\Delta} \in \mathbb{R}^{k \times d}$ is defined in (4). Let $\boldsymbol{M} \in \mathbb{R}^{k \times d}$ be any matrix. It holds:*

$$\langle \nabla\mathcal{L}(\boldsymbol{W}), \boldsymbol{W}^\dagger - \boldsymbol{W}\rangle \leq 2\eta\|\boldsymbol{\Omega}\|_{\mathrm{sum}} - \eta\gamma\mathcal{G}(\boldsymbol{W}),$$

*where $\boldsymbol{\Omega}$ is defined to be $\boldsymbol{\Omega} := \boldsymbol{M} - \nabla\mathcal{L}(\boldsymbol{W})$.*

*Proof.* We define $\boldsymbol{\Omega} := \boldsymbol{M} - \nabla\mathcal{L}(\boldsymbol{W})$ to obtain

$$\begin{aligned} \langle \nabla\mathcal{L}(\boldsymbol{W}), \boldsymbol{W}^\dagger - \boldsymbol{W}\rangle &= \langle \nabla\mathcal{L}(\boldsymbol{W}) - \boldsymbol{M}, \boldsymbol{W}^\dagger - \boldsymbol{W}\rangle + \langle \boldsymbol{M}, \boldsymbol{W}^\dagger - \boldsymbol{W}\rangle \\ &= -\eta\langle \nabla\mathcal{L}(\boldsymbol{W}) - \boldsymbol{M}, \boldsymbol{\Delta}\rangle - \eta\langle \boldsymbol{M}, \boldsymbol{\Delta}\rangle \\ &\overset{(a)}{\leq} \eta\|\nabla\mathcal{L}(\boldsymbol{W}) - \boldsymbol{M}\|_*\|\boldsymbol{\Delta}\| - \eta\|\boldsymbol{M}\|_* \\ &\overset{(b)}{\leq} \eta\|\boldsymbol{M} - \nabla\mathcal{L}(\boldsymbol{W})\|_* - \eta\|\boldsymbol{M} - \nabla\mathcal{L}(\boldsymbol{W}) + \nabla\mathcal{L}(\boldsymbol{W})\|_* \\ &\overset{(c)}{\leq} \eta\|\boldsymbol{\Omega}\|_{\mathrm{sum}} - \eta\|\boldsymbol{\Omega} - (-\nabla\mathcal{L}(\boldsymbol{W}))\|_* \\ &\overset{(d)}{\leq} \eta\|\boldsymbol{\Omega}\|_{\mathrm{sum}} - \eta(\|\nabla\mathcal{L}(\boldsymbol{W})\|_* - \|\boldsymbol{\Omega}\|_*) \\ &= 2\eta\|\boldsymbol{\Omega}\|_{\mathrm{sum}} - \eta\|\nabla\mathcal{L}(\boldsymbol{W})\|_* \\ &\overset{(e)}{\leq} 2\eta\|\boldsymbol{\Omega}\|_{\mathrm{sum}} - \eta\gamma\mathcal{G}(\boldsymbol{W}), \end{aligned}$$

where (a) is by Cauchy Schwarz inequality and $\langle \boldsymbol{M}, \boldsymbol{\Delta}\rangle = \|\boldsymbol{M}\|_*$, (b) is by $\|\boldsymbol{\Delta}\| \leq 1$, (c) is via Lemma 11, (d) is by reverse triangle inequality, and (e) is via Lemma 16. $\square$

The following Lemma bounds the entries of the momentum $(\boldsymbol{M}_t)$ of NMD in terms of the product of $\eta_t$ with the sum of $\mathcal{G}_c(\boldsymbol{W}_t)$ and $\mathcal{Q}_c(\boldsymbol{W}_t)$.

**Lemma 25.** *Suppose that Ass. 1, 2, 3, and 4 hold. Let $c \in [k]$ and $j \in [d]$. There exists time $t_0$ such that for all $t \geq t_0$:*

$$|\mathbf{M}_t[c, j] - (1 - \beta_1^{t+1})\nabla\mathcal{L}(\boldsymbol{W}_t)[c, j]| \leq \alpha_M \eta_t \big(\mathcal{G}_c(\boldsymbol{W}_t) + \mathcal{Q}_c(\boldsymbol{W}_t)\big),$$

*where $\alpha_M := B(1 - \beta_1)c_2$.*

*Proof.* For any fixed $c \in [k]$ and $j \in [d]$,

$$|\mathbf{M}_t[c, j] - (1 - \beta_1^{t+1})\nabla\mathcal{L}(\boldsymbol{W}_t)[c, j]| = |\sum_{\tau=0}^{t}(1 - \beta_1)\beta_1^\tau\big(\nabla\mathcal{L}(\boldsymbol{W}_{t-\tau})[c, j] - \nabla\mathcal{L}(\boldsymbol{W}_t)[c, j]\big)|$$

$$\leq \sum_{\tau=0}^{t}(1 - \beta_1)\beta_1^\tau \underbrace{|\nabla\mathcal{L}(\boldsymbol{W}_{t-\tau})[c, j] - \nabla\mathcal{L}(\boldsymbol{W}_t)[c, j]|}_{\clubsuit}.$$

$$(26)$$

We first notice that for any $\boldsymbol{W} \in \mathbb{R}^{k \times d}$, we have $\nabla\mathcal{L}(\boldsymbol{W})[c, j] = \boldsymbol{e}_c^T\nabla\mathcal{L}(\boldsymbol{W})\boldsymbol{e}_j = -\frac{1}{n}\sum_{i\in[n]}\boldsymbol{e}_c^T(\boldsymbol{e}_{y_i} - \mathbb{S}(\boldsymbol{W}\boldsymbol{h}_i))\boldsymbol{h}_i^T\boldsymbol{e}_j = -\frac{1}{n}\sum_{i\in[n]}\boldsymbol{e}_c^T(\boldsymbol{e}_{y_i} - \mathbb{S}(\boldsymbol{W}\boldsymbol{h}_i))h_{ij}$. Then, the gradient difference term becomes

$$\clubsuit = |-\frac{1}{n}\sum_{i\in[n]}\boldsymbol{e}_c^T\big(\boldsymbol{e}_{y_i} - \mathbb{S}(\boldsymbol{W}_{t-\tau}\boldsymbol{h}_i)\big)h_{ij} + \frac{1}{n}\sum_{i\in[n]}\boldsymbol{e}_c^T\big(\boldsymbol{e}_{y_i} - \mathbb{S}(\boldsymbol{W}_t\boldsymbol{h}_i)\big)h_{ij}|$$

$$= |\frac{1}{n}\sum_{i\in[n]}\boldsymbol{e}_c^T\big(\mathbb{S}(\boldsymbol{W}_{t-\tau}\boldsymbol{h}_i) - \mathbb{S}(\boldsymbol{W}_t\boldsymbol{h}_i)\big)h_{ij}|$$

$$= |\frac{1}{n}\sum_{i\in[n]}\big(\mathbb{S}_c(\boldsymbol{W}_{t-\tau}\boldsymbol{h}_i) - \mathbb{S}_c(\boldsymbol{W}_t\boldsymbol{h}_i)\big)h_{ij}|$$

$$\leq B\frac{1}{n}\sum_{i\in[n]}|\mathbb{S}_c(\boldsymbol{W}_{t-\tau}\boldsymbol{h}_i) - \mathbb{S}_c(\boldsymbol{W}_t\boldsymbol{h}_i)|$$

$$= B\frac{1}{n}\underbrace{\sum_{i\in[n],y_i\neq c}|\mathbb{S}_c(\boldsymbol{W}_{t-\tau}\boldsymbol{h}_i) - \mathbb{S}_c(\boldsymbol{W}_t\boldsymbol{h}_i)|}_{\clubsuit_1} + B\frac{1}{n}\underbrace{\sum_{i\in[n],y_i=c}|\mathbb{S}_c(\boldsymbol{W}_{t-\tau}\boldsymbol{h}_i) - \mathbb{S}_c(\boldsymbol{W}_t\boldsymbol{h}_i)|}_{\clubsuit_2}$$

Next, we link the $\clubsuit_1$ and $\clubsuit_2$ terms with $\mathcal{G}(\boldsymbol{W})$. Starting with the first term, we obtain:

$$\clubsuit_1 = \frac{1}{n}\sum_{i\in[n],y_i\neq c}\mathbb{S}_c(\boldsymbol{W}_t\boldsymbol{h}_i)|\frac{\mathbb{S}_c(\boldsymbol{W}_{t-\tau}\boldsymbol{h}_i)}{\mathbb{S}_c(\boldsymbol{W}_t\boldsymbol{h}_i)} - 1|$$

$$\overset{(a)}{\leq} \frac{1}{n}\sum_{i\in[n],y_i\neq c}\mathbb{S}_c(\boldsymbol{W}_t\boldsymbol{h}_i)(e^{2\|(\boldsymbol{W}_{t-\tau}-\boldsymbol{W}_t)\boldsymbol{h}_i\|_\infty} - 1)$$

$$\overset{(b)}{\leq} \frac{1}{n}\sum_{i\in[n],y_i\neq c}\mathbb{S}_c(\boldsymbol{W}_t\boldsymbol{h}_i)(e^{2B\|\boldsymbol{W}_{t-\tau}-\boldsymbol{W}_t\|_{\max}} - 1)$$

$$\overset{(c)}{\leq} \big(e^{2B\sum_{s=1}^{\tau}\eta_{t-s}\|\boldsymbol{\Delta}_{t-s}\|_{\max}} - 1\big)\big(\frac{1}{n}\sum_{i\in[n],y_i\neq c}\mathbb{S}_c(\boldsymbol{W}_t\boldsymbol{h}_i)\big)$$

$$\overset{(d)}{\leq} \big(e^{2B\sum_{s=1}^{\tau}\eta_{t-s}} - 1\big)\mathcal{Q}_c(\boldsymbol{W}_t),$$

where (a) is by Lemma 15, (b) is by $\|\boldsymbol{h}_i\|_1 \leq B$ for all $i \in [n]$, (c) is by (3) and triangle inequality, and (d) is by $\|\boldsymbol{\Delta}_{t-s}\|_{\max} \leq \|\boldsymbol{\Delta}_{t-s}\| \leq 1$ (for any entry-wise or Schatten p-norm) and the definition

of $\mathcal{G}(\boldsymbol{W}_t)$. For the second term, we obtain:

$$
\begin{aligned}
\clubsuit_2 &= \frac{1}{n} \sum_{i \in [n], y_i = c} |\mathbb{S}_c(\boldsymbol{W}_{t-\tau}\boldsymbol{h}_i) - \mathbb{S}_c(\boldsymbol{W}_t\boldsymbol{h}_i)| \\
&= \frac{1}{n} \sum_{i \in [n], y_i = c} |\mathbb{S}_{y_i}(\boldsymbol{W}_{t-\tau}\boldsymbol{h}_i) - 1 + 1 - \mathbb{S}_{y_i}(\boldsymbol{W}_t\boldsymbol{h}_i)| \\
&= \frac{1}{n} \sum_{i \in [n], y_i = c} \big(1 - \mathbb{S}_{y_i}(\boldsymbol{W}_t\boldsymbol{h}_i)\big)|\frac{\mathbb{S}_{y_i}(\boldsymbol{W}_{t-\tau}\boldsymbol{h}_i) - 1}{1 - \mathbb{S}_{y_i}(\boldsymbol{W}_t\boldsymbol{h}_i)} + 1| \\
&= \frac{1}{n} \sum_{i \in [n], y_i = c} \big(1 - \mathbb{S}_{y_i}(\boldsymbol{W}_t\boldsymbol{h}_i)\big)|\frac{1 - \mathbb{S}_{y_i}(\boldsymbol{W}_{t-\tau}\boldsymbol{h}_i)}{1 - \mathbb{S}_{y_i}(\boldsymbol{W}_t\boldsymbol{h}_i)} - 1| \\
&\overset{(e)}{\leq} \frac{1}{n} \sum_{i \in [n], y_i = c} \big(1 - \mathbb{S}_{y_i}(\boldsymbol{W}_t\boldsymbol{h}_i)\big)(e^{2\|(\boldsymbol{W}_{t-\tau} - \boldsymbol{W}_t)\boldsymbol{h}_i\|_\infty} - 1) \\
&\overset{(f)}{\leq} \big(e^{2B\sum_{s=1}^\tau \eta_{t-s}} - 1\big)\mathcal{G}_c(\boldsymbol{W}_t),
\end{aligned}
$$

where (e) is by Lemma 15, and (f) is by the same approach taken for $\clubsuit_1$. Based on the upper bounds for $\clubsuit_1$ and $\clubsuit_2$, we obtain the following: $\clubsuit \leq 2B\big(e^{2\alpha B\sum_{s=1}^\tau \eta_{t-s}} - 1\big)(\mathcal{G}_c(\boldsymbol{W}_t) + \mathcal{Q}_c(\boldsymbol{W}_t))$. Then, we substitute this into (26) to obtain:

$$
|\mathbf{M}_t[c,j] - (1-\beta_1^{t+1})\nabla\mathcal{L}(\boldsymbol{W}_t)[c,j]| \leq B(1-\beta_1)(\mathcal{G}_c(\boldsymbol{W}_t) + \mathcal{Q}_c(\boldsymbol{W}_t))\sum_{\tau=0}^t \beta_1^\tau\big(e^{2B\sum_{s=1}^\tau \eta_{t-s}} - 1\big)
$$

$$
\overset{(g)}{\leq} B(1-\beta_1)c_2\eta_t(\mathcal{G}_c(\boldsymbol{W}_t) + \mathcal{Q}_c(\boldsymbol{W}_t)),
$$

where (g) is by the Assumption 3. $\qquad\square$

**Lemma 26.** *Let $\boldsymbol{\Omega}_t = \boldsymbol{M}_t - \nabla\mathcal{L}(\boldsymbol{W}_t)$, where $\boldsymbol{M}_t$ is defined in (5). Then, it holds*

$$
\|\boldsymbol{\Omega}_t\|_{\mathrm{sum}} \leq 2B\beta_1^{t/2}\mathcal{G}(\boldsymbol{W}_t) + 2\alpha_M d\eta_t\mathcal{G}(\boldsymbol{W}_t),
$$

*where $\alpha_M := B(1-\beta_1)c_2$.*

*Proof.* For simplicity, we drop the subscripts $t$. Denote $\mathcal{T}_c(\boldsymbol{W}) := \mathcal{G}_c(\boldsymbol{W}) + \mathcal{Q}_c(\boldsymbol{W})$. Then, by Lemma 25, we have for any $c \in [k]$ and $j \in [d]$:

$$
\begin{aligned}
\boldsymbol{M}[c,j] &= (1-\beta_1^{t+1})\nabla\mathcal{L}(\boldsymbol{W})[c,j] + \alpha_M\eta\mathcal{T}_c(\boldsymbol{W})\epsilon_{m,c,j} \\
&= \nabla\mathcal{L}(\boldsymbol{W})[c,j] - \beta_1^{t+1}\nabla\mathcal{L}(\boldsymbol{W})[c,j] + \alpha_M\eta\mathcal{T}_c(\boldsymbol{W})\epsilon_{m,c,j},
\end{aligned}
$$

where $\alpha_M := B(1-\beta_1)c_2$ and $\epsilon_{m,c,j}$ is some constant s.t. $|\epsilon_{m,c,j}| \leq 1$. Recall that $\boldsymbol{\Omega} := \boldsymbol{M} - \nabla\mathcal{L}(\boldsymbol{W})$, then we have

$$
\begin{aligned}
|\boldsymbol{\Omega}[c,j]| &= |\boldsymbol{M}[c,j] - \nabla\mathcal{L}(\boldsymbol{W})[c,j]| \\
&= |-\beta_1^{t+1}\nabla\mathcal{L}(\boldsymbol{W})[c,j] + \alpha_M\eta\mathcal{T}_c(\boldsymbol{W})\epsilon_{m,c,j}| \\
&\leq \beta_1^{t+1}|\nabla\mathcal{L}(\boldsymbol{W})[c,j]| + \alpha_M\eta\mathcal{T}_c(\boldsymbol{W}).
\end{aligned}
$$

This implies the following:

$$
\begin{aligned}
\|\boldsymbol{\Omega}\|_{\mathrm{sum}} = \sum_{c,j} |\boldsymbol{\Omega}[c,j]| &\leq \beta_1^{t+1}\sum_{c,j} |\nabla\mathcal{L}(\boldsymbol{W})[c,j]| + \alpha_M\eta\sum_{c,j}\mathcal{T}_c(\boldsymbol{W}) \\
&= \beta_1^{t+1}\|\nabla\mathcal{L}(\boldsymbol{W})\|_{\mathrm{sum}} + 2\alpha_M d\eta\mathcal{G}(\boldsymbol{W}) \\
&\leq 2B\beta_1^{t/2}\mathcal{G}(\boldsymbol{W}) + 2\alpha_M d\eta\mathcal{G}(\boldsymbol{W}),
\end{aligned}
$$

where in the last inequality we have used Lemma 16. $\qquad\square$

**Lemma 27.** *Suppose that there exist $\tilde{t}$ such that $\mathcal{L}(\boldsymbol{W}_t) \leq \frac{\log 2}{n}$ for all $t > \tilde{t}$, then we have*

$$\min_{i \in [n], c \neq y_i} (\boldsymbol{e}_{y_i} - \boldsymbol{e}_c)^T \boldsymbol{W}_t \boldsymbol{h}_i \geq \gamma \sum_{s=\tilde{t}}^{t-1} \eta_s \frac{\mathcal{G}(\boldsymbol{W}_s)}{\mathcal{L}(\boldsymbol{W}_s)} - a_2 \sum \eta_s^2 - Q$$

*where $a_2 = (4\alpha_M + 2B^2 e^{2B\eta_0})d$ and $Q = 4B\eta_0 \frac{1}{1-\beta_1^{1/2}}$.*

*Proof.* We follow a similar approach as Lemma 21 to show the descent of NMD. Specifically, we apply Lemma 24 to bound the first-order term. For the Hessian term, we apply Lemma 14 and Lemma 20 similar to NSD. Then, we can obtain the following

$$\mathcal{L}(\boldsymbol{W}_{t+1}) \leq \mathcal{L}(\boldsymbol{W}_t) - \eta_t \gamma \mathcal{G}(\boldsymbol{W}_t) + 2\eta_t \|\boldsymbol{\Omega}_t\|_{\text{sum}} + 2\eta_t^2 B^2 e^{2B\eta_0} \mathcal{G}(\boldsymbol{W}_t)$$

$$\overset{(a)}{\leq} \mathcal{L}(\boldsymbol{W}_t) - \eta_t \gamma \mathcal{G}(\boldsymbol{W}_t) + 4B\beta_1^{t/2} \eta_t \mathcal{G}(\boldsymbol{W}_t) + 4\alpha_M \eta_t^2 d\mathcal{G}(\boldsymbol{W}_t) + 2\eta_t^2 B^2 e^{2B\eta_0} \mathcal{G}(\boldsymbol{W}_t)$$

$$\overset{(b)}{\leq} \mathcal{L}(\boldsymbol{W}_t) - \eta_t \gamma \mathcal{G}(\boldsymbol{W}_t) + a_1 \beta_1^{t/2} \eta_t \mathcal{G}(\boldsymbol{W}_t) + a_2 \eta_t^2 d\mathcal{G}(\boldsymbol{W}_t)$$

$$\leq \mathcal{L}(\boldsymbol{W}_{\tilde{t}}) \exp\Big(-\gamma \sum_{s=\tilde{t}}^{t} \eta_s \frac{\mathcal{G}(\boldsymbol{W}_s)}{\mathcal{L}(\boldsymbol{W}_s)} + a_1 \sum_{s=\tilde{t}}^{t} \beta_1^{s/2} \eta_s + a_2 d \sum_{s=\tilde{t}}^{t} \eta_s^2\Big)$$

$$\overset{(c)}{\leq} \frac{\log 2}{n} \exp\Big(-\gamma \sum_{s=\tilde{t}}^{t} \eta_s \frac{\mathcal{G}(\boldsymbol{W}_s)}{\mathcal{L}(\boldsymbol{W}_s)} + a_2 d \sum_{s=\tilde{t}}^{t} \eta_s^2 + Q\Big),$$

where (a) is by Lemma 24. In (b), we have defined $a_1 := 4B$ and $a_2 = (4\alpha_M + 2B^2 e^{2B\eta_0})d$. In (c), we have used the assumption and defined $Q := a_1 \eta_0 \frac{1}{1-\beta_1^{1/2}} \geq a_1 \sum_{s=\tilde{t}}^{t} \beta_1^{s/2} \eta_s$. The rest of the proof follows the same steps as Lemma 22. $\square$

**Theorem 4.** *Suppose that Ass. 1, 2, 3, and 4 hold. Set learning rate $\eta_t = \Theta(\frac{1}{t^{1/2}})$. The margin gap of NMD's iterates satisfy*

$$\gamma - \frac{\min_{i \in [n], c \neq y_i} (\boldsymbol{e}_{y_i} - \boldsymbol{e}_c)^T \boldsymbol{W}_t \boldsymbol{h}_i}{\|\boldsymbol{W}_t\|} \leq O\Big(\frac{d\log t + dn}{t^{1/2}}\Big).$$

*Proof.* Given the updates of NMD are normalized (i.e., $\|\boldsymbol{\Delta}\| \leq 1$), we can obtain the following via Lemma 27:

$$\gamma - \frac{\min_{i \in [n], c \neq y_i} (\boldsymbol{e}_{y_i} - \boldsymbol{e}_c)^T \boldsymbol{W}_t \boldsymbol{h}_i}{\|\boldsymbol{W}_t\|} \leq \frac{\gamma(\|W_0\| + \sum_{s=0}^{t_2-1} \eta_s + \sum_{s=t_2}^{t-1} \eta_s e^{\frac{\gamma}{4}\sum_{\tau=t_2}^{s-1}\eta_\tau}) + a_2 d \sum_{s=t_2}^{t-1} \eta_s^2 + Q}{\|W_0\| + \sum_{s=0}^{t-1} \eta_s}$$

$$\leq O\Big(\frac{\sum_{s=t_2}^{t-1} \eta_s e^{-\frac{\gamma}{4}\sum_{\tau=t_2}^{s-1}\eta_\tau} + \sum_{s=0}^{t_2-1} \eta_s + d \sum_{s=t_2}^{t-1} \eta_s^2}{\sum_{s=0}^{t-1} \eta_s}\Big).$$

Then, we follow the same approach as Corollary 1 for a decreasing learning rate of the form $\eta_t = \Theta(\frac{1}{t^a})$. Specifically, we have $t_1 = \Theta(d^{1/a})$ and $t_2 \leq C_1 n^{\frac{1}{1-a}} t_1 + C_2 n^{\frac{1}{1-a}} L(\boldsymbol{W}_0)^{\frac{1}{1-a}}$. This leads to

$$\sum_{s=0}^{t_2-1} \eta_s = O(t_2^{1-a}) = nt_1^{1-a} + nL(\boldsymbol{W}_0) + d\log(t).$$

Thus, we have the margin gap upper bounded by $O(\frac{nd+d\log(t)}{t^{1/2}})$. $\square$

# F    Other multiclass loss functions

## F.1    Exponential Loss

The multiclass exponential loss is given as

$$\mathcal{L}_{\exp}(\boldsymbol{W}) := \frac{1}{n} \sum_{i \in [n]} \sum_{c \neq y_i} \exp\big(-(\boldsymbol{e}_{y_i} - \boldsymbol{e}_c)^\top \boldsymbol{W} \boldsymbol{h}_i\big).$$

The gradient of $\mathcal{L}_{\exp}(\boldsymbol{W})$ is

$$\nabla \mathcal{L}_{\exp}(\boldsymbol{W}) = \frac{1}{n} \sum_{i \in [n]} \sum_{c \neq y_i} -\exp(-(\boldsymbol{e}_{y_i} - \boldsymbol{e}_c)^T \boldsymbol{W} \boldsymbol{h}_i)(\boldsymbol{e}_{y_i} - \boldsymbol{e}_c)\boldsymbol{h}_i^T.$$

Thus, for any matrix $\boldsymbol{A} \in \mathbb{R}^{k \times d}$, we have

$$\langle \boldsymbol{A}, -\nabla \mathcal{L}_{\exp}(\boldsymbol{W}) \rangle = \frac{1}{n} \sum_{i \in [n]} \sum_{c \neq y_i} \exp\left(-(\boldsymbol{e}_{y_i} - \boldsymbol{e}_c)^\top \boldsymbol{W} \boldsymbol{h}_i\right) \cdot (\boldsymbol{e}_{y_i} - \boldsymbol{e}_c)^\top \boldsymbol{A} \boldsymbol{h}_i.$$

This motivates us to define $\mathcal{G}(\boldsymbol{W})$ as

$$\mathcal{G}_{\exp}(\boldsymbol{W}) = \frac{1}{n} \sum_{i \in [n]} \sum_{c \neq y_i} \exp\left(-(\boldsymbol{e}_{y_i} - \boldsymbol{e}_c)^\top \boldsymbol{W} \boldsymbol{h}_i\right),$$

from which we recognize that $\mathcal{G}_{\exp}(\boldsymbol{W}) = \mathcal{L}_{\exp}(\boldsymbol{W})$. Then, the proof follows similar steps as the CE loss.

### F.2  PairLogLoss

The PairLogLoss loss [72] is given as

$$\mathcal{L}_{\mathrm{pll}}(\boldsymbol{W}) := \frac{1}{n} \sum_{i \in [n]} \sum_{c \neq y_i} \log\left(1 + \exp\left(-(\boldsymbol{e}_{y_i} - \boldsymbol{e}_c)^\top \boldsymbol{W} \boldsymbol{h}_i\right)\right).$$

Note that $\mathcal{L} = \frac{1}{n} \sum_{i \in [n]} \sum_{c \neq y_i} f\left((\boldsymbol{e}_{y_i} - \boldsymbol{e}_c)^\top \boldsymbol{W} \boldsymbol{h}_i\right)$ where $f(t) := \log(1 + e^{-t})$ denotes the logistic loss. Therefore, the Taylor expansion of PLL writes:

$$\mathcal{L}_{\mathrm{pll}}(\boldsymbol{W} + \boldsymbol{\Delta}) = \mathcal{L}(\boldsymbol{W}) + \frac{1}{n} \sum_{i \in [n]} \sum_{c \neq y_i} f'\left((\boldsymbol{e}_{y_i} - \boldsymbol{e}_c)^\top \boldsymbol{W} \boldsymbol{h}_i\right) \cdot (\boldsymbol{e}_{y_i} - \boldsymbol{e}_c)^\top \boldsymbol{\Delta} \boldsymbol{h}_i$$

$$+ \frac{1}{n} \sum_{i \in [n]} \sum_{c \neq y_i} f''\left((\boldsymbol{e}_{y_i} - \boldsymbol{e}_c)^\top \boldsymbol{W} \boldsymbol{h}_i\right) \cdot \boldsymbol{h}_i^\top \boldsymbol{\Delta}^\top (\boldsymbol{e}_{y_i} - \boldsymbol{e}_c)(\boldsymbol{e}_{y_i} - \boldsymbol{e}_c)^\top \boldsymbol{\Delta} \boldsymbol{h}_i + o\left(\|\boldsymbol{\Delta}\|^3\right). \tag{27}$$

From the above, the gradient of the PLL loss is:

$$\nabla \mathcal{L}_{\mathrm{pll}}(\boldsymbol{W}) = \frac{1}{n} \sum_{i \in [n]} \sum_{c \neq y_i} f'\left((\boldsymbol{e}_{y_i} - \boldsymbol{e}_c)^\top \boldsymbol{W} \boldsymbol{h}_i\right) \cdot (\boldsymbol{e}_{y_i} - \boldsymbol{e}_c)\boldsymbol{h}_i^\top$$

$$= \frac{1}{n} \sum_{i \in [n]} \sum_{c \neq y_i} \frac{-\exp\left(-(\boldsymbol{e}_{y_i} - \boldsymbol{e}_c)^\top \boldsymbol{W} \boldsymbol{h}_i\right)}{1 + \exp\left(-(\boldsymbol{e}_{y_i} - \boldsymbol{e}_c)^\top \boldsymbol{W} \boldsymbol{h}_i\right)} (\boldsymbol{e}_{y_i} - \boldsymbol{e}_c)\boldsymbol{h}_i^\top \tag{28}$$

Thus, for any matrix $\boldsymbol{A} \in \mathbb{R}^{k \times d}$,

$$\langle \boldsymbol{A}, -\nabla \mathcal{L}_{\mathrm{pll}}(\boldsymbol{W}) \rangle = \frac{1}{n} \sum_{i \in [n]} \sum_{c \neq y_i} |f'\left((\boldsymbol{e}_{y_i} - \boldsymbol{e}_c)^\top \boldsymbol{W} \boldsymbol{h}_i\right)| \cdot (\boldsymbol{e}_{y_i} - \boldsymbol{e}_c)^\top \boldsymbol{A} \boldsymbol{h}_i. \tag{29}$$

This motivates us to define

$$\mathcal{G}_{\mathrm{pll}}(\boldsymbol{W}) = \frac{1}{n} \sum_{i \in [n]} \sum_{c \neq y_i} |f'\left(-(\boldsymbol{e}_{y_i} - \boldsymbol{e}_c)^\top \boldsymbol{W} \boldsymbol{h}_i\right)| = \frac{1}{n} \sum_{i \in [n]} \sum_{c \neq y_i} \frac{\exp\left(-(\boldsymbol{e}_{y_i} - \boldsymbol{e}_c)^\top \boldsymbol{W} \boldsymbol{h}_i\right)}{1 + \exp\left(-(\boldsymbol{e}_{y_i} - \boldsymbol{e}_c)^\top \boldsymbol{W} \boldsymbol{h}_i\right)} \tag{30}$$

**Lemma 28** (Analogue of Lemma 16 for PLL). *For any $\boldsymbol{W}$, the PairLogLoss (PLL) satisfies:*

$$2B \cdot \mathcal{G}_{\mathrm{pll}}(\boldsymbol{W}) \geq \|\nabla \mathcal{L}_{\mathrm{pll}}(\boldsymbol{W})\| \geq \gamma \cdot \mathcal{G}_{\mathrm{pll}}(\boldsymbol{W}).$$

*Proof.* The lower bound follows immediately from (29) and expressing $\|\nabla \mathcal{L}_{\mathrm{pll}}(\boldsymbol{W})\|_* = \max_{\|\boldsymbol{A}\| \le 1} \langle \boldsymbol{A}, -\nabla \mathcal{L}_{\mathrm{pll}}(\boldsymbol{W}) \rangle$. The lower bound follows from triangle inequality applied to (28):

$$\|\nabla \mathcal{L}_{\mathrm{pll}}(\boldsymbol{W})\|_{\mathrm{sum}} \le \frac{1}{n} \sum_{i \in [n]} \sum_{c \ne y_i} \left| f'\left( -(\boldsymbol{e}_{y_i} - \boldsymbol{e}_c)^\top \boldsymbol{W} \boldsymbol{h}_i \right) \right| \|\boldsymbol{e}_{y_i} - \boldsymbol{e}_c\|_1 \|\boldsymbol{h}_i\|_1 \le 2B \cdot \mathcal{G}(\boldsymbol{W}),$$

and use the relationships in (7), i.e. $\|\nabla \mathcal{L}_{\mathrm{pll}}(\boldsymbol{W})\| \le \|\nabla \mathcal{L}_{\mathrm{pll}}(\boldsymbol{W})\|_{\mathrm{sum}}$ for any entry-wise or Schatten p-norm with $p \ge 1$. $\qquad \square$

For bounding with $\mathcal{G}(\boldsymbol{W})$ the second-order term in the Taylor expansion of PLL, note the following. First, for all $i \in [n], c \ne y_i$:

$$\begin{aligned}
\boldsymbol{h}_i^\top \boldsymbol{\Delta}^\top (\boldsymbol{e}_{y_i} - \boldsymbol{e}_c)(\boldsymbol{e}_{y_i} - \boldsymbol{e}_c)^\top \boldsymbol{\Delta} \boldsymbol{h}_i &= \langle (\boldsymbol{e}_{y_i} - \boldsymbol{e}_c)(\boldsymbol{e}_{y_i} - \boldsymbol{e}_c)^\top, \boldsymbol{\Delta} \boldsymbol{h}_i \boldsymbol{h}_i^\top \boldsymbol{\Delta}^T \rangle \\
&\le \left\| (\boldsymbol{e}_{y_i} - \boldsymbol{e}_c)(\boldsymbol{e}_{y_i} - \boldsymbol{e}_c)^\top \right\|_{\mathrm{sum}} \left\| \boldsymbol{\Delta} \boldsymbol{h}_i \boldsymbol{h}_i^\top \boldsymbol{\Delta}^T \right\|_{\mathrm{max}} \\
&\le \|\boldsymbol{e}_{y_i} - \boldsymbol{e}_c\|_1^2 \cdot (\|\boldsymbol{\Delta} \boldsymbol{h}_i\|_\infty)^2 \\
&\le 4 \cdot (\|\boldsymbol{\Delta}\|_{\mathrm{max}})^2 \cdot \|\boldsymbol{h}_i\|_1^2 \le 4B^2 (\|\boldsymbol{\Delta}\|_{\mathrm{max}})^2 \\
&\le 4B^2 \|\boldsymbol{\Delta}\|^2.
\end{aligned}$$

Second, the (easy to check) property of logistic loss that $f''(t) \le |f'(t)|$. Putting these together:

$$\frac{1}{n} \sum_{i \in [n]} \sum_{c \ne y_i} f''\left( (\boldsymbol{e}_{y_i} - \boldsymbol{e}_c)^\top \boldsymbol{W} \boldsymbol{h}_i \right) \cdot \boldsymbol{h}_i^\top \boldsymbol{\Delta}^\top (\boldsymbol{e}_{y_i} - \boldsymbol{e}_c)(\boldsymbol{e}_{y_i} - \boldsymbol{e}_c)^\top \boldsymbol{\Delta} \boldsymbol{h}_i \le 4B^2 \cdot \mathcal{G}(\boldsymbol{W}) \cdot (\|\boldsymbol{\Delta}\|)^2 .$$

Finally, we verify PLL satisfies Lemma 18.

**Lemma 29** (Analogue of Lemma 18 for PLL). *Let $\boldsymbol{W} \in \mathbb{R}^{k \times d}$, we have*

(i) $1 \ge \frac{\mathcal{G}_{\mathrm{pll}}(\boldsymbol{W})}{\mathcal{L}_{\mathrm{pll}}(\boldsymbol{W})} \ge 1 - \frac{n \mathcal{L}_{\mathrm{pll}}(\boldsymbol{W})}{2}$

(ii) *Suppose that $\boldsymbol{W}$ satisfies $\mathcal{L}_{\mathrm{pll}}(\boldsymbol{W}) \le \frac{\log 2}{n}$ or $\mathcal{G}_{\mathrm{pll}}(\boldsymbol{W}) \le \frac{1}{2n}$, then $\mathcal{L}_{\mathrm{pll}}(\boldsymbol{W}) \le 2\mathcal{G}_{\mathrm{pll}}(\boldsymbol{W})$.*

*Proof.* (i) The upper bound follows by the well-known self-boundedness property of the logistic loss, namely $|f'(t)| \le f(t)$

To prove the upper bound, it suffices to prove for for $x > 0$:

$$\frac{x}{1+x} \ge \log(1+x) - \frac{1}{2} \log^2(1+x). \tag{31}$$

The general case follows by summing over $x_{ic} = \exp\left( -(\boldsymbol{e}_{y_i} - \boldsymbol{e}_c)^\top \boldsymbol{W} \boldsymbol{h}_i \right), i \in [n], c \ne y_i$ since then we have

$$\mathcal{G}(\boldsymbol{W}) = \sum_{i \in [n]} \sum_{c \ne y_i} \frac{x_{ic}}{1 + x_{ic}} \ge \sum_{i \in [n]} \sum_{c \ne y_i} \log(1 + x_{ic}) - \frac{1}{2} \sum_{i \in [n]} \sum_{c \ne y_i} \log^2(1 + x_{ic})$$

$$\ge \sum_{i \in [n]} \sum_{c \ne y_i} \log(1 + x_{ic}) - \frac{1}{2} \left( \sum_{i \in [n]} \sum_{c \ne y_i} \log(1 + x_{ic}) \right)^2,$$

where the last line used $\log(1 + x_{ic}) \ge 0$. For (19), let $a = \log(1 + x) > 0$. The inequality becomes $e^{-a} \le 1 - a + a^2/2$, which holds for $a > 0$ by the second-order Taylor expansion of $e^{-a}$ around 0.

(ii) Denote $\mathcal{L} := \mathcal{L}_{pll}$ and $\mathcal{G} := \mathcal{G}_{pll}$. Given $\mathcal{L} \le \frac{\log(2)}{n} \le \frac{1}{n}$, we have $1 - \frac{n\mathcal{L}}{2} \ge \frac{1}{2}$, then the first part follows from (i). For the second part, denote $l_{ic} := (\boldsymbol{e}_{y_i} - \boldsymbol{e}_c)^\top \boldsymbol{W} \boldsymbol{h}_i, i \in [n], c \ne y_i$. For $\mathcal{L} \le 2\mathcal{G}$ to hold, it is sufficient to show that $\log(1 + e^{-l_{ic}}) \le 2 \frac{e^{-l_{ic}}}{1 + e^{-l_{ic}}}$ for all $i \in [n], c \ne y_i$. This holds true when $l_{ic} \ge -1.366$, which is clearly satisfied given the assumption $\mathcal{G} \le \frac{1}{2n}$ implying $l_{ic} \ge 0$. $\qquad \square$

**Lemma 30** (Analogue of Lemma 20 for PLL). *For any $\psi \in [0, 1]$, we have the following:*

$$\frac{\mathcal{G}_{pll}(\boldsymbol{W} - \psi \eta \boldsymbol{\Delta})}{\mathcal{G}_{pll}(\boldsymbol{W})} \le e^{2B\psi\|\triangle\boldsymbol{W}\|} + 2$$

*Proof.* For logistic loss $f(z) = \log(1 + e^{-z})$, for any $z_1, z_2 \in \mathbb{R}$, we have the following

$$
\begin{aligned}
\left| \frac{f'(z_1)}{f'(z_2)} \right| = \left| \frac{1 + e^{z_2}}{1 + e^{z_1}} \right| &= \left| \frac{1 + e^{z_2} - e^{z_1} + e^{z_1}}{1 + e^{z_1}} \right| \\
&= \left| \frac{e^{z_2} - e^{z_1}}{1 + e^{z_1}} + 1 \right| \leq \left| \frac{e^{z_2} - e^{z_1}}{1 + e^{z_1}} \right| + 1 \\
&\leq \left| e^{z_2 - z_1} - 1 \right| + 1 \\
&\leq e^{|z_2 - z_1|} + 2.
\end{aligned}
$$

Denote $x_{ic}^{\boldsymbol{W}} := (\boldsymbol{e}_{y_i} - \boldsymbol{e}_c)^T \boldsymbol{W} \boldsymbol{h}_i$ and $x_{ic}^{\boldsymbol{W}'} := (\boldsymbol{e}_{y_i} - \boldsymbol{e}_c)^T (\boldsymbol{W} - \psi\eta\boldsymbol{\Delta}) \boldsymbol{h}_i$, then we have for $i \in [n]$, $c \neq y_i$

$$
\begin{aligned}
\frac{f'(x_{ic}^{\boldsymbol{W}'})}{f'(x_{ic}^{\boldsymbol{W}})} = |\frac{f'(x_{ic}^{\boldsymbol{W}'})}{f'(x_{ic}^{\boldsymbol{W}})}| &\leq e^{|x_{ic}^{\boldsymbol{W}} - x_{ic}^{\boldsymbol{W}'}|} + 2 = e^{\psi\eta|(\boldsymbol{e}_c - \boldsymbol{e}_{y_i})^T \boldsymbol{\Delta} \boldsymbol{h}_i|} + 2 = e^{\psi\eta|\langle\boldsymbol{\Delta}, (\boldsymbol{e}_c - \boldsymbol{e}_{y_i})\boldsymbol{h}_i^T\rangle|} + 2 \\
&\leq e^{\psi\eta\|\boldsymbol{\Delta}\|_{\max}\left\|(\boldsymbol{e}_c - \boldsymbol{e}_{y_i})\boldsymbol{h}_i^T\right\|_{\text{sum}}} + 2 \\
&= e^{\psi\eta\|\boldsymbol{\Delta}\|_{\max}\left\|\boldsymbol{e}_c - \boldsymbol{e}_{y_i}\right\|_{\text{sum}}\|\boldsymbol{h}_i\|_{\text{sum}}} + 2 \\
&\leq e^{2B\psi\eta\|\boldsymbol{\Delta}\|_{\max}} + 2.
\end{aligned}
$$

This leads to $\sum_{i\in[n]}\sum_{c\neq y_i} f'(x_{ic}^{\boldsymbol{W}'}) \leq (e^{2B\psi\|\boldsymbol{\Delta W}\|_{\max}} + 2) \sum_{i\in[n]}\sum_{c\neq y_i} f'(x_{ic}^{\boldsymbol{W}})$. Rearrange and using the definition of $\mathcal{G}_{pll}(\boldsymbol{W})$ and relationships in (7), we obtain the desired. $\qquad\square$

**Lemma 31** (Analogue of Lemma 19 for PLL). *Suppose that there exists $\boldsymbol{W} \in \mathbb{R}^{k\times d}$ such that $\mathcal{L}_{pll}(\boldsymbol{W}) \leq \frac{\log 2}{n}$, then we have*

$$
(\boldsymbol{e}_{y_i} - \boldsymbol{e}_c)^T \boldsymbol{W}\boldsymbol{h}_i \geq 0, \quad \text{for all } i \in [n] \text{ and for all } c \in [k] \text{ such that } c \neq y_i. \tag{32}
$$

*Proof.* Denote $x_{ic} = (\boldsymbol{e}_{y_i} - \boldsymbol{e}_c)^T \boldsymbol{W}\boldsymbol{h}_i$. Then, by the assumption, we have for any $i \in [n], c \neq y_i$

$$
\log(1 + e^{-x_{ic}}) \leq \sum_{i\in[n]}\sum_{c\neq y_i} \log(1 + e^{-x_{ic}}) \leq \log(2).
$$

This implies that $x_{ic} \geq 0$ for all $i \in [n], c \neq y_i$. $\qquad\square$

**Lemma 32** (Analogue of Lemma 17 for PLL). *For any $\boldsymbol{W}, \boldsymbol{W}_0 \in \mathbb{R}^{k\times d}$, suppose that $\mathcal{L}(\boldsymbol{W})$ is convex, we have*

$$
|\mathcal{L}_{pll}(\boldsymbol{W}) - \mathcal{L}_{pll}(\boldsymbol{W}_0)| \leq 2B\|\boldsymbol{W} - \boldsymbol{W}_0\|.
$$

*Proof.* This lemma is a direct consequence of Lemma 28 and can be proved in the same way as Lemma 17. $\qquad\square$

Thus, we have proved all the Lemmas for $\mathcal{G}_{pll}(\boldsymbol{W})$ and its relationships to $\mathcal{L}_{pll}(\boldsymbol{W})$ in analogous to those in section C. The proof of NSD ((3)) with PairLogLoss follow the same steps as with cross-entropy loss given in section D.

# G  Implicit Bias of Adam

We consider Adam without the stability constant ($\epsilon$), which performs the following coordinate-wise updates for iteration $t \geq 0$ and initialization $\boldsymbol{W}_0$ [33]:

$$
\mathbf{M}_t = \beta_1 \mathbf{M}_{t-1} + (1 - \beta_1)\nabla\mathcal{L}(\boldsymbol{W}_t) \tag{33a}
$$

$$
\mathbf{V}_t = \beta_2 \mathbf{V}_{t-1} + (1 - \beta_2)\nabla\mathcal{L}(\boldsymbol{W}_t)^2 \tag{33b}
$$

$$
\boldsymbol{W}_{t+1} = \boldsymbol{W}_t - \eta_t \frac{\mathbf{M}_t}{\sqrt{\mathbf{V}_t}}, \tag{33c}
$$

where $\mathbf{M}_t$ and $\mathbf{V}_t$ are the first and second moment estimates of the gradient (with momentum parameters $\beta_1$ and $\beta_2$) respectively. The squaring $(\cdot)^2$ and dividing $\div$ operations are applied *entry-wise*. In the special case of $\beta_1 = \beta_2 = 0$, this simplifies to the SignGD updates.

To study the implicit bias of Adam, we further make the following assumption, which ensures that all entries of the second moment buffer $V_t$ of Adam are bounded away from 0 for all $t \geq 0$. Previously used by Zhang et al. [77] in binary classification, this assumption is satisfied when the data distribution is continuous and non-degenerate. A similar assumption appears in [76].

**Assumption 5.** *The Adam initialization satisfies* $\nabla\mathcal{L}(\boldsymbol{W}_0)[c, j]^2 \geq \omega$ *for all* $c \in [k]$ *and* $j \in [d]$.

The proof of Adam follows the similar approach as NSD. The key challenge is to connect $M_t$ and $V_t$ to a per-class decomposition of $\mathcal{G}(\boldsymbol{W}_t)$. The following Lemma in [77, Lemma 6.5] is useful. It provides an entry-wise bound on the ratio between the first moment and square root of the second moment.

**Lemma 33.** *Considering the Adam updates given in* (5), (33b), *and* (33c), *suppose that* $\beta_1 \leq \beta_2$ *and set* $\alpha = \sqrt{\frac{\beta_2(1-\beta_1)^2}{(1-\beta_2)(\beta_2-\beta_1^2)^2}}$, *then we obtain* $M_t[c, j] \leq \alpha \cdot \sqrt{V_t[c, j]}$ *for all* $c \in [k]$ *and* $j \in [d]$.

The following Lemma bounds the first moment buffer ($M_t$) of Adam in terms of the product of $\eta_t$ with the sume of $\mathcal{G}_c(\boldsymbol{W}_t)$ and $\mathcal{Q}_c(\boldsymbol{W}_t)$. It is used in the proof of Lemma 36.

**Lemma 34.** *Let* $c \in [k]$. *Under the same setting as Theorem 5, there exists a time* $t_0$ *such that the following holds for all* $t \geq t_0$

$$|\mathbf{M}_t[c, j] - (1 - \beta_1^{t+1})\nabla\mathcal{L}(\boldsymbol{W}_t)[c, j]| \leq \alpha_M \eta_t(\mathcal{G}_c(\boldsymbol{W}_t) + \mathcal{Q}_c(\boldsymbol{W}_t)),$$

*where* $j \in [d]$ *and* $\alpha_M$ *is some constant that depends on* $B$ *and* $\beta_1$.

*Proof.* The proof follows the same steps as Lemma 25 with $\|\mathbf{\Delta}\|$ replaced by $\left\|\frac{M}{\sqrt{V}}\right\|_{\max}$. $\square$

The following Lemma bounds the first moment buffer ($V_t$) of Adam in terms of the product of $\eta_t$ and with $\mathcal{G}_c(\boldsymbol{W}_t)$ and $\mathcal{Q}_c(\boldsymbol{W}_t)$. It is used in the proof of Lemma 36.

**Lemma 35.** *Let* $c \in [k]$. *Under the same setting as Theorem 5, there exists a time* $t_0$ *such that the following holds for all* $t \geq t_0$

$$\left| \sqrt{V_t[c, j]} - \sqrt{(1 - \beta_2^{t+1})}|\nabla\mathcal{L}(\boldsymbol{W}_t)[c, j]| \right| \leq \alpha_V \sqrt{\eta_t}(\mathcal{Q}_c(\boldsymbol{W}_t) + \mathcal{G}_c(\boldsymbol{W}_t)),$$

*where* $j \in [d]$, *and* $\alpha_V$ *is some constant that depends on* $B$ *and* $\beta_2$.

*Proof.* Consider any fixed $c \in [k]$ and $j \in [d]$,

$$|V_t[c, j] - (1 - \beta_2^{t+1})\nabla\mathcal{L}(\boldsymbol{W}_t)[c, j]^2| = |\sum_{\tau=0}^{t}(1 - \beta_2)\beta_2^\tau\left(\nabla\mathcal{L}(\boldsymbol{W}_{t-\tau})[c, j]^2 - \nabla\mathcal{L}(\boldsymbol{W}_t)[c, j]^2\right)|$$

$$\leq \sum_{\tau=0}^{t}(1 - \beta_2)\beta_2^\tau \underbrace{|\nabla\mathcal{L}(\boldsymbol{W}_{t-\tau})[c, j]^2 - \nabla\mathcal{L}(\boldsymbol{W}_t)[c, j]^2|}_{\spadesuit}.$$

(34)

For any $\boldsymbol{W} \in \mathbb{R}^{k \times d}$, recall that $-\frac{1}{n}\sum_{i\in[n]}\boldsymbol{e}_c^T\left(\boldsymbol{e}_{y_i} - \mathbb{S}(\boldsymbol{W}\boldsymbol{h}_i)\right)h_{ij}$. Then, we can obtain $\nabla\mathcal{L}(\boldsymbol{W})[c, j]^2 = \frac{1}{n^2}\sum_{i\in[n]}\sum_{p\in[n]}h_{ij}h_{pj}(\delta_{cy_i} - \mathbb{S}_c(\boldsymbol{W}\boldsymbol{h}_i))(\delta_{cy_p} - \mathbb{S}_c(\boldsymbol{W}\boldsymbol{h}_p))$ where $\delta_{cy} = 1$ if and only if $c = y$. Next, we define the function $f_{c,i,p}$ to be $f_{c,i,p}(\boldsymbol{W}) := (\delta_{cy_i} - \mathbb{S}_c(\boldsymbol{W}\boldsymbol{h}_i))(\delta_{cy_p} - \mathbb{S}_c(\boldsymbol{W}\boldsymbol{h}_p))$. Then, we have

$$|f_{c,i,p}(\boldsymbol{W}_{t-\tau}) - f_{c,i,p}(\boldsymbol{W}_t)| = \delta_{cy_i}\left(\mathbb{S}_c(\boldsymbol{W}_t\boldsymbol{h}_p) - \mathbb{S}_c(\boldsymbol{W}_{t-\tau}\boldsymbol{h}_p)\right) + \delta_{cy_p}\left(\mathbb{S}_c(\boldsymbol{W}_t\boldsymbol{h}_i) - \mathbb{S}_c(\boldsymbol{W}_{t-\tau}\boldsymbol{h}_i)\right)$$

$$+ \left(\mathbb{S}_c(\boldsymbol{W}_{t-\tau}\boldsymbol{h}_i)\mathbb{S}_c(\boldsymbol{W}_{t-\tau}\boldsymbol{h}_p) - \mathbb{S}_c(\boldsymbol{W}_t\boldsymbol{h}_i)\mathbb{S}_c(\boldsymbol{W}_t\boldsymbol{h}_p)\right)$$

We can substitute this result into ♠ to obtain

$$
\spadesuit = |\frac{1}{n^2} \sum_{i\in[n]} \sum_{p\in[n]} h_{ij} h_{pj} (f_{c,i,p}(\boldsymbol{W}_{t-\tau}) - f_{c,i,p}(\boldsymbol{W}_t))|
$$

$$
\leq \frac{B^2}{n^2} \sum_{i\in[n]} \sum_{p\in[n]} |f_{c,i,p}(\boldsymbol{W}_{t-\tau}) - f_{c,i,p}(\boldsymbol{W}_t)|
$$

$$
= B^2 \frac{1}{n^2} \underbrace{\sum_{i\in[n],y_i\neq c} \sum_{p\in[n],y_p\neq c} |f_{c,i,p}(\boldsymbol{W}_{t-\tau}) - f_{c,i,p}(\boldsymbol{W}_t)|}_{\spadesuit_1}
$$

$$
+ B^2 \frac{1}{n^2} \underbrace{\sum_{i\in[n],y_i\neq c} \sum_{p\in[n],y_p= c} |f_{c,i,p}(\boldsymbol{W}_{t-\tau}) - f_{c,i,p}(\boldsymbol{W}_t)|}_{\spadesuit_2}
$$

$$
+ B^2 \frac{1}{n^2} \underbrace{\sum_{i\in[n],y_i= c} \sum_{p\in[n],y_p\neq c} |f_{c,i,p}(\boldsymbol{W}_{t-\tau}) - f_{c,i,p}(\boldsymbol{W}_t)|}_{\spadesuit_3}
$$

$$
+ B^2 \frac{1}{n^2} \underbrace{\sum_{i\in[n],y_i= c} \sum_{p\in[n],y_p= c} |f_{c,i,p}(\boldsymbol{W}_{t-\tau}) - f_{c,i,p}(\boldsymbol{W}_t)|}_{\spadesuit_4}
$$

We deal with the 4 terms $\spadesuit_1$, $\spadesuit_2$, $\spadesuit_3$, and $\spadesuit_4$ separately. Starting with the first term, we have

$$
\spadesuit_1 = \frac{1}{n^2} \sum_{i\in[n],y_i\neq c} \sum_{p\in[n],y_p\neq c} |\mathbb{S}_c(\boldsymbol{W}_{t-\tau}\boldsymbol{h}_i)\mathbb{S}_c(\boldsymbol{W}_{t-\tau}\boldsymbol{h}_p) - \mathbb{S}_c(\boldsymbol{W}_t\boldsymbol{h}_i)\mathbb{S}_c(\boldsymbol{W}_t\boldsymbol{h}_p)|
$$

$$
= \frac{1}{n^2} \sum_{i\in[n],y_i\neq c} \sum_{p\in[n],y_p\neq c} \mathbb{S}_c(\boldsymbol{W}_t\boldsymbol{h}_i)\mathbb{S}_c(\boldsymbol{W}_t\boldsymbol{h}_p)|\frac{\mathbb{S}_c(\boldsymbol{W}_{t-\tau}\boldsymbol{h}_i)\mathbb{S}_c(\boldsymbol{W}_{t-\tau}\boldsymbol{h}_p)}{\mathbb{S}_c(\boldsymbol{W}_t\boldsymbol{h}_i)\mathbb{S}_c(\boldsymbol{W}_t\boldsymbol{h}_p)} - 1|
$$

$$
\overset{(a)}{\leq} \frac{1}{n^2} \sum_{i\in[n],y_i\neq c} \sum_{p\in[n],y_p\neq c} \mathbb{S}_c(\boldsymbol{W}_t\boldsymbol{h}_i)\mathbb{S}_c(\boldsymbol{W}_t\boldsymbol{h}_p)\big(e^{2\big(\|(\boldsymbol{W}_{t-\tau}-\boldsymbol{W}_t)\boldsymbol{h}_i\|_\infty + \|(\boldsymbol{W}_{t-\tau}-\boldsymbol{W}_t)\boldsymbol{h}_p\|_\infty\big)} - 1\big)
$$

$$
\overset{(b)}{\leq} \frac{1}{n^2} \sum_{i\in[n],y_i\neq c} \sum_{p\in[n],y_p\neq c} \mathbb{S}_c(\boldsymbol{W}_t\boldsymbol{h}_i)\mathbb{S}_c(\boldsymbol{W}_t\boldsymbol{h}_p)\big(e^{4B\|\boldsymbol{W}_{t-\tau}-\boldsymbol{W}_t\|_{\max}} - 1\big)
$$

$$
\overset{(c)}{\leq} \big(e^{4B\sum_{s=1}^\tau \eta_{t-s}\left\|\frac{\boldsymbol{M}_{t-s}}{\sqrt{\boldsymbol{V}_{t-s}}}\right\|_{\max}} - 1\big) \frac{1}{n^2} \sum_{i\in[n],y_i\neq c} \sum_{p\in[n],y_p\neq c} \mathbb{S}_c(\boldsymbol{W}_t\boldsymbol{h}_i)\mathbb{S}_c(\boldsymbol{W}_t\boldsymbol{h}_p)
$$

$$
\overset{(d)}{\leq} \big(e^{4B\alpha\sum_{s=1}^\tau \eta_{t-s}} - 1\big)\mathcal{Q}_c(\boldsymbol{W}_t)^2,
$$

where (a) is by Lemma 15, (b) is by $\|\boldsymbol{h}_i\|_1 \leq B$ for all $i \in [n]$, (c) is by (33c) and the triangle inequality, and (d) is by Lemma 33 and the definition of $\mathcal{G}(\boldsymbol{W}_t)$. For the second term, we have

$$
\spadesuit_2 = \frac{1}{n^2} \sum_{i\in[n],y_i\neq c} \sum_{p\in[n],y_p= c} |\big(\mathbb{S}_c(\boldsymbol{W}_t\boldsymbol{h}_i) - \mathbb{S}_c(\boldsymbol{W}_{t-\tau}\boldsymbol{h}_i)\big)
$$

$$
+ \big(\mathbb{S}_c(\boldsymbol{W}_{t-\tau}\boldsymbol{h}_i)\mathbb{S}_c(\boldsymbol{W}_{t-\tau}\boldsymbol{h}_p) - \mathbb{S}_c(\boldsymbol{W}_t\boldsymbol{h}_i)\mathbb{S}_c(\boldsymbol{W}_t\boldsymbol{h}_p)\big)|
$$

$$
= \frac{1}{n^2} \sum_{i\in[n],y_i\neq c} \sum_{p\in[n],y_p= c} |\mathbb{S}_c(\boldsymbol{W}_t\boldsymbol{h}_i)\big(1 - \mathbb{S}_c(\boldsymbol{W}_t\boldsymbol{h}_p)\big) - \big(1 - \mathbb{S}_c(\boldsymbol{W}_{t-\tau}\boldsymbol{h}_p)\big)\mathbb{S}_c(\boldsymbol{W}_{t-\tau}\boldsymbol{h}_i)|
$$

$$
= \frac{1}{n^2} \sum_{i\in[n],y_i\neq c} \sum_{p\in[n],y_p= c} \mathbb{S}_c(\boldsymbol{W}_t\boldsymbol{h}_i)\big(1 - \mathbb{S}_c(\boldsymbol{W}_t\boldsymbol{h}_p)\big)|1 - \frac{\big(1 - \mathbb{S}_c(\boldsymbol{W}_{t-\tau}\boldsymbol{h}_p)\big)\mathbb{S}_c(\boldsymbol{W}_{t-\tau}\boldsymbol{h}_i)}{\big(1 - \mathbb{S}_c(\boldsymbol{W}_t\boldsymbol{h}_p)\big)\mathbb{S}_c(\boldsymbol{W}_t\boldsymbol{h}_i)}|
$$

$$
\leq \big(e^{4B\alpha\sum_{s=1}^\tau \eta_{t-s}} - 1\big)\mathcal{Q}_c(\boldsymbol{W}_t)\mathcal{G}_c(\boldsymbol{W}_t),
$$

where the last inequality is by Lemma 15 and the same steps taken for $\spadesuit_1$. The third term can be derived similarly as the second term and we can obtain the same bound as follows:

$$\spadesuit_3 = \frac{1}{n^2} \sum_{i\in[n], y_i=c} \sum_{p\in[n], y_p\neq c} \mathbb{S}_c(\boldsymbol{W}_t \boldsymbol{h}_p)\big(1 - \mathbb{S}_c(\boldsymbol{W}_t \boldsymbol{h}_i)\big)|1 - \frac{\big(1 - \mathbb{S}_c(\boldsymbol{W}_{t-\tau}\boldsymbol{h}_p)\big)\mathbb{S}_c(\boldsymbol{W}_{t-\tau}\boldsymbol{h}_i)}{\big(1 - \mathbb{S}_c(\boldsymbol{W}_t\boldsymbol{h}_i)\big)\mathbb{S}_c(\boldsymbol{W}_t\boldsymbol{h}_p)}|$$

$$\leq \big(e^{4B\alpha \sum_{s=1}^{\tau} \eta_{t-s}} - 1\big)\mathcal{Q}_c(\boldsymbol{W}_t)\mathcal{G}_c(\boldsymbol{W}_t).$$

For the fourth term, we obtain:

$$\spadesuit_4 = \frac{1}{n^2} \sum_{i\in[n], y_i=c} \sum_{p\in[n], y_p=c} |\big(1 - \mathbb{S}_c(\boldsymbol{W}_{t-\tau}\boldsymbol{h}_i)\big)\big(1 - \mathbb{S}_c(\boldsymbol{W}_{t-\tau}\boldsymbol{h}_p)\big) - \big(1 - \mathbb{S}_c(\boldsymbol{W}_t\boldsymbol{h}_i)\big)\big(1 - \mathbb{S}_c(\boldsymbol{W}_t\boldsymbol{h}_p)\big)|$$

$$= \frac{1}{n^2} \sum_{i\in[n], y_i=c} \sum_{p\in[n], y_p=c} \big(1 - \mathbb{S}_c(\boldsymbol{W}_t\boldsymbol{h}_i)\big)\big(1 - \mathbb{S}_c(\boldsymbol{W}_t\boldsymbol{h}_p)\big)|\frac{\big(1 - \mathbb{S}_c(\boldsymbol{W}_{t-\tau}\boldsymbol{h}_i)\big)\big(1 - \mathbb{S}_c(\boldsymbol{W}_{t-\tau}\boldsymbol{h}_p)\big)}{\big(1 - \mathbb{S}_c(\boldsymbol{W}_t\boldsymbol{h}_i)\big)\big(1 - \mathbb{S}_c(\boldsymbol{W}_t\boldsymbol{h}_p)\big)} - 1|$$

$$\leq \big(e^{4B\alpha \sum_{s=1}^{\tau} \eta_{t-s}} - 1\big)\mathcal{G}_c(\boldsymbol{W}_t)^2,$$

where the last inequality is by Lemma 15 and the same steps taken for $\spadesuit_1$. We combine the bounds for $\spadesuit_1$, $\spadesuit_2$, $\spadesuit_3$, and $\spadesuit_4$ to obtain: $\spadesuit \leq 4B^2\big(e^{4B\alpha \sum_{s=1}^{\tau} \eta_{t-s}} - 1\big)(\mathcal{G}_c(\boldsymbol{W}_t) + \mathcal{Q}_c(\boldsymbol{W}_t))^2$. Then, we substitute this into (34) to obtain:

$$|\boldsymbol{V}_t[c,j] - (1 - \beta_2^{t+1})\nabla\mathcal{L}(\boldsymbol{W}_t)[c,j]^2| \leq B^2(1-\beta_2)(\mathcal{Q}_c(\boldsymbol{W}_t) + \mathcal{G}_c(\boldsymbol{W}_t))^2 \sum_{\tau=0}^{t} \beta_2^{\tau}\big(e^{4\alpha B \sum_{s=1}^{\tau} \eta_{t-s}} - 1\big)$$

$$\leq B^2(1-\beta_2)c_2\eta_t(\mathcal{Q}_c(\boldsymbol{W}_t) + \mathcal{G}_c(\boldsymbol{W}_t))^2,$$

where the last inequality is by the Assumption 3. The final result follows from the fact that $|p - q|^2 \leq |p^2 - q^2|$ when both $p$ and $q$ are positive. $\qquad\square$

The following Lemma bounds the term $\big| \langle \nabla\mathcal{L}(\boldsymbol{W}_t), \frac{\boldsymbol{M}_t}{\sqrt{\boldsymbol{V}_t}} - \frac{\nabla\mathcal{L}(\boldsymbol{W}_t)}{|\nabla\mathcal{L}(\boldsymbol{W}_t)|}\rangle \big|$ using $\mathcal{G}(\boldsymbol{W}_t)$. It is used in Lemma 37 to show the decrease in the risk. The proof is similar to that of Zhang et al. [77, Lemma A.3], but here we need to carefully track the index $c \in [k]$ using both $\mathcal{G}_c(\boldsymbol{W})$ and $\mathcal{Q}_c(\boldsymbol{W})$ to avoid $k$ dependence. The final result crucially relies on the decomposition $\mathcal{G}(\boldsymbol{W}_t) = \sum_{c\in[k]} \mathcal{T}_c(\boldsymbol{W}_t) = \sum_{c\in[k]} \mathcal{Q}_c(\boldsymbol{W}_t)$.

**Lemma 36.** *Under the same setting as Theorem 5, we have*

$$\underbrace{\big| \langle \nabla\mathcal{L}(\boldsymbol{W}_t), \frac{\boldsymbol{M}_t}{\sqrt{\boldsymbol{V}_t}} - \frac{\nabla\mathcal{L}(\boldsymbol{W}_t)}{|\nabla\mathcal{L}(\boldsymbol{W}_t)|}\rangle \big|}_{\clubsuit} \leq 4\sqrt{\frac{\beta_1^{t+1}}{1-\beta_2^{t+1}}}\|\nabla\mathcal{L}(\boldsymbol{W}_t)\|_{\text{sum}}+$$

$$\frac{2d}{\sqrt{1-\beta_2}}\big(\frac{6\alpha_V}{\sqrt{1-\beta_2^{t+1}}}\sqrt{\eta_t} + 3\alpha_M\eta_t\big)\mathcal{G}(\boldsymbol{W}_t).$$

*Proof.* For simplicity, we drop the subscripts $t$. Denote $\mathcal{T}_c(\boldsymbol{W}) := \mathcal{G}_c(\boldsymbol{W}) + \mathcal{Q}_c(\boldsymbol{W})$. Then, by Lemmas 34 and 35, we have for any $c \in [k]$ and $j \in [d]$:

$$\boldsymbol{M}[c,j] = (1 - \beta_1^{t+1})\nabla\mathcal{L}(\boldsymbol{W})[c,j] + \alpha_M\eta_t\mathcal{T}_c(\boldsymbol{W})\epsilon_{m,c,j}, \tag{35}$$

$$\sqrt{\boldsymbol{V}[c,j]} = \sqrt{1 - \beta_2^{t+1}}|\nabla\mathcal{L}(\boldsymbol{W})[c,j]| + \alpha_V\sqrt{\eta_t}\mathcal{T}_c(\boldsymbol{W})\epsilon_{v,c,j}, \tag{36}$$

where $|\epsilon_{m,c,j}| \leq 1$ and $|\epsilon_{v,c,j}| \leq 1$ are some residual terms. We denote $\psi_{c,j} := \nabla\mathcal{L}(\boldsymbol{W})[c,j](\frac{\boldsymbol{M}[c,j]}{\sqrt{\boldsymbol{V}[c,j]}} - \frac{\nabla\mathcal{L}(\boldsymbol{W})[c,j]}{|\nabla\mathcal{L}(\boldsymbol{W})[c,j]|})$, the set of index $E_{c,j} := \{j \in [d] \mid \sqrt{1-\beta_2^{t+1}}|\nabla\mathcal{L}(\boldsymbol{W})[c,j]| \geq 2\alpha_V\sqrt{\eta_t}\mathcal{T}_c(\boldsymbol{W})|\epsilon_{v,c,j}|\}$, and its complement $E_{c,j}^c = [d]\backslash E_{c,j}$. The

goal is to bound $|\psi_{c,j}|$ when $j \in E_{c,j}^c$ or $j \in E_{c,j}$ using $\mathcal{T}_c(\boldsymbol{W})$. We start with the indices in $E_{c,j}^c$:

$$\sum_{j \in E_{c,j}^c} |\psi_{c,j}| \leq \sum_{j \in E_{c,j}^c} |\nabla\mathcal{L}(\boldsymbol{W})[c,j]|\left(\frac{|\boldsymbol{M}[c,j]|}{\sqrt{\boldsymbol{V}[c,j]}} + 1\right)$$

$$\overset{(a)}{\leq} \sum_{j \in E_{c,j}^c} |\nabla\mathcal{L}(\boldsymbol{W})[c,j]|\left(\frac{(1-\beta_1^{t+1})|\nabla\mathcal{L}(\boldsymbol{W})[c,j]| + \alpha_M\eta_t\mathcal{T}_c(\boldsymbol{W})}{\sqrt{1-\beta_2}|\nabla\mathcal{L}(\boldsymbol{W})[c,j]|} + 1\right)$$

$$\overset{(b)}{\leq} \sum_{j \in E_{c,j}^c} \left(\frac{1-\beta_1^{t+1}}{\sqrt{1-\beta_2}} + 1\right)\frac{2\alpha_V\sqrt{\eta_t}\mathcal{T}_c(\boldsymbol{W})}{\sqrt{1-\beta_2^{t+1}}} + \frac{\alpha_M\eta_t\mathcal{T}_c(\boldsymbol{W})}{\sqrt{1-\beta_2}}$$

$$\leq \frac{d}{\sqrt{1-\beta_2}}\left(\frac{4\alpha_V}{\sqrt{1-\beta_2^{t+1}}}\sqrt{\eta_t} + \alpha_M\eta_t\right)\mathcal{T}_c(\boldsymbol{W}),$$

where (a) is by (35), $|\epsilon_{m,c,j}| \leq 1$, and $\boldsymbol{V}[c,j] \geq (1-\beta_2)\nabla\mathcal{L}(\boldsymbol{W})[c,j]^2$; and (b) is by $j \in E_{c,j}^c$ s.t. $|\nabla\mathcal{L}(\boldsymbol{W})[c,j]| \leq \frac{2\alpha_V\sqrt{\eta_t}\mathcal{T}_c(\boldsymbol{W})}{\sqrt{1-\beta_2^{t+1}}}$. Next, we focus on the indices $j \in E_{c,j}$. In this case, we have

$$\psi_{c,j} = \nabla\mathcal{L}(\boldsymbol{W})[c,j]\left(\frac{\boldsymbol{M}[c,j]}{\sqrt{1-\beta_2^{t+1}}|\nabla\mathcal{L}(\boldsymbol{W})[c,j]| + \alpha_V\sqrt{\eta_t}\mathcal{T}_c(\boldsymbol{W})\epsilon_{v,c,j}} - \frac{\nabla\mathcal{L}(\boldsymbol{W})[c,j]}{|\nabla\mathcal{L}(\boldsymbol{W})[c,j]|}\right)$$

$$= \nabla\mathcal{L}(\boldsymbol{W})[c,j]\underbrace{\frac{\boldsymbol{M}[c,j]|\nabla\mathcal{L}(\boldsymbol{W})[c,j]| - \left(\sqrt{1-\beta_2^{t+1}}|\nabla\mathcal{L}(\boldsymbol{W})[c,j]| + \alpha_V\sqrt{\eta_t}\mathcal{T}_c(\boldsymbol{W})\epsilon_{v,c,j}\right)\nabla\mathcal{L}(\boldsymbol{W})[c,j]}{\left(\sqrt{1-\beta_2^{t+1}}|\nabla\mathcal{L}(\boldsymbol{W})[c,j]| + \alpha_V\sqrt{\eta_t}\mathcal{T}_c(\boldsymbol{W})\epsilon_{v,c,j}\right)|\nabla\mathcal{L}(\boldsymbol{W})[c,j]|}}_{\frac{\spadesuit_1}{\spadesuit_2}},$$

where

$$\left|\spadesuit_1\right| = \Big|\left(1-\beta_1^{t+1} - \sqrt{1-\beta_2^{t+1}}\right)\nabla\mathcal{L}(\boldsymbol{W})[c,j]|\nabla\mathcal{L}(\boldsymbol{W})[c,j]| +$$

$$\alpha_M\eta_t\mathcal{T}_c(\boldsymbol{W})\epsilon_{m,c,j}|\nabla\mathcal{L}(\boldsymbol{W})[c,j]| - \alpha_V\sqrt{\eta_t}\mathcal{T}_c(\boldsymbol{W})\epsilon_{v,c,j}\nabla\mathcal{L}(\boldsymbol{W})[c,j]\Big|$$

$$\overset{(c)}{\leq} \left|1-\beta_1^{t+1} - \sqrt{1-\beta_2^{t+1}}\right||\nabla\mathcal{L}(\boldsymbol{W})[c,j]|^3 + (\alpha_M\eta_t + \alpha_V\sqrt{\eta_t})\mathcal{T}_c(\boldsymbol{W})|\nabla\mathcal{L}(\boldsymbol{W})[c,j]|^2,$$

and

$$\left|\spadesuit_2\right| = \spadesuit_2 \overset{(d)}{\geq} \frac{1}{2}\sqrt{1-\beta_2^{t+1}}|\nabla\mathcal{L}(\boldsymbol{W})[c,j]|^2.$$

Inequality (c) is by $|\epsilon_{m,c,j}| \leq 1$ and $|\epsilon_{v,c,j}| \leq 1$, and (d) is by $\alpha_V\sqrt{\eta_t}\mathcal{T}_c(\boldsymbol{W})\epsilon_{v,c,j} \geq -\frac{1}{2}\sqrt{1-\beta_2^{t+1}}|\nabla\mathcal{L}(\boldsymbol{W})[c,j]|$ for any $j \in E_{c,j}$. Putting these two pieces together, we obtain

$$\sum_{j \in E_{c,j}} |\psi_{c,j}| \leq \sum_{j \in E_{c,j}} \frac{|1-\beta_1^{t+1} - \sqrt{1-\beta_2^{t+1}}||\nabla\mathcal{L}(\boldsymbol{W})[c,j]|^3 + (\alpha_M\eta_t + \alpha_V\sqrt{\eta_t})\mathcal{T}_c(\boldsymbol{W})|\nabla\mathcal{L}(\boldsymbol{W})[c,j]|^2}{\frac{1}{2}\sqrt{1-\beta_2^{t+1}}|\nabla\mathcal{L}(\boldsymbol{W})[c,j]|^2}$$

$$\overset{(e)}{\leq} \left(\sum_{j \in E_{c,j}} 4\sqrt{\frac{\beta_1^{t+1}}{1-\beta_2^{t+1}}}|\nabla\mathcal{L}(\boldsymbol{W})[c,j]|\right) + d\left(\frac{2\alpha_V}{\sqrt{1-\beta_2^{t+1}}}\sqrt{\eta_t} + \frac{2\alpha_M}{\sqrt{1-\beta_2^{t+1}}}\eta_t\right)\mathcal{T}_c(\boldsymbol{W})$$

$$\leq 4\sqrt{\frac{\beta_1^{t+1}}{1-\beta_2^{t+1}}}\|\nabla\mathcal{L}(\boldsymbol{W})[c,:]\|_{\text{sum}} + \frac{d}{\sqrt{1-\beta_2}}\left(\frac{2\alpha_V}{\sqrt{1-\beta_2^{t+1}}}\sqrt{\eta_t} + 2\alpha_M\eta_t\right)\mathcal{T}_c(\boldsymbol{W}),$$

where (e) is by $\sqrt{a} \leq \sqrt{a-b} + \sqrt{b}$ implying $1 - \sqrt{1-\beta_2^{t+1}} \leq \beta_1^{\frac{t+1}{2}}$, and $\nabla\mathcal{L}(\boldsymbol{W})[c,:]$ denotes the $c$th row of $\nabla\mathcal{L}(\boldsymbol{W})$. Finally, we note that $|\langle\nabla\mathcal{L}(\boldsymbol{W}), \frac{\boldsymbol{M}}{\sqrt{\boldsymbol{V}}} - \frac{\nabla\mathcal{L}(\boldsymbol{W})}{|\nabla\mathcal{L}(\boldsymbol{W})|}\rangle| = |\sum_{(c,j)} \psi_{c,j}| \leq$

$\sum_{(c,j)} |\psi_{c,j}|$. Then, we obtain

$$\sum_{c,j} |\psi_{c,j}| = \sum_{c\in[k]} \Big( \sum_{j\in E_{c,j}^c} |\psi_{c,j}| + \sum_{j\in E_{c,j}} |\psi_{c,j}| \Big)$$

$$= \sum_{c\in[k]} 4\sqrt{\frac{\beta_1^{t+1}}{1-\beta_2^{t+1}}} \|\nabla\mathcal{L}(\boldsymbol{W})[c,:]\|_{\mathrm{sum}} + \sum_{c\in[k]} \frac{d}{\sqrt{1-\beta_2}}\Big(\frac{2\alpha_V}{\sqrt{1-\beta_2^{t+1}}}\sqrt{\eta_t} + 2\alpha_M\eta_t\Big)\mathcal{T}_c(\boldsymbol{W})$$

$$\overset{(f)}{=} 4\sqrt{\frac{\beta_1^{t+1}}{1-\beta_2^{t+1}}} \|\nabla\mathcal{L}(\boldsymbol{W})\|_{\mathrm{sum}} + \frac{2d}{\sqrt{1-\beta_2}}\Big(\frac{2\alpha_V}{\sqrt{1-\beta_2^{t+1}}}\sqrt{\eta_t} + 2\alpha_M\eta_t\Big)\mathcal{G}(\boldsymbol{W}),$$

where (f) is by $\sum_{c\in[k]} \mathcal{T}_c(\boldsymbol{W}) = \sum_{c\in[k]} \mathcal{Q}_c(\boldsymbol{W}) + \mathcal{G}_c(\boldsymbol{W}) = 2\mathcal{G}(\boldsymbol{W})$. $\qquad\square$

**Lemma 37** (Adam Descent). *Under the same setting as Theorem 5, set* $t_A := \frac{2\log(\frac{\sqrt{1-\beta_2}}{4})}{\log(\beta_1)}$, *then we have for all* $t \geq t_A$

$$\mathcal{L}(\boldsymbol{W}_{t+1}) \leq \mathcal{L}(\boldsymbol{W}_t) - \eta_t\gamma\big(1 - \alpha_{a_1}\beta_1^{t/2} - \alpha_{a_2}d\eta_t^{\frac{1}{2}} - \alpha_{a_3}d\eta_t\big)\mathcal{G}(\boldsymbol{W}_t),$$

*where* $\alpha_{a_1}$, $\alpha_{a_2}$, *and* $\alpha_{a_3}$ *are some constants that depend on* $B$, $\gamma$, $\beta_1$, *and* $\beta_2$.

*Proof.* We follow the same notations and strategy of Lemma 21, and recall the definitions $\spadesuit_t = \langle\nabla\mathcal{L}(\boldsymbol{W}_t),\boldsymbol{\Delta}_t\rangle$ and $\clubsuit_t = \boldsymbol{h}_i^\top\boldsymbol{\Delta}_t^\top\big(\mathrm{diag}(\mathbb{S}(\boldsymbol{W}_{t,t+1,\gamma}\boldsymbol{h}_i)) - \mathbb{S}(\boldsymbol{W}_{t,t+1,\zeta^*}\boldsymbol{h}_i)\mathbb{S}(\boldsymbol{W}_{t,t+1,\zeta^*}\boldsymbol{h}_i)^\top\big)\boldsymbol{\Delta}_t\boldsymbol{h}_i$. In the case of Adam, we have $\boldsymbol{\Delta}_t = \frac{\boldsymbol{M}_t}{\sqrt{\boldsymbol{V}_t}}$. We bound $\spadesuit_t$ and $\clubsuit_t$ separately. Starting with $\spadesuit_t$, we have for all $t \geq t_A$

$$\spadesuit_t = -\eta_t\langle\nabla\mathcal{L}(\boldsymbol{W}_t), \frac{\boldsymbol{M}_t}{\sqrt{\boldsymbol{V}_t}}\rangle$$

$$= -\eta_t\big(\langle\nabla\mathcal{L}(\boldsymbol{W}_t), \frac{\boldsymbol{M}_t}{\sqrt{\boldsymbol{V}_t}} - \frac{\nabla\mathcal{L}(\boldsymbol{W}_t)}{|\nabla\mathcal{L}(\boldsymbol{W}_t)|}\rangle + \langle\nabla\mathcal{L}(\boldsymbol{W}_t), \frac{\nabla\mathcal{L}(\boldsymbol{W}_t)}{|\nabla\mathcal{L}(\boldsymbol{W}_t)|}\rangle\big)$$

$$\leq -\eta_t\|\nabla\mathcal{L}(\boldsymbol{W}_t)\|_{\mathrm{sum}} + \eta_t\big|\langle\nabla\mathcal{L}(\boldsymbol{W}_t), \frac{\boldsymbol{M}_t}{\sqrt{\boldsymbol{V}_t}} - \frac{\nabla\mathcal{L}(\boldsymbol{W}_t)}{|\nabla\mathcal{L}(\boldsymbol{W}_t)|}\rangle\big|$$

$$\overset{(a)}{\leq} -\eta_t\big(1 - 4\sqrt{\frac{\beta_1^{t+1}}{1-\beta_2^{t+1}}}\big)\|\nabla\mathcal{L}(\boldsymbol{W}_t)\|_{\mathrm{sum}} + \frac{2d}{\sqrt{1-\beta_2}}\Big(\frac{6\alpha_V}{\sqrt{1-\beta_2^{t+1}}}\eta_t^{\frac{3}{2}} + 3\alpha_M\eta_t^2\Big)\mathcal{G}(\boldsymbol{W}_t)$$

$$\leq -\eta_t\big(1 - 4\frac{\beta_1^{\frac{t}{2}}}{\sqrt{1-\beta_2}}\big)\|\nabla\mathcal{L}(\boldsymbol{W}_t)\|_{\mathrm{sum}} + \frac{12\alpha_V}{1-\beta_2}d\eta_t^{3/2}\mathcal{G}(\boldsymbol{W}_t) + \frac{6\alpha_M}{\sqrt{1-\beta_2}}d\eta_t^2\mathcal{G}(\boldsymbol{W}_t)$$

$$\overset{(b)}{\leq} -\eta_t\gamma\big(1 - 4\frac{\beta_1^{\frac{t}{2}}}{\sqrt{1-\beta_2}}\big)\mathcal{G}(\boldsymbol{W}_t) + \frac{12\alpha_V}{1-\beta_2}d\eta_t^{3/2}\mathcal{G}(\boldsymbol{W}_t) + \frac{6\alpha_M}{\sqrt{1-\beta_2}}d\eta_t^2\mathcal{G}(\boldsymbol{W}_t),$$

where (a) is by Lemma 36, and (b) is by Lemma 16. For $\clubsuit_t$, we apply Lemma 12 to obtain

$$\clubsuit_t \leq 4\|\boldsymbol{\Delta}_t\boldsymbol{h}_i\|_\infty^2(1 - \mathbb{S}_{y_i}(\boldsymbol{W}_{t,t+1,\zeta^*}\boldsymbol{h}_i)) \leq 4\eta_t^2\alpha^2B^2(1 - \mathbb{S}_{y_i}(\boldsymbol{W}_{t,t+1,\zeta^*}\boldsymbol{h}_i)),$$

where in the second inequality we have used $\|\boldsymbol{\Delta}_t\boldsymbol{h}_i\|_\infty \leq \|\boldsymbol{\Delta}_t\|_{\mathrm{max}}\|\boldsymbol{h}_i\|_1$, $\|\boldsymbol{h}_i\|_1 \leq B$, and $\|\boldsymbol{\Delta}_t\|_{\mathrm{max}} = \eta_t\big\|\frac{\boldsymbol{M}_t}{\sqrt{\boldsymbol{V}_t}}\big\|_{\mathrm{max}} \leq \eta_t\alpha$ by Lemma 33 given $t \geq t_A$ implying that $1 \geq 4\frac{\beta_1^{\frac{t}{2}}}{\sqrt{1-\beta_2}}$. Combing this with Lemma 12, we obtain

$$\frac{1}{2n}\sum_{i\in[n]} \boldsymbol{h}_i^\top\boldsymbol{\Delta}_t^\top\big(\mathrm{diag}(\mathbb{S}(\boldsymbol{W}_{t,t+1,\gamma}\boldsymbol{h}_i)) - \mathbb{S}(\boldsymbol{W}_{t,t+1,\zeta^*}\boldsymbol{h}_i)\mathbb{S}(\boldsymbol{W}_{t,t+1,\zeta^*}\boldsymbol{h}_i)^\top\big)\boldsymbol{\Delta}_t\boldsymbol{h}_i$$

$$\leq \frac{1}{2n}\sum_{i\in[n]} 4\eta_t^2\alpha^2B^2(1 - \mathbb{S}_{y_i}(\boldsymbol{W}_{t,t+1,\zeta^*}\boldsymbol{h}_i)) \leq 2\alpha^2\eta_t^2B^2e^{2B\eta_0}\mathcal{G}(\boldsymbol{W}_t),$$

where the derivation of the second inequality can be found in the derivation of 22. Putting everything together, we obtain

$$\mathcal{L}(\boldsymbol{W}_{t+1}) \leq \mathcal{L}(\boldsymbol{W}_t) - \eta_t \gamma \mathcal{G}(\boldsymbol{W}_t) + 4 \frac{\beta_1^{\frac{t}{2}}}{\sqrt{1-\beta_2}} \gamma \eta_t \mathcal{G}(\boldsymbol{W}_t) + \frac{12\alpha_V}{1-\beta_2} d\eta_t^{3/2} \mathcal{G}(\boldsymbol{W}_t) +$$

$$\left(\frac{6\alpha_M}{\sqrt{1-\beta_2}} + 2\alpha^2 B^2 e^{2B\eta_0}\right) d\eta_t^2 \mathcal{G}(\boldsymbol{W}_t)$$

$$= \mathcal{L}(\boldsymbol{W}_t) - \eta_t \gamma \left(1 - \alpha_{a_1}\beta_1^{t/2} - \alpha_{a_2} d\eta_t^{\frac{1}{2}} - \alpha_{a_3} d\eta_t\right) \mathcal{G}(\boldsymbol{W}_t),$$

where we have defined $\alpha_{a_1} := \frac{4}{\sqrt{1-\beta_2}}$, $\alpha_{a_2} := \frac{12\alpha_V}{\gamma(1-\beta_2)}$, and $\alpha_{a_3} := \frac{6\alpha_M}{\gamma\sqrt{1-\beta_2}} + \frac{2\alpha^2 B^2 e^{2B\eta_0}}{\gamma}$. $\qquad\square$

Built upon Lemma 37, we can further lower bound the unnormalized margin of Adam iterates for a sufficiently large $t$. The proof is similar to that of NSD (i.e., Lemma 22), which crucially depends on the separability condition obtained after achieving a low loss (Lemma 19). The time $\tilde{t}_A$ will be specified in the proof of Theorem 5.

**Lemma 38** (Adam Unnormalized Margin). *Under the same setting as Theorem 5, suppose that there exist $\tilde{t}$ such that $\mathcal{L}(\boldsymbol{W}_t) \leq \frac{\log 2}{n}$ for all $t > \tilde{t}$, then we have for all $t \geq \tilde{t}_A := \max\{t_A, \tilde{t}\}$*

$$\min_{i \in [n], c \neq y_i} (\boldsymbol{e}_{y_i} - \boldsymbol{e}_c)^T \boldsymbol{W}_t \boldsymbol{h}_i \geq \gamma \sum_{s=\tilde{t}_A}^{t-1} \eta_s \frac{\mathcal{G}(\boldsymbol{W}_s)}{\mathcal{L}(\boldsymbol{W}_s)} - \alpha_{a_5} d \sum_{s=\tilde{t}_A}^{t-1} \eta_s^{\frac{3}{2}} - \alpha_{a_6} d \sum_{s=\tilde{t}_A}^{t-1} \eta_s^2 - \alpha_{a_7},$$

*where $t_A = \frac{2\log(\frac{\sqrt{1-\beta_2}}{4})}{\log(\beta_1)}$, and $\alpha_{a_5}$, $\alpha_{a_6}$, and $\alpha_{a_7}$ are some constants that depend on $B$, $\beta_1$, and $\beta_2$.*

*Proof.* We denote $\alpha_{a_4} := \frac{4}{\sqrt{1-\beta_2}}$, $\alpha_{a_5} := \frac{12\alpha_V}{1-\beta_2}$, and $\alpha_{a_6} := \frac{6\alpha_M}{\sqrt{1-\beta_2}} + 2\alpha^2 B^2 e^{2B\eta_0}$. Under the assumption that $\mathcal{L}(\boldsymbol{W}_t) \leq \frac{\log 2}{n}$ for all $t \geq \tilde{t}$, we have for all $t \geq \tilde{t}_A := \max\{t_A, \tilde{t}\}$ (recall that $t_A = \frac{2\log(\frac{\sqrt{1-\beta_2}}{4})}{\log(\beta_1)}$)

$$\mathcal{L}(\boldsymbol{W}_{t+1}) \overset{(a)}{\leq} \mathcal{L}(\boldsymbol{W}_t) - \eta_t \gamma \mathcal{G}(\boldsymbol{W}_t) + \alpha_{a_4} \beta_1^{\frac{t}{2}} \gamma \eta_t \mathcal{G}(\boldsymbol{W}_t) + \alpha_{a_5} d\eta_t^{\frac{3}{2}} \mathcal{G}(\boldsymbol{W}_t) + \alpha_{a_6} d\eta_t^2 \mathcal{G}(\boldsymbol{W}_t)$$

$$\overset{(b)}{\leq} \mathcal{L}(\boldsymbol{W}_t)\left(1 - \eta_t \gamma \frac{\mathcal{G}(\boldsymbol{W}_t)}{\mathcal{L}(\boldsymbol{W}_t)} + \alpha_{a_4}\beta_1^{\frac{t}{2}}\gamma\eta_t + \alpha_{a_5} d\eta_t^{\frac{3}{2}} + \alpha_{a_6} d\eta_t^2\right)$$

$$\leq \mathcal{L}(\boldsymbol{W}_{\tilde{t}_A}) \exp\left(-\gamma \sum_{s=\tilde{t}_A}^{t} \eta_s \frac{\mathcal{G}(\boldsymbol{W}_s)}{\mathcal{L}(\boldsymbol{W}_s)} + \alpha_{a_4}\gamma \sum_{s=\tilde{t}_A}^{t} \beta_1^{\frac{s}{2}}\eta_s + \alpha_{a_5} d \sum_{s=\tilde{t}_A}^{t} \eta_s^{\frac{3}{2}} + \alpha_{a_6} d \sum_{s=\tilde{t}_A}^{t} \eta_s^2\right)$$

$$\overset{(c)}{\leq} \frac{\log 2}{n} \exp\left(-\gamma \sum_{s=\tilde{t}_A}^{t} \eta_s \frac{\mathcal{G}(\boldsymbol{W}_s)}{\mathcal{L}(\boldsymbol{W}_s)} + \alpha_{a_5} d \sum_{s=\tilde{t}_A}^{t} \eta_s^{\frac{3}{2}} + \alpha_{a_6} d \sum_{s=\tilde{t}_A}^{t} \eta_s^2 + \alpha_{a_7}\right),$$

where (a) is by Lemma 37, (b) is by $\frac{\mathcal{G}(\boldsymbol{W}_t)}{\mathcal{L}(\boldsymbol{W}_t)} \leq 1$ (shown in Lemma 18), and (c) is by $\mathcal{L}(\boldsymbol{W}_{\tilde{t}_A}) \leq \frac{\log 2}{n}$ and $\alpha_{a_4}\gamma \sum_{s=\tilde{t}_A}^{t} \beta_1^{\frac{s}{2}}\eta_s \leq \frac{\alpha_{a_4}\gamma\eta_0}{1-\beta_1^{\frac{1}{2}}} =: \alpha_{a_7}$. The rest of the proof follows the same arguments in Lemma 22. Namely, the assumption $\mathcal{L}(\boldsymbol{W}_t) \leq \frac{\log 2}{n}$ implies that $\min_{c \neq y_i}(\boldsymbol{e}_{y_i} - \boldsymbol{e}_c)^T \boldsymbol{W}_t \boldsymbol{h}_i \geq 0$ for all $i \in [n]$. This separability condition can be used further to show that for all $t \geq \tilde{t}_A$

$$e^{-\min_{i \in [n], c \neq y_i}(\boldsymbol{e}_{y_i}-\boldsymbol{e}_c)^T \boldsymbol{W}_t \boldsymbol{h}_i} \leq \exp\left(-\gamma \sum_{s=\tilde{t}_A}^{t-1} \eta_s \frac{\mathcal{G}(\boldsymbol{W}_s)}{\mathcal{L}(\boldsymbol{W}_s)} + \alpha_{a_5} d \sum_{s=\tilde{t}_A}^{t-1} \eta_s^{\frac{3}{2}} + \alpha_{a_6} d \sum_{s=\tilde{t}_A}^{t-1} \eta_s^2 + \alpha_{a_7}\right).$$

Taking the $\log$ on both sides leads to the final result. $\qquad\square$

Next lemma upper bounds the max-norm of Adam iterates. It involves showing that the risk upper bounds entry-wise second moment, which will become small after the risk starts to monotonically decrease. Its proof can be found in Zhang et al. [77, Lemma 6.4]. Here, we only show the steps that are specific in our settings.

**Lemma 39** (Adam $\|\boldsymbol{W}_t\|_{\max}$). *Under the same setting as Theorem 5, suppose that there exists $\tilde{t}_B > \log(\frac{1}{\omega})$ such that $\mathcal{L}(\boldsymbol{W}_t) \leq \frac{1}{\sqrt{4B^2 + \alpha_V \eta_0}}$ for all $t \geq \tilde{t}_B$, then we have*

$$\|\boldsymbol{W}_t\|_{\max} \leq \alpha_{a_8} \sum_{s=0}^{\tilde{t}_B - 1} \eta_s + \sum_{s=\tilde{t}_B}^{t-1} \eta_s + \|\boldsymbol{W}_0\|_{\max},$$

*where $\alpha_{a_8}$ is some constant that depends on $B$, $\beta_1$, and $\beta_2$.*

*Proof.* For any $c \in [k]$ and $j \in [d]$, we have for all $t \geq \tilde{t}_B$

$$\begin{aligned}
\boldsymbol{V}_t[c, j] &\overset{(a)}{\leq} (1 - \beta_2^{t+1}) \nabla \mathcal{L}(\boldsymbol{W}_t)[c, j]^2 + \alpha_V \eta_t \mathcal{G}(\boldsymbol{W}_t)^2 \\
&\leq \nabla \mathcal{L}(\boldsymbol{W}_t)[c, j]^2 + \alpha_V \eta_t \mathcal{G}(\boldsymbol{W}_t)^2 \\
&\overset{(b)}{\leq} 4B^2 \mathcal{G}(\boldsymbol{W}_t)^2 + \alpha_V \eta_0 \mathcal{G}(\boldsymbol{W}_t)^2 \\
&\overset{(c)}{\leq} (4B^2 + \alpha_V \eta_0) \mathcal{L}(\boldsymbol{W}_t)^2 \overset{(d)}{\leq} 1,
\end{aligned}$$

where (a) is by Lemma 34, (b) is by Lemma 16, (c) is by Lemma 18, and (d) is by the assumption. This implies that for all $t \geq \tilde{t}_B$

$$0 \geq \log(\boldsymbol{V}_t[c, j]) \geq \log(\beta_2^t (1 - \beta_2) \mathcal{L}(\boldsymbol{W}_t)[c, j]^2) \overset{(e)}{\geq} t \log(\beta_2) + \log(1 - \beta_2) + \log(\omega),$$

where (e) is by the Assumption 5. The rest proof follows the same arguments in Zhang et al. [77, Lemma 6.4]. $\qquad \square$

**Theorem 5.** *Suppose that Assumption 1, 2, 3, 4, and 5 hold, and $\beta_1 \leq \beta_2$, then there exists $t_{a_2} = t_{a_2}(n, d, \gamma, B, \boldsymbol{W}_0, \beta_1, \beta_2, \omega)$ such that Adam achieves the following for all $t > t_{a_2}$*

$$\left| \frac{\min_{i \in [n], c \neq y_i} (\boldsymbol{e}_{y_i} - \boldsymbol{e}_c)^T \boldsymbol{W}_t \boldsymbol{h}_i}{\|\boldsymbol{W}_t\|_{\max}} - \gamma \right| \leq \mathcal{O}\left( \frac{\sum_{s=t_{a_2}}^{t-1} \eta_s e^{-\frac{\gamma}{4} \sum_{\tau=t_{a_2}}^{s-1} \eta_\tau} + \sum_{s=0}^{t_{a_2}-1} \eta_s + d \sum_{s=t_{a_2}}^{t-1} \eta_s^{3/2}}{\sum_{s=0}^{t-1} \eta_s} \right).$$

*Proof.* **Determination of $t_{a_1}$.** Here, we consider learning rate schedule of the form $\eta_t = \Theta(\frac{1}{t^a})$ where $a \in (0, 1]$. We choose $t_{a_1}$ after $(\max\{t_0, t_A, \log(\frac{1}{\omega})\}$ where $t_0$ satisfies Assumption 3 and $t_A = \frac{2 \log(\frac{\sqrt{1-\beta_2}}{4})}{\log \beta_1})$ such that the following conditions are met: $\alpha_{a_1} \beta_1^{t/2} \leq \frac{1}{6}$, $\alpha_{a_2} d \eta_t^{1/2} \leq \frac{1}{6}$, and $\alpha_{a_3} d \eta_t \leq \frac{1}{6}$. Concretely, we can set $t_{a_1} = \max\{\frac{-2 \log(6\alpha_{a_1})}{\log \beta_1}, (36\alpha_{a_2}^2 d^2)^{1/a}, (6\alpha_{a_3} d)^{1/a}\} = \Theta(d^{2/a})$. Then, we have for all $t \geq t_{a_1}$

$$\mathcal{L}(\boldsymbol{W}_{t+1}) \leq \mathcal{L}(\boldsymbol{W}_t) - \frac{\eta_t \gamma}{2} \mathcal{G}(\boldsymbol{W}_t). \tag{37}$$

Rearranging this equation and using non-negativity of the loss we obtain $\gamma \sum_{s=t_{a_1}}^{t} \eta_s \mathcal{G}(\boldsymbol{W}_s) \leq 2\mathcal{L}(\boldsymbol{W}_{t_{a_1}})$.

**Determination of $t_{a_2}$.** By Lemma 17, we can bound $\mathcal{L}(\boldsymbol{W}_{t_{s_1}})$ as follows

$$|\mathcal{L}(\boldsymbol{W}_{t_{a_1}}) - \mathcal{L}(\boldsymbol{W}_0)| \leq 2B \|\boldsymbol{W}_{t_{a_1}} - \boldsymbol{W}_0\|_{\max} \leq 2B \sum_{s=0}^{t_{a_1}-1} \eta_s \left\| \frac{\boldsymbol{M}_s}{\sqrt{\boldsymbol{V}_s}} \right\|_{\max} \leq 2B\alpha \sum_{s=0}^{t_{a_1}-1} \eta_s,$$

where the last inequality is by Lemma 33. Combining this with the result above and letting $\tilde{\mathcal{L}} = \min\{\frac{\log 2}{n}, \frac{1}{\sqrt{4B^2 + \alpha_V \eta_0}}\})$, we obtain

$$\mathcal{G}(\boldsymbol{W}_{t^*}) = \min_{s \in [t_{a_1}, t_{a_2}]} \mathcal{G}(\boldsymbol{W}_s) \leq \frac{2\mathcal{L}(\boldsymbol{W}_0) + 4B\alpha \sum_{s=1}^{t_{a_1}-1} \eta_s}{\gamma \sum_{s=t_{a_1}}^{t_{a_2}} \eta_s} \leq \frac{\tilde{\mathcal{L}}}{2} \leq \frac{1}{2n},$$

from which we derive the sufficient condition on $t_{a_2}$ to be $\sum_{s=t_{a_1}}^{t_{a_2}} \eta_s \geq \frac{4\mathcal{L}(\boldsymbol{W}_0)+8B\alpha \sum_{s=1}^{t_{a_1}-1} \eta_s}{\gamma\tilde{\mathcal{L}}}$.

**Convergence of** $\frac{\mathcal{G}(\boldsymbol{W}_t)}{\mathcal{L}(\boldsymbol{W}_t)}$ We follow the same arguments in the proof of NSD (Theorem 3) to conclude that

$$\frac{\mathcal{G}(\boldsymbol{W}_t)}{\mathcal{L}(\boldsymbol{W}_t)} \geq 1 - e^{-\frac{\gamma}{4} \sum_{s=t_{a_2}}^{t-1} \eta_s}. \tag{38}$$

We note that $t_{a_2}$ satisfies the assumptions in Lemma 38 and Lemma 39.

**Margin Convergence** Finally, we combine Lemma 38, Lemma 39, and (38) to obtain

$$\left|\frac{\min_{i\in[n],c\neq y_i}(\boldsymbol{e}_{y_i}-\boldsymbol{e}_c)^T\boldsymbol{W}_t\boldsymbol{h}_i}{\|\boldsymbol{W}_t\|_{\max}} - \gamma\right| \leq \mathcal{O}\left(\frac{\sum_{s=t_{a_2}}^{t-1} \eta_s e^{-\frac{\gamma}{4}\sum_{\tau=t_{a_2}}^{s-1}\eta_\tau} + \sum_{s=0}^{t_{a_2}-1} \eta_s + d\sum_{s=t_{a_2}}^{t-1} \eta_s^{3/2} + d\sum_{s=t_{a_2}}^{t-1} \eta_s^2}{\sum_{s=0}^{t-1} \eta_s}\right)$$

$$\leq \mathcal{O}\left(\frac{\sum_{s=t_{a_2}}^{t-1} \eta_s e^{-\frac{\gamma}{4}\sum_{\tau=t_{a_2}}^{s-1}\eta_\tau} + \sum_{s=0}^{t_{a_2}-1} \eta_s + d\sum_{s=t_{a_2}}^{t-1} \eta_s^{3/2}}{\sum_{s=0}^{t-1} \eta_s}\right)$$

$$\square$$

Similar to the case of NSD, we can derive the margin convergence rates for Adam.

**Corollary 2.** *Consider learning rate schedule of the form $\eta_t = \Theta(\frac{1}{t^a})$ where $a \in (0,1]$, under the same setting as Theorem 5, then we have for Adam*

$$\left|\frac{\min_{i\in[n],c\neq y_i}(\boldsymbol{e}_{y_i}-\boldsymbol{e}_c)^T\boldsymbol{W}_t\boldsymbol{h}_i}{\|\boldsymbol{W}_t\|_{\max}}-\gamma\right| = \begin{cases} \mathcal{O}\left(\frac{dt^{1-\frac{3a}{2}}+nd^{\frac{2(1-a)}{a}}+n\mathcal{L}(\boldsymbol{W}_0)+[\log(1/\omega)]^{1-a}}{t^{1-a}}\right) & \text{if} \quad a < \frac{2}{3} \\ \mathcal{O}\left(\frac{d\log(t)+nd+n\mathcal{L}(\boldsymbol{W}_0)+[\log(1/\omega)]^{1/3}}{t^{1/3}}\right) & \text{if} \quad a = \frac{2}{3} \\ \mathcal{O}\left(\frac{d+nd^{\frac{2(1-a)}{a}}+n\mathcal{L}(\boldsymbol{W}_0)+[\log(1/\omega)]^{1-a}}{t^{1-a}}\right) & \text{if} \quad \frac{2}{3} < a < 1 \\ \mathcal{O}\left(\frac{d+n\log(d)+n\mathcal{L}(\boldsymbol{W}_0)+\log\log(1/\omega)}{\log t}\right) & \text{if} \quad a = 1 \end{cases}$$

*Proof.* Recall that $t_{a_1} = \Theta(d^{2/a}) = C_{a_1} d^{2/a}$, and the condition on $t_{a_2}$ is $\frac{2\mathcal{L}(\boldsymbol{W}_0)+4B\alpha \sum_{s=1}^{t_{a_1}-1} \eta_s}{\gamma \sum_{s=t_{a_1}}^{t_{a_2}} \eta_s} \leq \frac{\tilde{\mathcal{L}}}{2}$, where $\tilde{\mathcal{L}} = \min\{\frac{\log 2}{n}, \frac{1}{\sqrt{4B^2+\alpha_V \eta_0}}\}$. Then, we apply integral approximations and the rest of the proof can be found in [77, Corollary 4.7 and Lemma C.1]. $\square$

**Remark 3.** *These rates match exactly those in the binary case of Zhang et al. [77] with logarithmic dependence on the initialization parameter $\omega$ (Ass. 5). This is only made possible through the fine-grained per-class bounding of the first and second moments using both $\mathcal{G}_c(\boldsymbol{W})$ and $\mathcal{Q}_c(\boldsymbol{W})$. Note that Lemma 36 takes the same form as Zhang et al. [77, Lemma A.4]. However, without the tight per-class bound and the equivalent decomposition of $\mathcal{G}(\boldsymbol{W})$ using either $\mathcal{Q}_c(\boldsymbol{W})$ or $\mathcal{G}_c(\boldsymbol{W})$, an extra factor of $k$ would appear. Interestingly, our rates for SignGD in Corollary 1 reveal a theoretical gap: Adam's optimal choice $a = \frac{2}{3}$ yields $\mathcal{O}(\frac{d\log(t)+nd}{t^{1/3}})$ while SignGD achieves $\mathcal{O}(\frac{\log(t)+n}{t^{1/2}})$ with $a = \frac{1}{2}$. Despite achieving tightness w.r.t. class-dimension ($k$), this gap emerges from our entry-wise analysis of the $\clubsuit$ term in Lemma 36 across the feature dimension ($d$) using scalar functions $\mathcal{G}_c(\boldsymbol{W}), \mathcal{Q}_c(\boldsymbol{W})$. Closing this theoretical gap–revealed through our NSD analysis–that also appears in the binary case [77], forms an important direction for future work.*

**Numerical Validations** We test the margin converge of Adam with different stability constants ($\epsilon$) chosen from the set $\{0, 10^{-6}, 10^{-7}, 10^{-8}\}$. We make the following observations: **(1) SignGD/Adam vs NGD:** SignGD and Adam(with zero stability) iterates favor the max-norm margin over the 2-norm margin. The opposite is true for NGD (Figs. 5c, 5d); **(2) Speed of convergence:** For the learning rate schedules considered, the margin convergence of SignGD is faster than that of Adam (with zero stability constant), consistent with our theoretical results (Fig. 5d); **(3) Effect of Adam's stability constant:** For Adam with non-zero stability constants, when the magnitude of the gradient (or the second moment) is above the stability constant, the convergence to the max-norm margin is favored over that to the 2-norm margin. However, when the gradient values approach or fall below the stability constant, the max-norm margin starts to decrease while the 2-norm margin increases. This is shown in Figs. 6a and 6b (compared against the case of zero stability constant in which the max-norm margin

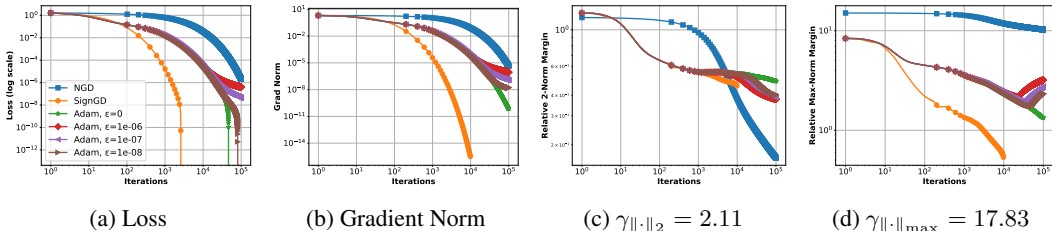

(a) Loss          (b) Gradient Norm          (c) $\gamma_{\|\cdot\|_2} = 2.11$          (d) $\gamma_{\|\cdot\|_{\max}} = 17.83$

Figure 5: Implicit bias of NGD, SignGD, and Adam on multiclass separable data ($k = 5$, $d = 25$, and 50 data points in each class). **(a,b)** Loss and gradient 2-norm vs. iterations: SignGD converges faster than others. **(c)** We normalize the iterates w.r.t. 2-norm (aka Frobenius), compute the margin, then plot its difference to the dataset's max-margin w.r.t. 2-norm (given in captions). Only NGD converges to the max 2-norm margin. **(d)** Same as **(c)** with 2-norm replaced by max-norm. Margins of SignGD/Adam(with $\epsilon = 0$) converge to max-margin w.r.t max-norm. For SignGD, the training is stopped after $10^4$ iterations due to the numerical instabilities caused by the small gradient norm.

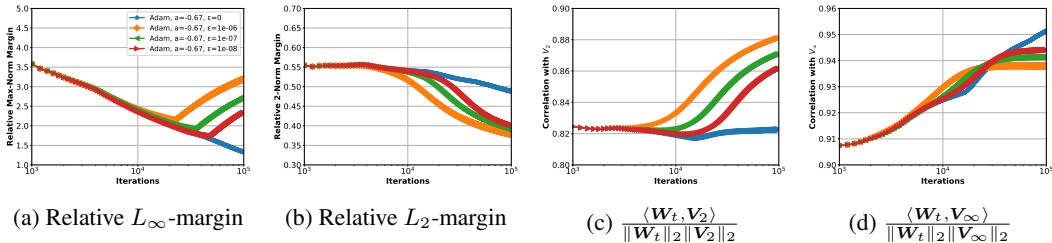

(a) Relative $L_\infty$-margin     (b) Relative $L_2$-margin     (c) $\frac{\langle \boldsymbol{W}_t, \boldsymbol{V}_2 \rangle}{\|\boldsymbol{W}_t\|_2 \|\boldsymbol{V}_2\|_2}$     (d) $\frac{\langle \boldsymbol{W}_t, \boldsymbol{V}_\infty \rangle}{\|\boldsymbol{W}_t\|_2 \|\boldsymbol{V}_\infty\|_2}$

Figure 6: Effects of non-zero stability constants $\epsilon$ for Adam. We focus on late training stage between iteration $10^3$ to iteration $10^5$. **(a)** Gradient 2-norm vs. iterations. **(b)** Same quantity as Figure 5d. We observe the max-norm margin of non-zero stability constants decreases after gradient magnitudes approach or fall below the values of the constants (drawn as horizontal lines in (a)). **(c)** Same quantity as Figure 5c. After $10^4$ iterations, the non-zero stability constants start to increase towards the max 2-norm margin. **(d, e)** Correlations between $\boldsymbol{W}_t$ and max-margin classifiers $\boldsymbol{V}_2$, $\boldsymbol{V}_\infty$ against iterations. Considering correlations to $\boldsymbol{V}_2$, its value stays nearly constant for the zero stability constant, whereas rise after $10^4$ iterations for non-zero ones. We also observe the transitions occur earlier for larger stability constants. Considering correlations to $\boldsymbol{V}_\infty$, the values rise and plateau for non-zero stability constants. However, its value keeps increasing for the zero stability constant.

keeps increasing). Moreover, the max-norm to 2-norm margin transition occurs earlier in training for larger stability constants. The experiments in Figs 6d and 6c further confirm this two-phase behavior given the correlations to the 2-norm separator $\boldsymbol{V}_2$ rise while those to the max-norm separator $\boldsymbol{V}_\infty$ plateau in the late training stage. These experiments confirm that the results of Wang et al. [68] also hold in the multiclass setting provided that the stability constant is non-zero and the magnitudes of the second moment are small (compared against the stability constant). Note, however, that the stability constant is typically chosen to be very small ($\epsilon \sim 10^{-8}$) and the training is not long enough until all gradients are below this value. Thus, the max-norm margin convergence results are practically relevant.

