# OpenReview forum: "Implicit Bias of Spectral Descent and Muon on Multiclass Separable Data"
_NeurIPS.cc/2025/Conference — NeurIPS 2025 spotlight_

### Official Review · Reviewer_VTFD · 2025-06-21

**Clarity:** 4
**Significance:** 3
**Originality:** 3
**Rating:** 5
**Confidence:** 3

**Summary:**

This paper proves implicit bias results for linear multi-class classification on linearly separable data with respect to various matrix norms. Specificially, it considers norms given by entry-wise p-norms and Schatten norms (p-norms of the set of singular values of the matrix) with and without momentum. In each of these cases, the parameters converge in direction to a max-margin solution with respect to the corresponding norm.

**Questions:**

- Isn't Assumption 4 satisfied automatically since the dataset is finite? Is the purpose of this assumption just to introduce the notation $B = \max_i \|h_i\|_1$?
- I find it surprising that the learning rate needs to decrease over time to prove convergence in direction, as the proof of implicit bias for linear models with binary data works with a constant step size. Do you have any intuition as to which part of your setting makes this necessary?

**Ethical Concerns:**

["NO or VERY MINOR ethics concerns only"]

**Final Justification:**

No changes to my score. The weakness of linearly separable data remains, but it is not a big problem considering how common of an assumption it is. The authors addressed my questions.

**Limitations:**

yes

**Quality:**

4

**Strengths And Weaknesses:**

### Strengths
- The results are comprehensive and consider a variety of different optimization methods, unifying them under the umbrella of normalized steepest descent under different matrix norms.
- Quantitative rates of how quickly the margin shrinks in the number of iterations of gradient descent.
- The paper is well-written and easy to follow. It includes proofs/sketches of intermediate steps within the main text.
- The paper describes how specific quantities tracked in the proof, such as $\mathcal{G}(\mathbf{W}),$ are modified from the univariate case. This intuition could be helpful for future results for multiclass classification.

### Weaknesses
- The dataset is assumed to be linearly separable and only linear models are considered. The types of functions that get learned by a neural network for linearly separable versus non-linearly separable data are qualitatively different, so this limits how much we can interpret from the results. But this is a standard assumption when studying the implicit bias of gradient descent for classification, so this is not a major weakness.

Small typos:
- In table 1: "Refernce" -> "Reference".
- On line 136, no comma after "algorithms".
- On line 154, $\mathbb{S}'(\mathbf{v})$ does not match its original definition.
- In equation (6), $o(\|\Delta\|_F^3)$ -> $O(\|\Delta\|_F^3)$.

---

> ### Author Rebuttal · Authors · 2025-07-30
>
> We sincerely thank you for your positive assessment of our work. Here, we address your comments/questions in sequence:
>
> ***Linear setting*** We provide a first complete characterization of the implicit bias of NSD/NMD algorithms on multiclass data, and we believe the insights from our work can be helpful for future studies on the non-linear models.
>
> ***Feature-norm upper bound*** You are right: Here, we introduce $B$ as the upper bound on the maximum $L_1$-norm of the features. For finite dataset in a finite-dimensional space, such a $B$ always exists. We choose to state this as an assumption for clarity (particularly since $B$ appears throughout the analysis). We will add a remark about the existence of $B$ as mentioned above.
>
> ***Constant step-size*** For GD[1,2] and NGD[3], it is indeed true that the implicit bias can be shown with constant step-size for binary data, utilizing the underlying Euclidean geometry. However, the class of algorithms (including Schatten norms) that we consider is much wider and can have different geometries. It is challenging to unify their analyses under the same framework for multiclass data. We manage to do this by constructing a proxy function (that works for all p-norms) to bound both the first and second-order terms in the Taylor expansion of the loss. This results in a bound on the margin gap with explicit dependence on the learning rates as shown in Theorem 3 in Appendix. From there, we observe that the step size needs to satisfy Assumption 2 to ensure convergence. It remains interesting and (potentially) challenging to show the implicit bias of NSD/NMD  with constant step-size for the multiclass setting and for the range of p-norms that we consider (we will add this in the future work discussion). Note also that decreasing step-size has been used in other implicit bias works [4,5].
>
> ***Additional comments*** Thank you for pointing out these typos. We will fix them in the final version of the paper.
>
> [1] The Implicit Bias of Gradient Descent on Separable Data, Soudry et al
>
> [2] The implicit bias of gradient descent on nonseparable data, Ji and Telgarsky
>
> [3] Characterizing the implicit bias via a primal-dual analysis, Ji and Telgarsky
>
> [4] Convergence of Gradient Descent on Separable Data, Nacson et al
>
> [5] The Implicit Bias of Adam on Separable Data, Zhang et al
>
> Thank you again for your time and suggestions. Please let us know if there are any further questions.

---

> > ### Comment · Reviewer_VTFD · 2025-07-31
> >
> > Thank you for your response and for answering the questions I had. I will maintain my positive score.

---

### Official Review · Reviewer_cptr · 2025-06-23

**Clarity:** 4
**Significance:** 4
**Originality:** 4
**Rating:** 5
**Confidence:** 5

**Summary:**

This work analyzes the implicit bias of normalized steepest descent NSD (with and without moment. the version with moment is abbreviated as NMD) family of optimization algorithms for multiclass classification. as the authors point out, even for linear classification The multiclass setting is more rich due to the linear classifiers being matrices rather than vectors.
The NSD/NMD families encompass a broad range of gradient-based optimizers currently being studied by the community.

The authors focuses on spectral-descent, which contains Muon and Shampoo.
They show that normalized steepest descent w.r.t either an entrywise matrix norm or a Schatten matrix norm (Lp norm of the spectrum of the matrix) has implicit bias toward that same norm.

**Questions:**

Eqn (8) looks quite different from the binary one cited on line 172. could you help elucidate/reconcile this difference, at least on an intuitive level?

Theorem 1 gives the convergence in margin. How about convergence in direction? Does that also hold?

Suggestions:
in the paragraph beginning on line 158, the word "former" is used twice and is a bit hard as a reader to parse. consider replacing the word with the actual things the word "latter" were intended to reference.

consider adding a brief comment about how line 211 elementary fact is used to get Property (ii) in Lemma 3

**Ethical Concerns:**

["NO or VERY MINOR ethics concerns only"]

**Final Justification:**

I am happy with the authors' rebuttal and will maintain my positive score and support for the submission.

**Limitations:**

Yes

**Paper Formatting Concerns:**

No concern

**Quality:**

4

**Strengths And Weaknesses:**

Strength:
The authors nicely fill in a gap in the multiclass analysis.
Thus, there are richer set of norms/regularizers on these matrix linear classifiers, compared to the binary case.
The analysis is clean and slick.

Weaknesses:
Theorem 1 gives the convergence in margin. How about convergence in direction? Does that also hold?
There can be a better comparison with the two previous works Zhang et al. [63] and Xie and Li [59] on line 284 and line 289. How does this work's result compare to those obtained in the two cited papers? Are they better/worse/not comparable?

---

> ### Author Rebuttal · Authors · 2025-07-30
>
> We sincerely thank you for your positive assessment of our work. Here, we address your comments/questions in sequence:
>
> ***Comparison to previous works*** We have provided discussions on the convergence rates of Adam (and comparisons to Zhang et al.) in Appendix G, see Remark 3. Here, we reemphasize the key differences on the results/scope of our work and others. The work by Zhang et al. studied Adam in the binary setting [1], instead we study a broad family of NSD/NMD algorithms in the multiclass setting. As a byproduct of our NMD analysis, we get a rate for Adam in the multiclass setting that matches the rate of Zhang et al. In fact, direct use of their result would introduce an extra factor of $k$ (number of classes) into the bound, we are able to avoid this by the tight decomposition of the proxy function (below Lemma 6). Regarding Xie and Li [2], their non-linear setting and asymptotic convergence (to a KKT point of a constrained optimization problem) differ from our setting and the non-asymptotic rate.
>
> ***Proxy function*** In the binary case, the proxy function is defined using gradients. As loss decreases, the proxy function converges to $0$ as gradients become small. In the multiclass setting, the logit corresponding the true class saturates as loss decreases. Hence, the proxy function for the multiclass setting also converges to $0$.
>
> ***Parameter convergence*** Parameter convergence (and its rate) is more general to show and we can’t conclude formally from our proof although the simulation seems to suggest that this is the case. Note that the more subtle issue is that for certain norms of interest (e.g., max-norm (SignGD) and spectral-norm (specGD/Muon)), the max-margin problem does not necessarily admit a unique minimizer, which further complicates parameter convergence. Moreover, the parameter convergence of NGD (i.e. NSD w.r.t 2-norm) utilizes the properties of Euclidean geometry [3]. Here, the algorithms (and their associated norms) that we consider are much wider and beyond NGD.
>
> ***Additional comments*** Thank you for your suggestions regarding line 158 and 211. We will revise them in the final version of the paper.
>
> [1] The Implicit Bias of Adam on Separable Data, Zhang et al
>
> [2] Implicit Bias of AdamW: $L_\infty$ Norm Constrained Optimization, Xie and Li
>
> [3] Characterizing the implicit bias via a primal-dual analysis, Ji and Telgarsky
>
> Thank you again for your time and suggestions. Please let us know if there are any further questions.

---

> > ### Comment · Reviewer_cptr · 2025-08-04
> >
> > I thank the authors for a thorough rebuttal. I will maintain my positive score and support for the submission.

---

### Official Review · Reviewer_KwWH · 2025-06-29

**Clarity:** 3
**Significance:** 3
**Originality:** 3
**Rating:** 4
**Confidence:** 4

**Summary:**

This paper provides a comprehensive theoretical characterization of the implicit bias of a wide family of normalized steepest descent (NSD) and momentum descent (NMD) algorithms in the setting of multiclass linear classification with cross-entropy loss. It includes special cases such as Spectral Descent and Muon, both of which are shown to converge to the maximum-margin classifier with respect to the spectral norm. The authors present a unified analysis framework using a carefully constructed proxy function $G(W)$ to track both the loss and margin convergence, and they further extend their results to Adam. Experimental results validate the theoretical claims.

**Questions:**

1. In Figure 2, your results show that Muon and Spectral-GD converge to the same solution $V_{\text{spec}}$, right? Within your theoretical framework, what is the implicit bias of the momentum term? Is it possible that the linear model setup is too simple to reveal any difference? As we know, in general nonlinear networks, momentum-based methods like Adam and SGD often prefer different solutions.

2. During optimization, we typically do not constrain the norm of the parameter matrix. So in Theorem 1, is the margin gap result obtained by normalizing with $\|W_t\|$? In the caption of Figure 1, how exactly is $\gamma_{\bar{W}_t}$ computed? Is it also a normalized result?

3. For Adam, you consider the version without the stability constant. In practice, Adam with $\varepsilon$ is more common. Why was $\varepsilon$ omitted, and how would including it affect the implicit bias?

4. Do you expect your convergence results for NSD/NMD to generalize to nonlinear models (e.g., two-layer MLPs)? Can the proxy function idea be extended to such cases?

5. How sensitive are your results to non-separable data? Is there still an implicit bias guarantee when separability only approximately holds (e.g., with margin slack)?

Other minor comments:

1. Table 1: "Refernce" → should be "Reference".

2. The symbol $e_{y_i}$ in Equation (2) appears to be a standard one-hot vector, but this is not explained upon its first appearance.

**Ethical Concerns:**

["NO or VERY MINOR ethics concerns only"]

**Final Justification:**

Resolved Issues:

Technical issues have been addressed, including clarification of additional experimental details.

Unresolved Issues:

The work lacks detailed discussion and analysis of nonlinear model results.

While most of my questions have been adequately addressed, the paper remains constrained to linear classifier models. This presents a significant gap when considering nonlinear models, limiting the broader applicability of the findings.

Overall, I maintain my score of 4.

**Limitations:**

The experimental scope is limited—restricted to small-scale, linearly separable synthetic data.

**Quality:**

3

**Strengths And Weaknesses:**

**Strengths**

1. Despite being set in a relatively simple setting, the paper develops an elegant and reusable framework via a proxy function  $G(W)$ that enables generalization across a broad family of optimization algorithms.

2. Muon has recently become popular, and the authors provide new insights into the implicit bias of spectral descent and Muon in multiclass linear settings.

3. The paper is well-organized, with a clear exposition of the background. The related work is thoroughly cited, the theoretical results are solid, and they are supported by experiments. The writing and logical flow are excellent.

**Weaknesses**

1. Strong Linearity Assumption: All analysis is restricted to linear classifier models, which limits the applicability to real-world nonlinear neural networks. There is no discussion of whether or how the results might extend to nonlinear models.

2. Empirical Results Are Limited: The experiments are only conducted on synthetic linearly separable datasets. Whether the approximate implicit bias predicted by theory still holds in more general settings remains an open question worth further exploration. That said, I believe the theoretical contributions of this paper are already substantial.

---

> ### Author Rebuttal · Authors · 2025-07-30
>
> We sincerely thank you for your positive assessment of our work. Here, we address your comments/questions in sequence:
>
> ***Effects of momentum*** For the setting considered in this work, momentum does not change the implicit bias in terms of margin convergence w.r.t a specific norm. Indeed, Figure 1,2 reveal that both spectral-GD and Muon favor the spectral-norm margin over the others. We are not making any claims regarding the non-linear setting. One thing to note is that preconditioning may also play a role in the implicit-bias difference between Adam and GD.
>
> ***Normalization*** In Thm 1, the norm used for normalizing the weights is the one used to define the algorithm (eqn. (3)), and the margin (denoted as $\gamma$) is defined w.r.t. the same norm (eqn. (2)). For example, the margin of (normalized) iterates of SignGD (i.e. NSD w.r.t max-norm) converge to the data margin defined w.r.t the max-norm according to Thm 1. We want to emphasize that normalization in Thm 1 has nothing to do with modifying the algorithm. The algorithm runs as if it has no norm constraints. But because the norm of the parameters diverges in training, normalization is necessary to show convergence. In experiments, we compute the margin of normalized iterates (denoted as $\gamma_{\bar{W}_t}$) where the norm used in normalization and defining the data margin are the same. For example, the iterates of SignGD can be normalized w.r.t max-norm, Frobenius-norm, or spectral-norm. Regardless of which norm is used, same norm needs to be used to compute the data margin when plotting their differences. We are doing this for the purpose of demonstrating norm-preferences of a chosen NSD/NMD algorithm.
>
> ***Stability Constant*** The stability constant ($\epsilon$) can play a role in the implicit bias of Adam. Adam can favor a $L_{2}$-norm margin with a non-negligible $\epsilon$ [1]. However, this happens when the values of gradients are below $\epsilon$ (which is often chosen to be very small such as $10^{-8}$). In practice, training is typically not long enough for this realization to occur. Thus, Adam with negligible $\epsilon$ (i.e. max-norm solution favorance) is still practically relevant. We have provided a detailed discussion on the effect of $\epsilon$ in Appendix G (see pg 39,40). Our experiments in Figure 5 (in Appendix) demonstrate the transition from max-norm to $L_{2}$-norm margin favorance (after extended training), and hence consolidate that the findings of [1] and [2] also hold in the multi-class setting.
>
> ***Non-linear settings*** Despite the analysis is done in the linear setting, our work is the first to completely characterize the implicit bias of NSD/NMD algorithms for multiclass data. The extension to the non-linear settings is challenging and we believe that our theoretical techniques/insights can be helpful for such future studies. To motivate such theory extensions, we have performed (and will include in the final version of the paper) the following non-linear experiment: train a 2-layer MLP (with Relu activation denoted as $\sigma(\cdot)$ and hidden dimension being 100) on MNIST dataset (100 data points sampled from each class) and track the spectral-norm margin of the model defined as $\min_{i \in [n], c \neq y_i} \frac{(e_{y_i} - e_{c})^T W \sigma(H x_i)}{||| H |||_2}$ (note that $H$ and $W$ denote the weights of the first and second-layer respectively, and we only train the first layer). We observe that the spectral-norm margin of spectral-GD grows the fastest among the algorithms considered, suggesting that the implicit bias to respective norm (demonstrated with spectral-norm) is preserved in these non-linear settings.
>
> Table: Spetral-norm margin of SignGD, NGD, and Spectral-GD at different iterations
> |        |  $t=5000$     | $t=10000$    | $t=15000$   | $t=20000$  |
> |--------|--------|--------|--------|--------|
> | SignGD   |  0.1292 | 0.1248 | 0.1138 | 0.1056 |
> | NGD | 0.1643 | 0.1766 | 0.1821 | 0.1820 |
> | Spectral-GD  |  0.2474 | 0.2920| 0.3142 |  0.3280 |
>
> ***Non-separable dataset*** For the non-separable data setting (e.g. [3]), we would expect margin convergence (w.r.t a specific norm) to hold in the separable subspace. This type of implicit bias has been shown for GD (as a follow-up to implicit bias on the separable binary data) [3]. Our work makes it possible to extend the analysis to NSD on multiclass data, which is an interesting future direction. We will add this in the future work discussion.
>
> ***Additional comments*** Thank you for these comments. We will fix them in the final version of the paper.
>
> [1] Does Momentum Change the Implicit Regularization on Separable Data, Wang et al
>
> [2] The Implicit Bias of Adam on Separable Data, Zhang et al.
>
> [3] The implicit bias of gradient descent on nonseparable data,  Ji and Telgarsky
>
> Thank you again for your time and suggestions. Please let us know if there are any further questions.

---

> > ### Comment · Reviewer_KwWH · 2025-08-02
> >
> > I appreciate the author's response, most of my questions have been addressed, and I maintain my positive score.

---

### Official Review · Reviewer_ZRz3 · 2025-07-03

**Clarity:** 3
**Significance:** 3
**Originality:** 3
**Rating:** 5
**Confidence:** 4

**Summary:**

The authors characterize the implicit bias of p-norm normalized steepest descent and its momentum variant for multiclass linear classification with cross-entropy loss, proving that both families of algorithms converge in direction to maximum-margin classifiers defined by the same norm. They derive explicit non-asymptotic convergence rates of $O(1/\sqrt{t})$ for NSD and NMD. They further show that Adam converges to the $\ell_\infty$-margin solution under an appropriate step size.

**Questions:**

1. How sensitive are the convergence rates and maximum-margin solutions to the choice of p in the p-norm normalization?
Can you specify how the hidden constants in the $O(1/\sqrt{t})$ convergence rates depend on the choice of p?

2. Have you conducted any empirical validation to confirm that the theoretical convergence rates or implicit bias manifest in real datasets?

3.  What intuition can you provide on which norm normalization is most beneficial under different data or task conditions?

4. How do the results change when we deal with non-linear models in experiments?

**Ethical Concerns:**

["NO or VERY MINOR ethics concerns only"]

**Final Justification:**

As I described the paper is good and I maintain the positive score

**Limitations:**

While this work provides finite-time implicit-bias guarantees for p-norm normalized steepest descent (NSD) and its momentum variant (NMD), its scope is limited. The analysis is confined to fully linear, multiclass separable problems trained with exact, full-precision gradient updates and cross-entropy loss, leaving open the question of whether these results extend to deep or non-linear architectures, as well as stochastic or mini-batch training.  I understand the complexity of the theoretical analysis with more complicated settings, but it would be nice to conduct the experiments to explore whether these results can be extended and made more practically relevant.

**Quality:**

3

**Strengths And Weaknesses:**

The paper’s key contributions stand out in its combination of the finite‐time analysis, a unifying optimization framework, and empirical confirmation:

Strengths:
- The authors provide the first non‐asymptotic implicit‐bias guarantees for normalized steepest descent (NSD) and its momentum variant (NMD) on linearly separable multiclass classification tasks.
- The authors consider a broad algorithmic family—p-norm normalized steepest descent (NSD) and its momentum variant (NMD)—which includes practical optimizers such as Muon and  Adam, unifying their implicit-bias dynamics under the steepest descent with unit balls.
- They provide explicit non-asymptotic convergence rates of $O(1/\sqrt{t})$ for NSD and NMD.
- They present rigorous, non-asymptotic proofs that connect the algorithm's convergence directly to maximum-margin classifiers.
- They also empirically verify that signGD, NGD, spectral descent, and Muon converge in direction to their corresponding maximum-margin solutions.

Weaknesses:
- Analysis is limited to linear models with cross-entropy loss; extensions to non-linear or deep architectures are not addressed.
- No empirical validation is provided to confirm the theoretical convergence rates or practical impact on real datasets.

---

> ### Author Rebuttal · Authors · 2025-07-30
>
> We sincerely thank you for your positive assessment of our work. Here, we address your comments/questions in sequence:
>
> ***Other Losses*** The convergence rates also hold for other losses. We have discussed these extensions in Appendix F.
>
> ***Dependence of p*** The margin $\gamma$ naturally depends on p via its definition in eqn. (2), and the convergence rates in Thm 1 and 2 hold for any p. Our current analysis does not explicitly track how the constant depends on p. However, inspecting the analysis steps in Sec. 3 (i.e. lower and upper bounding the first and second-order Taylor terms respectively) reveal some insights. First, for all values of p, the dual-norm of the gradients is lower bounded by the same proxy function multiplied by the margin $\gamma$. The proxy function itself has no p-dependence. When it is used to upper bound the second-order term (see the display after Line 198), the bound depends on the norm of the features. Currently, we are using the upper bound on $L_1$-norm of the features, denoted as $B$ (i.e. $max\_{i \in [n]} \lVert h_i \rVert_1 \leq B, \forall i \in [n]$). The constant $B^2$ can be substituted by $max\_{i \in [n]} |||h_i h_i^T|||_{p}$ (note that this reduces to $max\_{i \in [n]} \lVert h_i \rVert_p^2$ for entry-wise norms, see lines 195-197), which captures the geometry of the features. We will include this comment in the final version of the paper.
>
> ***Intuition on the choice of norms*** This is a good question and in-part is one of the motivations behind implicit-bias analyses like our paper. Specifically, having shown that different norms in NSD/NMD maximize margins of the dataset w.r.t corresponding norms, facilitates the generalization question: which p-norm generalizes better? Of course, the answer depends on the data in complicated ways. In the linear case, this can be thought as follows: p-norm NSD favors convergence to a classifier that maximizes p-norm margin. Let the underlying data distribution be parameterized by a matrix which can represent the parameters of multinomial logistic model or the mean-vectors of multiclass Gaussian mixture (as an example). If the matrix that generates the data is sparse, SignGD is preferred; if it has a balanced spectrum, Spectral-GD is preferred; if it is low-rank,  low-rank GD is preferred and so on. While we recognize the limitations of this setting, it can provide some initial intuitions towards the choices of algorithms in different cases.
>
> ***Experiments (real datasets)/theory on non-linear models*** We performed additional experiments on MNIST dataset (100 data points sampled from each of the 10 classes) in the  non-linear multi-class setting. We use a two-layer MLP with random initializations and the hidden dimension being 100. We only train the first layer and track the spectral-norm margin (as an example) of the overall network defined as follows: $\min_{i \in [n], c \neq y_i} \frac{(e_{y_i} - e_{c})^T W \sigma(H x_i)}{||| H |||_2}$, where we denote the first and second-layer weights as $H$ and $W$ respectively, and $\sigma(\cdot)$ is the Relu activation function. We observe that the spectral-norm margin of Spectral-GD grows the fastest among the three algorithms considered (SignGD, NGD, and Spectral-GD). This confirms that the norm preferences of NSD algorithms (exhibited in the linear settings) can also show up in the non-linear settings. We will incorporate these experiments in the final version of the paper. From theoretical perspectives, despite being in the linear setting, we are the first to formally prove the implicit bias of NSD/NMD for the multiclass data. The extension to the non-linear (multiclass) settings is challenging and we believe our work provides the ground for such future studies.
>
> Table: Spetral-norm margin of SignGD, NGD, and Spectral-GD at different iterations
> |        |  $t=5000$     | $t=10000$    | $t=15000$   | $t=20000$  |
> |--------|--------|--------|--------|--------|
> | SignGD   |  0.1292 | 0.1248 | 0.1138 | 0.1056 |
> | NGD | 0.1643 | 0.1766 | 0.1821 | 0.1820 |
> | Spectral-GD  |  0.2474 | 0.2920| 0.3142 |  0.3280 |
>
> Thank you again for your time and suggestions. Please let us know if there are any further questions.

---

> > ### Comment · Reviewer_ZRz3 · 2025-08-04
> >
> > Thank you for the author's responses. I have no further questions.

---

### Decision · Program_Chairs · 2025-09-17

**Decision:**

Accept (spotlight)

**Comment:**

The reviewers agreed that the paper provides a comprehensive theoretical characterization of the implicit bias of a wide family of normalized steepest descent (NSD) and momentum descent (NMD) algorithms in the setting of multiclass linear classification with cross-entropy loss, including special cases such as Spectral Descent and Muon, both of which are shown to converge to the maximum-margin classifier with respect to the spectral norm.

The authors present a unified analysis using a carefully constructed proxy function to track both the loss and margin convergence, and they further extend their results to Adam, with experimental results to validate the theoretical claims. The NSD/NMD families encompass a broad range of gradient-based optimizers currently being studied by the community. The paper’s key contributions stand out in its combination of the finite‐time analysis, a unifying optimization framework, and empirical evaluation providing new insights into the implicit bias of spectral descent and Muon in multiclass linear settings, with excellent writing and logical flow.

There was clear support for the recommendation to accept this work, and the rebuttals contributed to confirm the support while offering commitments to implement some clarifications and improvements on the manuscript. I strongly encourage the authors to integrate the feedback received into the revised version of their paper.